# A cultivated planet in 2010: 2. the global gridded agricultural production maps

Qiangyi Yu[1], Liangzhi You[1,2], Ulrike Wood-Sichra[2], Yating Ru[2], Alison K. B. Joglekar[3], Steffen Fritz[4], Wei Xiong[5], Miao Lu[1], Wenbin Wu[1,*], Peng Yang[1,*]

[1]Key Laboratory of Agricultural Remote Sensing (AGRIRS), Ministry of Agriculture and Rural Affairs / Institute of Agricultural Resources and Regional Planning, Chinese Academy of Agricultural Sciences, Beijing 100081, China
[2]International Food Policy Research Institute (IFPRI), Washington DC, USA
[3]GEMS Agroinformatics Initiative, University of Minnesota, Saint Paul, Minnesota, USA
[4]International Institute for Applied Systems Analysis (IIASA), Laxenburg, Austria
[5]International Maize and Wheat Improvement Center (CIMMYT), Mexico, México

*Correspondence to*: Wenbin Wu (wuwenbin@caas.cn) and Peng Yang (yangpeng@caas.cn)

**Abstract.** Data on global agricultural production are usually available as statistics at administrative units, which does not give any diversity and spatial patterns thus is less informative for subsequent spatially explicit agricultural and environmental analyses. In the second part of the two-paper series, we introduce SPAM2010—the latest global spatially explicit datasets on agricultural production circa year 2010—and elaborate on the improvement of the SPAM (Spatial Production Allocation Model) dataset family since year 2000. SPAM2010 adds further methodological and data enhancements to the available crop downscaling modeling, which mainly include the update of base year, the extension of crop list and the expansion of sub-national administrative unit coverage. Specifically, it not only applies the latest global synergy cropland layer (see Lu et al., submitted to the current journal) and other relevant data, but also expands the estimates of crop area, yield and production from 20 to 42 major crops under four farming systems across a global 5 arc-minute grid. All the SPAM maps are freely available at the MapSPAM website (http://mapspam.info/), which not only acts as a tool for validating and improving the performance of the SPAM maps by collecting feedbacks from users, but also dedicates as a platform providing archived global agricultural production maps for better targeting the Sustainable Development Goals. In particular, SPAM2010 can be downloaded via an open-data repository (DOI: https://doi.org/10.7910/DVN/PRFF8V. IFPRI, 2019).

## 1 Introduction

Civilization is founded on the agricultural use of land (Fu and Liu, 2019), which remains as important today as it was 10 000 years ago (Lev-Yadun et al., 2000). Agricultural land, which refers to the land area that is arable, under permanent crops, and under permanent meadows and pastures according to Food and Agriculture Organization of United Nations (FAO), is currently 4.9 billion hectares (FAOSTAT, 2019). This is 37.6% of the earth's terrestrial surface—the largest use of land on

the planet. Historically, the agricultural use of land has transformed ecosystem patterns and processes across most of the terrestrial biosphere (Ellis et al., 2013). The way we use agricultural land will significantly determine whether we are able to solve the multiple challenges embodied in the 17 Sustainable Development Goals (SDGs), e.g., feeding the world's growing

population, mitigating climate change, and halting biodiversity loss (FAO, 2018;Ehrensperger et al., 2019). As the fundamental connection between people and the planet, the spatial-temporal characteristics of agricultural land is important for the anthroposphere and beyond, as such information allows us to undertake more responsive and evidence-based analysis on the interaction and better resource allocation across land, water, energy, and the environment.

Cropland mapping has made great progress in the past few decades and provided great support for global agricultural monitoring and assessment. For example, it allows us to be able to know where agriculture has infringed into natural ecosystems and where cropland has been taken as a consequence of urbanization (Chen et al., 2015;Gong et al., 2019). However, this type of work mainly focuses on the agricultural changes at the land cover level, without paying attention to the subtle characteristics at the land use and land management level (Verburg et al., 2011). These subtle level characteristics

related to agricultural production, ranging from crop allocation to land use intensity, are the core of agricultural management and have been proved to have equally important impacts on food systems (Sun et al., 2018;Pretty, 2018), climate systems (Searchinger et al., 2018;Bonan and Doney, 2018) and ecosystems (Peters et al., 2019;Poore and Nemecek, 2018). Yet data on global agricultural production are usually representative at national and sub-national administrative units (e.g., provinces, districts). This level of statistics does not give a sense of the diversity and spatial patterns in agricultural production and is

not spatially explicit which is critical for many environmental and ecological assessments (Yu et al., 2012).

There are a few attempts to develop global spatially explicit datasets on agricultural production by fusing censuses statistics with maps of agricultural land cover (Figure 1). Leff et al. (2004) applied a simplified proportional disaggregating approach and mapped the global harvested area of 18 major crops circa 1992. By using a similar approach, Monfreda et al. (2008)

mapped both harvested area and yield for a full coverage of 175 crops circa 2000 (the dataset is referred as M3 hereafter). Portmann et al. (2010) developed the MIRCA dataset that contains the harvested area for 26 crops circa 2000 by using M3 as a starting point. It further allocates the total harvested area for each crop into rainfed and irrigated areas. Fischer et al. (2012) developed the GAEZ dataset (Global Agro-ecological Zones) that contains the potential harvested area and yield for 23 crops circa 2000 considering the crop-specific agro-climatic and edaphic suitability criteria. You and Wood (2006)

developed the Spatial Production Allocation Model (SPAM) firstly at the continental scale then subsequently at the global scale by using an entropy-based model to down scale crop production. The first global SPAM dataset is available for the year 2000, at the time when M3, MIRCA and GAEZ were also available (Figure 1).

*(insert Figure 1 here)*

**Figure 1: Overview of the global spatially explicit datasets on agricultural production.**

Changes in agricultural lands over time is as important as that over space, especially given that the changes in cropping pattern and crop yields are more frequent than that at the land cover level (Verburg et al., 2011). While there are four spatially explicit datasets on global agricultural production available around the year 2000 (Anderson et al., 2015), three of them, i.e. M3, MIRCA, GAEZ, are no longer available after 2000. Agricultural production systems are constantly changing, and these changes are not trivial. However, a lot of recent agricultural and environmental assessments were still based on those maps produced decades ago (Deutsch et al., 2018;Nanni et al., 2019;Estes et al., 2018;Prestele et al., 2018;Erb et al., 2018;Porwollik et al., 2019;Yu et al., 2017b), suggesting that an update of existing global agricultural production maps is very desirable for subsequent analysis.

SPAM had committed to update maps in every five years (You et al., 2014;Wood-Sichra et al., 2016), which substantially fills the data gap and extends the work for global agricultural production mapping by operating a global gridscape at the confluence between earth and farming systems in multiple time stages. The SPAM model has become a critical tool to many initiatives within and beyond the Consultative Group for International Agricultural Research (CGIAR). Moreover, SPAM data are frequently downloaded and widely used by researchers and analysts from international originations, academia, governments agencies all over the world. The global spatially explicit datasets in multiple time stages enable scientists as well as policymakers to better address the global change challenges within the anthroposphere and beyond, such as targeting agricultural and rural development policies and investments, increasing food security and growth with minimal environmental impacts. Successful examples include AGRODEP Library (http://www.agrodep.org/fr/node/1794), GEOGLAM (www.geoglam.org), USAID Feed the Future Innovation Lab for Small-scale Irrigation (https://ilssi.tamu.edu/), Africa Infrastructure (https://openknowledge.worldbank.org/handle/10986/2692), and so on. In this paper, we introduce SPAM2010, the latest update of the SPAM family. The next section gives an overview of the SPAM model. Section 3 provides a detailed description and improvements of SPAM2010. Section 4 introduces the data preparation, and Section 5 presents some of the results produced by SPAM2010. Finally, we conclude with some advice on using the maps and our own plan for the future of SPAM.

## 2 SPAM overview

The main purpose of SPAM is to disaggregate crop statistics (e.g., harvested area, production quantity and yield) by different farming systems, and to further allocate such disaggregated statistics into spatially gridded units (Figure 2). In SPAM, disaggregation is processed before allocation, because crop yields are likely to be substantially different between different farming systems (e.g., irrigation versus rainfed) even at the same location. The whole procedure entails a data fusion approach that combines information from different sources and at different spatial scales by deploying various matching and

calibration processes. Then all the data elements are processed by the optimization model which generates results at the grid level (Figure 2).

*(insert Figure 2 here)*
**Figure 2: The overall structure of the SPAM model.**

The SPAM methodology was first developed in a trial project for six major crops in Latin America and the Caribbean by combining satellite imagery and crop statistics. Later on, it was used to derive regional estimates of spatially disaggregated

crop production in Brazil and sub-Saharan Africa (You and Wood, 2006;You et al., 2009). Over the years the model has evolved, adding more crops, using additional data and increasingly complicated optimization equations, as well as expanding to global coverage (You et al., 2014). The SPAM methodology is different from its counterparts. For example, M3 has no distinction across farming systems and is allocated proportionally (within each crop) to each grid cell within each sub-national unit, hence the M3 dataset provides interpolated estimates of output by crop at the resolution of the satellite data

(Figure 1). SPAM not only considers the crop yield variation across farming systems but also assigns production weighted by price to grid cells rather than pure proportionality (Donaldson and Storeygard, 2016). Moreover, MIRCA and GAEZ focus more on the biophysical aspects of agricultural production, while SPAM uses a triangulation of any and all relevant background and partial information, which not only include national or sub-national crop production statistics, satellite data on land cover, but also include maps of irrigated areas, crop potential suitability, secondary data on population density,

market accessibility, cropping intensity and crop prices (Figure 2).

The SPAM model produces global gridded maps of agricultural production at a 5 arc-minute spatial resolution. The first SPAM maps, known as SPAM2000, represent global agricultural production circa year 2000 for 20 crops, with the exception of a few small island states and conflict zones (You et al., 2014). Subsequently, the SPAM maps have been updated every 5

years. SPAM2005 acts as an intermediate update which expands the coverage of crops from SPAM2000. The 42 crop categories are further adopted in SPAM2010 (Figure 1).

**3 Model improvement for SPAM2010**

There are three sub-modules in a standardized SPAM model: disaggregation, optimization and allocation. We conceptualize

the SPAM2010 based on this general setting while adds further methodological and data improvements, which mainly include the update of base year, the expansion of sub-national administrative unit coverage, the extension of crop list, and the substitution of the latest hybrid cropland map as the basic allocation layer. Considering the huge amount of input data and

multiple year efforts, such an update is not trivial and will be critical for the user community. In this section, we briefly introduce the model structure and how these sub-modules are processed and connected.


## 3.1 Disaggregation

The first step for SPAM is to disaggregate crop statistics of agricultural production (e.g. the yield, harvested area, and total production) by administrative unit levels ($k$), crop type ($j$), and farming system ($l$) from coarser scale to finer scale (illustrated by orange shapes in Figure 2). For example, the national-level statistics are disaggregated into sub-national
levels, statistics for crop aggregates are divided into individual crop types, and the crop statistics are further separated by rainfed and irrigated conditions. Disaggregation is a non-spatial module. For the administrative unit (ADM), we consider three levels: $k = 0$ (national level), 1 (sub-national level 1), or 2 (sub-national level 2), and refer to the country-specific administrative level as the statistical reporting units (SRUs, SRU = $k_0$, $k_1$ or $k_2$). In general, The SPAM model will have better performance if crop statistics are more disaggregated by ADM. Therefore, we prefer to collect crop statistics for
ADM1 and ADM2, despite statistics are mostly available at the ADM0 level, and the sub-national coverage always being less complete. Comparing to the previous SPAM products, the sub-national coverage percentage has increased markedly for SPAM2010, which is described in detail in Section 4.1.

We improve the model capacity in SPAM2010 as well: we simultaneously allocate 42 crops and crop aggregates (versus the
20 crops and crop aggregates in SPAM2000) and consider four farming systems for each crop (Figure 2). In SPAM2010, we keep the farming systems conceptualized for SPAM2000, which have been approved useful to represent the different crop performances under different management systems, e.g. the irrigated yields of a particular crop are likely to be substantially different from the corresponding rainfed yields. The four farming systems are defined as:

- The irrigated farming system (I) refers to the crop area equipped with either full or partial control irrigation. Normally
the crop production on the irrigated fields uses a high level of inputs such as modern varieties and fertilizer as well as advanced management such as soil/water conservation measures.
- The rainfed high input farming system (H) refers to the market-oriented crop area, which uses high-yield varieties, machinery with low labor intensity, and optimum applications of nutrients and chemical pest, disease and weed control.
- The rainfed low input farming system (L) refers to crop area which uses traditional varieties and mainly manual labor
without (or with little) application of nutrients or chemicals for pest and disease control. Production is mostly for own consumption.
- The rainfed subsistence farming system (S) is introduced to account for situations where cropland and suitable areas do not exist, but farmland is still present in some way. Production is mostly for own consumption, which is also low input as well.


The four conceptualized farming systems are mainly delineated by the water supply system and inputs used by farmers, despite global data on farming system shares for each crop being largely absent. For a small number of large countries, e.g. Brazil, China, India, Russia, the United States (see more details in Section 4.1), we have data on farming system shares at the ADM1 level. For the other countries we first assign the national farming system shares to each ADM1 level, and then adjust individual ADM1 farming system shares in light of the supporting evidence. For example, if the national share for irrigation of wheat was 30%, we assign that to all ADM1 units. Then we look at individual units, and if supporting evidence (e.g., the Global Map of Irrigation Areas (GMIA) data) indicates that there was no irrigated area present in a particular AMD1 unit, we set the irrigation share of wheat to zero in that administrative unit. Finally the farming system shares at national level are recalculated as the weighted average of the adjusted ADM1 estimates. For a few countries which have very limited data accessibility, experts may give their opinions. For example, it was often necessary to use farming system shares from one crop as proxies for similar crops (e.g., farming system shares for beans are used for all pulses) or shares from one country and apply them to similar countries (e.g., the geographically smaller countries in the Middle East, including Kuwait, Oman and Qatar, are assigned the same farming system shares).

For irrigated farming systems, the crop-specific shares are derived by dividing the harvested area cultivated under full control irrigation obtained from AQUASTAT, MIRCA, and country-level statistics by the overall harvested area. For rainfed farming systems, crop-specific shares are primarily estimated based on generalized assumptions for individual countries and crops. For example, all cereals in Western Europe are produced with high inputs, whereas 20% cereals in Sub-Sahara Africa are grown under a subsistence farming system. We also assume fertilization as a proxy for high-input use, so if irrigated crop areas and overall fertilized and non-fertilized areas of a crop are known, it is possible to deduce rainfed–high shares by subtracting the irrigated areas from fertilized areas. The remainder of fertilized area will be then classified as rainfed–high and the non-fertilized areas will be further split between rainfed–low and rainfed–subsistence. In addition, the shares of rainfed–subsistence are assigned when there has not enough suitable area for rainfed–low conditions to satisfy the completeness of disaggregated crop statistics in terms of area extent and/or production quantity. In such cases a portion of the rainfed–low statistics were assumed to stem from rainfed–subsistence. Although disaggregation is a non-spatial module of SPAM, it is applied interactively with the spatial modules by the support of multiple spatial data and non-spatial data, which are elaborated in detail in the Section of "data preparation".

### 3.2 Optimization

The core part of the SPAM model is the cross-entropy module (illustrated by the green dash frame in Figure 2), which is used to achieve the allocation for each spatial grid ($i$). It works by (iteratively) minimizing the error between the pre-allocated shares of physical area ($\pi_{ijl}$) and the allocated shares of physical area ($s_{ijl}$) in each pixel $i$ by crop $j$ and production system $l$:

$$\min_{\{s_{ijl}\}} CE(s_{ijl}, \pi_{ijl}) = \sum_i \sum_j \sum_l s_{ijl} \ln s_{ijl} - \sum_i \sum_j \sum_l s_{ijl} \ln \pi_{ijl} \quad (1)$$

Where: *CE* is the abbreviation for cross entropy, which is defined as the log function of probability. The difference between {s ln s} versus {s ln π} means the estimated probability *s* and its prior probability *π* are minimized subject to certain constrains:

(i) Constraint specifying the range of allocated physical area shares:
$$0 \le s_{ijl} \le 1, \quad \forall i \forall j \forall l \quad (2)$$

(ii) Constraint specifying the sum of allocated physical area shares within a grid:
$$\sum_i s_{ijl} = 1, \quad \forall j \forall l \quad (3)$$

(iii) Constraint specifying that the sum of allocated physical area over all crops and farming systems within a grid should not exceed the actual cropland within the same grid:
$$\sum_j \sum_l AdjCropA_{jlk_{SRU}} \times s_{ijl} \le AdjCropLand_i, \ \forall i \in k_{SRU} \quad (4)$$

(iv) Constraint specifying that the allocated physical area by grid, crop and farming system should not exceed the suitable area within the grid with corresponding crop and farming system:
$$AdjCropA_{jlk_{SRU}} \times s_{ijl} \le AdjSuitArea_{ijl}, \ \forall i \in k_{SRU} \quad (5)$$

(v) Constraint specifying that the sum of allocated physical area over all farming systems within a sub-national unit should be equal to the sum of statistical physical area overall all farming systems within the corresponding sub-national unit:
$$\sum_{i \in k_{SRU}} \sum_l AdjCropA_{jlk_{SRU}} \times s_{ijl} = \sum_l AdjCropA_{jlk_{SRU}}, \ \forall j \in P \quad (6)$$

where: *P* is the set of commodities for which sub-national statistics exist.

(vi) Constraint specifying that the sum of allocated physical area under an irrigated farming system within the grid should not exceed the area equipped for irrigation in the grid:
$$\sum_{j \in Q} AdjCropA_{jlk_{SRU}} \times s_{ijl} \le AdjIrrArea_i, \ \forall i \in k_{SRU} \quad (7)$$

where: *Q* is the set of commodities which are fully or partly irrigated within grid *i*.

Shares $s_{ijl}$ are the probability values between 0 and 1:
$$s_{ijl} = \frac{AllocA_{ijl}}{AdjCropA_{jl}} \quad (8)$$

where: $AdjCropA_{jl}$ is the total physical area of a given SRU for crop *j* at input level *l* to be allocated. $AllocA_{ijl}$ is the area allocated to grid *i* for crop *j* at input level *l*.

$\pi_{ijl}$ indicates the decision to produce a particular crop under a specific production system, which is normally dependent on both biological and economic factors. However, subsistence farmers mainly grow crops for their own consumption, largely uncoupled from price, market access or crop potential suitability conditions. Therefore, we first assume the prior allocation

for subsistence physical area ($\overline{CropA}_{ijS}$) in grid $i$ by crop $j$ under this circumstance is simply dependent on rural population density:

$$\overline{CropA}_{ijS} = AdjCropA_{jkS} \times \frac{AggRurPop_i}{\sum_{i \in k} AggRurPop_i} \qquad \forall i \forall j \quad (9)$$

where: $AdjCropA_{jkS}$ is the generated physical area for crop $j$ at subsistence farming system for the given SRU $k$, and $AggRurPop_i$ is the rural population density at grid $i$ (see the detailed description in the following Section 4).

Then for the three remaining farming systems, we assume the potential unit revenue of planting a certain crop ($Rev_{ijl}$) would affect farmers' crop choices:

$$Rev_{ijl} = \frac{AdjCropA_{jlk_{SRU}}}{\sum_i AdjCropland_i} \times Price_j \times Access_i \times PotYield_{ijl} \quad (10)$$

where: $AdjCropland_i$, $Price_j$, $Access_{ij}$ and $PotYield_{ijl}$ are the adjusted cropland area, market price, accessibility parameter and potential yield values for crop $j$ in farming system $l$ and grid $i$ (see the detailed description in the following Section 4).

Then we assume the priors for the remaining three farming systems are mainly influenced by the estimated revenue, cropland area and irrigated area.

For an irrigated farming system (I):

$$\overline{CropA}_{ijI} = AdjIrrArea_i \times \frac{Rev_{ijI}}{\sum_j Rev_{ijI}} \qquad \forall i \forall j \quad (11)$$

For a rainfed–high (H) and/or a rainfed–low (L) farming systems:

$$\overline{CropA}_{ijl} = \left( AdjCropLand_i - AdjIrrArea_i - \overline{CropA}_{ijS} \right) \times \frac{Rev_{ijl}}{\sum_j Rev_{ijl}}, \; l = H, L \quad \forall i \forall j \quad (12)$$

where: $AdjCropLand_i$ and $AdjIrrArea_i$ are the cropland area and irrigated area at grid $i$ (see the detailed description in the following Section 4).

Finally, the main inputs for the optimization procedure are converted to shares and written as:

$$\pi_{ijl} = \frac{\overline{CropA}_{ijl}}{\sum_{i \in k_{SRU}} \overline{CropA}_{ijl}} \quad (13)$$

The optimization module in SPAM2010 is almost the same as that in previous versions. We apply the cross-entropy process in the General Algebraic Modeling System (GAMS), which ensures the optimization procedure iterates until a solution is found. Once the allocation is successful, meaning that an optimal or locally optimal solution has been found, the routine immediately returns the allocated physical area ($AllocA_{ijl}$) by grid $i$, crop $j$ and farming system $l$, and the program continues with post processing automatically (Figure 2). If the solution is infeasible or non-optimal, the program stops, allowing for manual scrutiny, adjustment and re-run (see data harmonization in the following section).

### 3.3 Allocation

Using the results of the optimization, the allocation module produce maps of harvested area ($AllocH_{ijl}$), yield ($AllocY_{ijl}$), and production quantity ($AllocP_{ijl}$) for each grid $i$ by crop $j$ and farming system $l$ (Figure 2). For harvested area, we convert the allocated physical area ($AllocA_{ijl}$) to allocated harvested area ($AllocH_{ijl}$) by multipling by cropping intensity ($CropIntensity_{jlk}$):

$$AllocH_{ijl} = AllocA_{ijl} \times CropIntensity_{jlk} \quad (14)$$

For yield, we first calculate an average potential yield ($\overline{PotYield}_{jlk_{SRU}}$) within an SRU using the allocated harvested area as weight, then the allocated yield ($AllocY_{ijl}$) is estimated as:

$$AllocY_{ijl} = \frac{AdjPotYield_{ijl} \times AdjCropY_{jlk_{SRU}}}{\overline{PotYield}_{jlk_{SRU}}} \quad (15)$$

where the average potential yield is calculated as:

$$\overline{PotYield}_{jlk_{SRU}} = \frac{\sum_{i \in SRU}(AdjPotYield_{ijl} \times AllocH_{ijl})}{\sum_{i \in k_{SRU}} AllocH_{ijl}} \quad (16)$$

We finally estimate the production quantity ($AllocP_{ijl}$) as:

$$AllocP_{ijl} = AllocH_{ijl} \times AllocY_{ijl} \quad (17)$$

### 4 Data preparation for SPAM2010

The largest amount of effort to create a SPAM map is spent on identifying, collecting and harmonizing data. For the production of SPAM2010, we collect raw data from two major sources: we first collect non-spatial crop statistics for the data disaggregation process; we then collect and/or create multiple spatially explicit constraint maps at a 5 arc-minute resolution from both biophysical and socioeconomic aspects for the spatial optimization and allocation processes. Afterwards, we introduce how these multi-sourced data are harmonized and how data adjustment is taking place.

### 4.1 Crop statistics

#### 4.1.1 Crop statistics disaggregated by administrative units

We start with the administrative units ($k$) for which we have been able to obtain crop production statistics (Figure 2). We primarily used FAO's Global Administrative Unit Layers (GAUL) at both the national and sub-national levels to relate the tabulated crop statistics to gridded data during the allocation process. GAUL contains shapefiles for three administrative level units: ADM0 (national level 0), ADM1 (sub-national level 1) and ADM2 (sub-national level 2). Shape files from the

285 Database of Global Administrative Areas (GADM) are used for ADM1 and ADM2 in China, since they proved to be easier to match to the statistics.

We collect crop statistics from FAOSTAT, EUROSTAT, CountrySTAT, ReSAKSS, national statistical offices, ministries of agriculture or planning bureaus of individual countries, household surveys and a variety of ad hoc reports related to a

290 particular crop within a particular country (Figure 2). SPAM estimates are most dependent on the degree of disaggregation of the underlying national and sub-national production statistics, so it is important to identify and collect as many subnational statistics as possible (Joglekar et al., 2019). Although we prefer to collect crop statistics for ADM1 and ADM2 and run the model at the ADM1 level for all countries, unfortunately, crop statistics are mostly available at the ADM0 level, the sub-national coverage being less complete. Therefore, for most countries we run SPAM at an ADM0 level, except for

some (geographically) large countries that are modeled at an ADM1 level. We summarize the sub-national data coverage by region in Table 1. We present the detailed procedure for collecting crop statistics in the Supplementary Information (SI, Section S1), which further contains a table listing all countries that are modeled at an ADM1 level (Table S1) and a table listing the sources of crop statistics by country and sub-national coverage (Table S2) for all countries.

**Table 1: Sub-national coverage of crop production statistics by region.**

*(insert Table 1 here)*

We collect data in all the ADM1 units in the United States, Russia and Canada, and at least 80% of the ADM1 units for the rest regions worldwide. While Europe, Middle East, Oceania, Russia and Sub-Saharan Africa have data collected on the full

set of crops in below 80% of their ADM2 units. This coverage is substantially improved for SPAM2010 than that for SPAM2005, which are only 66.2% and 43.2% for ADM1 and ADM2 respectively (Table 1).

Monfreda et al. (2008) reported that 81% of the year 2000 global harvested area data in their M3 came from sub-national sources, but it does not distinguish coverage by sub-national levels 1 and 2. SPAM often has higher levels of sub-national

coverage than M3, especially in Africa and the former states of the Soviet Union. This can be seen in SPAM2005, e.g., 93.4% of global data came from ADM1 sources and 54.6% from ADM2 sources (Wood-Sichra et al., 2016). While in SPAM2010, such coverage rates are further increased to 96.1% and 68.0% respectively (Table 1).

### 4.1.2 Crop statistics disaggregated by crop types

We simultaneously allocate 42 crops and crop aggregates (*j*) for SPAM2010 (Figure 2). The crop categories are driven by FAO's Statistical Database (FAOSTAT)'s definitions. Comprised of 33 individual crops (e.g., wheat, rice, maize, barley, potato, bean, cotton) and 9 crop aggregates (e.g., other cereal, vegetables), the SPAM2010 crop list covers all crops reported

by FAO, except for explicit fodder crops (mostly grasses) which are not modeled. When multiple FAO crops fall into a single SPAM2010 crop category (e.g., vegetables), FAO's corresponding area and production data was summed up and yields were calculated as a weighted average. We present the detailed procedure for aligning the crop types in the SI (Section S2), which further contains a full list of crops and their respective FAO code (Table S3).

We collect statistics on harvested area ($H$), production ($P$) and yield ($Y$) ($CropHPY$) by each crop $j$ in each administrative unit $k$ for data disaggregation (Figure 2). We prepare data for the model based on the 2009-2011 average of the crop production statistics ($AvgCropHPY_{jk}$). If data is missing from this time period, we use the average from the available data spanning the closest years between 2005 and 2015. We make corrections for discrepancies in statistical reporting units, crop names, and units of measurement during the initial cleaning phase of the data. For example, we adjust all national and sub-national statistics ($AdjCropHPY_{jk}$) using the national 2009-2011 average from FAO, in order to improve the comparability of the crop production statistics across countries, we explicitly distinguish between crops not grown in an area (coded as zero) and crop data that is not available for an area (coded as a missing value). Despite the possible uncertainties in FAO data, it has been chosen as the baseline in the adjustment of country statistics mainly because: (1) FAO data is the most widely acknowledged global agricultural statistics, hence it is the most appropriate source for the purpose. (2) SPAM products have been used by many global models such as IFPRI's IMPACT (https://www.ifpri.org/project/ifpri-impact-model), IIASA's GLOBIOM (https://iiasa.ac.at/web/home/research/GLOBIOM/GLOBIOM.html). These models use FAO country data for cross-country comparisons and they need our maps to be consistent with FAO data. In fact, the idea of conceptualizing SPAM is to spatially allocate statistics from administrative units to spatial grids, and the maps could be easily adjusted to any other country data. We present the detailed procedure for adjusting the crop statistics in the SI (Section S3).

### 4.1.3 Crop statistics disaggregated by farming systems

We elaborated the disaggregation module for obtaining the farming system shares by crop $j$ and administrative unit $k$ ($Percent_{jlk}$) in Section 3.1. In some countries there are statistics, in others experts may give their opinions, or assumptions are made as to how some crops are grown in a similar way as other crops. In supplementing to Section 3.1, we present more details on the procedure for obtaining the farming system shares in the SI (Section S4), which further contains a table listing the sources of sub-national farming systems data (Table S4) and a table listing the farming system shares by crop groups and selected countries (Table S5). For example, shares of *irrigated farming system* were taken directly for country statistics like Brazil, China and the United States, at ADM1. For some countries these figures were found in MIRCA and yet for the rest of the countries AQUASTAT provides information on irrigated areas per crop at the national level. We are able to source data on farming system shares at the ADM1 level for limited large countries (Table S4). Based on this list we showcase the shares of production under irrigated and rainfed systems for selected crop groups and countries (Table S5). We choose Brazil, China, Ethiopia, France, India, Indonesia, Nigeria, Turkey and the United States, because they vary in agro-ecology, region,

income level and geographical size. For cereal crops, the three Asian countries (China, India and Indonesia) have the highest shares of irrigated area, whereas the two Sub-Saharan countries (Ethiopia and Nigeria) have the lowest shares of irrigated area. For roots, tubers and pulses production, the United States and both European countries have the highest shares of irrigated areas, while the Sub-Saharan countries again have less than one percent each. Aggregating across all crops, the three Asian countries rank highest in terms of irrigated area shares while the two Sub-Saharan countries rank lowest.

We disaggregate the adjusted statistics on harvested area and yield ($AdjCropHPY_{jk}$) for each of the four farming systems (Figure 2). Harvested area by farming system $l$ ($AdjCropH_{jlk}$) is directly calculated by multiplying the farming system shares ($Percent_{jlk}$), while the yields by farming system $l$ ($AdjCropY_{jlk}$) are more complicated to calculate. Here we not only consider the farming system shares but also the yield conversion factors (determined by expert judgement) to distinguish the yield variations for irrigated versus rainfed systems and rainfed–high versus rainfed–low systems. We present the detailed procedure for disaggregating the crop statistics by farming systems in the SI (Section S5), which further contains a list of the yield conversion factors, i.e. both the factor of crop yield under irrigated versus crop yield under rainfed (with a "I") and that of yield under rainfed high input versus yield under rainfed low input (with a "R"), for selected crops and countries (Table S6).

### 4.1.4 Physical area

We create a new variable—physical area ($AdjCropA$, i.e., the area footprint of the crop irrespective of the number of times per year the same area was planted and harvested)—for the model, recognizing that crop production may take place over several seasons within a year. SPAM does not have a direct mechanism for modeling sequential or intercropping processes, and thus we use harvested area and cropping intensity ($CropIntensity$) per crop as a proxy for these processes:

$$AdjCropA_{jlk} = \frac{AdjCropH_{jlk}}{CropIntensity_{jlk}} \quad (18)$$

Where $AdjCropA_{jlk}$ indicates the generated physical area by crop $j$, farming system $l$ and administrative unit $k$.

Implementing the crop allocation calculations by farming system enables more flexibility when accounting for variation in these cropping intensity practices. However, such data is still scarce. Only some country statistics have such figures, e.g., Bangladesh and India, thus we rely primarily on expert judgment to seek information on the number of cropping seasons by crop, farming system and country. We present the detailed procedure for generating physical area in the SI (Section S6), which further contains a table listing $CropIntensity_{jlk}$ by crop groups and selected countries (Table S7).

## 4.2 Spatial constraints

### 4.2.1 Cropland extent

We apply an already classified land-cover image—where cropland has been identified (*CropLand*)—to determine the places where production statistics can be allocated. Comparing to SPAM2000 and SPAM 2005, SPAM2010 not only updates the statistics but also the cropland distribution: it uses the global cropland synergy map with spatial resolution of 500 m circa 2010, jointly produced by CAAS and IFPRI (Figure 1). The CAAS-IFPRI cropland dataset fuses national and sub-national statistics with multiple existing global land cover maps including GlobeLand30, CCI-LC, GlobCover 2009, MODIS C5 and Unified Cropland. It reports three major parameters by grid around the year 2010: the median and maximum cropland percentage (*MedCropLand$_i$* and *MaxCropLand$_i$*) and a confidence score between 0 to 1 in the cropland estimation (*ProbCropLand$_i$*). Although the synergy dataset does not delineate the geography of specific crops, it designates the total cropland extent with a higher accuracy than the input datasets and tries to be consistent with administrative cropland statistics. The detailed description of the CAAS-IFPRI cropland dataset is submitted as a parallel paper, see Lu et al. (2020). Before using the cropland extent in SPAM2010, we aggregate the cropland synergy map from 500 m grid cells to 5 arc-minute grid cells for the three major parameters. We present the cropland data preparation in the SI (Section S7), which further contains the resampled maps on median cropland (*AggMedCropLand$_i$*), maximum cropland (*AggMaxCropLand$_i$*) and cropland confidence (*AggProbCropLand$_i$*) (Figure S1).

### 4.2.2 Crop potential suitability

We estimate the crop suitable area (*SuitArea*) from GAEZv3.0 to consider the spatially varied, potential suitability for different crops in terms of different thermal, moisture and soil requirements as an allocating parameter. GAEZv3.0 produces a 5 arc-minute gridded suitability index for 49 major crops, four input levels (i.e., high, intermediate, low or mixed) and two main water regimes (i.e., irrigated or rainfed). The major crops surveyed by GAEZ include most of the SPAM2010 crops—those not included are assigned values from similar GAEZ crops. We present the detailed procedure for estimating the suitable area (*SuitArea$_{ijl}$*) for grid *i*, crop *j* and input *l* in the SI (Section S8), which further contains a table illustrating the concordance between GAEZ crops and SPAM2010 crops (Table S8), and maps of suitable areas for maize irrigated, rainfed—high and rainfed low farming systems (Figure S2).

### 4.2.3 Irrigated area

We adopt the irrigated area (*IrrArea*) from the Global Map of Irrigation Areas (GMIA) to consider the share of irrigated area within a grid as an allocation parameter. GMIAv5.0 is the only irrigated area dataset with global coverage, which estimates the amount of area equipped for irrigation at a 5 arc-minute resolution for the period around 2005 (Siebert et al., 2013).

GIMAv5.0 does not include information on the functionality or quality of irrigation equipment and makes no distinctions between different types of irrigation which may introduce errors and inconsistencies into the allocation. We present a map of area equipped for irrigation at the grid level ($IrrArea_i$) in Figure S3 in the SI (Section S9).

### 4.2.4 Protected area

We select the protected area ($Protect$) from the World Database on Protected Areas (WDPA), released by the International Union for Conservation of Nature (Deguignet et al., 2014), as an allocation parameter to indicate the locations where crop production is least likely to take place. Notionally, crop production does not occur within protected areas (such as national

parks, wilderness areas and nature reserves), but in reality it does. During the initial allocation process SPAM allows for crop allocation in protected areas to allow for this reality, but if the model does not solve, one option is to increase the area designated as cropland, suitable land or irrigated land. That expansion is not allowed into protected areas. The data is originally in a polygon format. We convert it to 5 arc-minute grids ($Protect_i$) and map it in Figure S4 in the SI (Section S10).

### 4.2.5 Accessibility

We adopt the population count from the Gridded Population of the World (GPWv4.0) as a proxy to consider the influence of market accessibility ($Access$) on farmers' crop choices (Equation 10). GPWv4.0 provides a gridded representation of human populations across the globe at a 30 arc-second resolution (CIESIN, 2016). For SPAM 2010, we aggregate the population count grid to a 5 arc-minute resolution and re-calculated the population density. Then we derive rural population density

($AggRurPop_i$) based on the assumption that if there is cropland within the 5 arc-minute grids, then the population residing within the grids should be rural people. We do not aim to distinguish rural area from urban area. Instead, the variable $AggRurPop_i$ is introduced to estimate the market accessibility and to account for subsistence production. Therefore, it does not mean the accessibility of getting food. As crop-specific revenue is divided by the total revenue within a pixel (Equation 11 and 12), the prior is not affected by market accessibility if it is not crop-specific. In other words, crop-specific market

accessibility is preferable for the current SPAM model. Such accessibility doesn't exist now. We create a measure of market accessibiliity ($Access_i$) from the grid level estimates of rural population by considering the relationship between $AggRurPop_i$ and maximum and minimum rural population densities within a country. Population in grids with no cropland are not used in further calculation. We present the detailed procedure for measuring $Access_i$ in the SI (Section S11), which further contains a map of $AggRurPop_i$ (Figure S5) and a table of minimum and maximum rural population densities in select countries (Table

S9).

### 4.2.6 Crop revenue

We measure the crop potential revenue (*Rev*)—determined by market accessibility (*Access*), crop prices (*Price*) and crop potential yield (*PotYield*)—as an allocation parameter, which fully considers the influence of farmers' crop choices. We adopt the crop-specific prices (*Price$_j$*) from FAO's Gross Production Value. Prices for crop aggregates (e.g., tropical fruit) are calculated as a weighted average from FAO world totals. It is important to note that these are not spatially-specific prices, and they likely misrepresent the local economic realities and associated cropping choices faced by farmers. We list the crop prices in Table S10 in the SI (Section S12). We estimate the crop-specific potential yield (*PotYield$_{ijl}$*) as a composite measure of potential harvested yield (*PotHarvYield$_{ijl}$*) based on GAEZ. We present the detailed procedure for estimating *PotYield$_{ijl}$* in the SI (Section S12), which further contains a table listing the dry matter yield conversion factors (Table S11). Finally, we calculate the grid-level potential unit revenue of planting a certain crop according to Formula 1.

### 4.3 Data harmonization and adjustment

### 4.3.1 Adjusting input data

We list the main input variables for SPAM2010 in Table 2. As we collect data from various sources, it might inevitably cause information inconsistancies. Therefore, we set rules to harmonize all these data. At the beginning, we adjust all the area-related parameters (e.g., cropland area, irrigated area and suitable area) to satisfy the constraints at the administrative unit level before calculating the priors of physical area. When the model runs, it might be unable to find the optimal allocation solution for a particular country, administrative unit or crop. Under these circumstances, we set several options to "force" a solution, including adjusting the entropy conditions, and adjusting the data harmonization rules. We elaborate on the details for adjusting areas (Section S13), entropy conditions (Section S14) and harmonization rules (Section S15) respectively in the SI.

**Table 2: The main input variables used in SPAM2010.**

*(insert Table 2 here)*

### 4.3.2 Adjusting allocation results

The model produces the allocated harvested area (*AllocH$_{ijl}$*), the allocated yield (*AllocY$_{ijl}$*), and the production quantity (*AllocP$_{ijl}$*) for SPAM2010. As a final step, we need to adjust the allocation results in order to keep the grid level results consistent with the statistics. In each step of estimation, we scale the results to the national 2009-2011 FAO average (*AvgCropHPY$_{jk0}$*) by crop *j* and country $k_0$ to even-out potential inaccuracies introduced by the allocation adjustments. This means all the allocated results in this subsection could be adjusted (if necessary) before being applied in the next phase.

We first scale the allocated harvested area ($AllocH_{ijl}$) to the national FAO average to even-out potential inaccuracies introduced by the allocation adjustments:

$$AdjAllocH_{ijl} = \frac{AllocH_{ijl}}{\sum_{i \in k_0} \sum_l AllocH_{ijl}} \times AvgFAOCropH_{jk_0} \quad (19)$$

Total harvested area of each crop in the grid was calculated by summing estimates across the four farming systems:

$$AdjAllocH_{ij} = \sum_l AdjAllocH_{ijl}, \; \forall l \quad (20)$$

For yield, we begin with the potential harvested yields ($PotHarvYield_{ijl}$) developed earlier (see Section S12 in the SI). Missing values were filled in sequentially using the following values in order of availability:

    i.    Potential yield from potential suitability surfaces:

$$AdjPotYield_{ijl} = PotHarvYield_{ijl} \quad (21)$$

    ii.   Average potential yield in SRU:

$$AdjPotYield_{ijl} = \frac{\sum_{i \in k_{SRU}}(PotHarvYield_{ijl} \times AdjSuitArea_{ijl})}{\sum_{i \in k_{SRU}} AdjSuitArea_{ijl}} \quad (22)$$

    iii.  Sub-national yield by crop $j$, input $l$ and ADM2 unit $k_2$:

$$AdjPotYield_{ijl} = AdjCropY_{jlk_2} \quad (23)$$

    iv.  Sub-national yield by crop $j$, input $l$ and ADM1 unit $k_1$:

$$AdjPotYield_{ijl} = AdjCropY_{jlk_1} \quad (24)$$

    v.   National yield by crop $j$, input $l$ and ADM0 unit $k_0$:

$$AdjPotYield_{ijl} = AdjCropY_{jlk_0} \quad (25)$$

Then we modify the allocated yield ($AllocY_{ijl}$) according to the minimum and maximum yields in the administrative unit:

$$\begin{aligned} ModAllocY_{ijl} &= MinYield_{jlk_{SRU}} &&\text{if } AllocY_{ijl} < MinYield_{jlk_{SRU}} \\ ModAllocY_{ijl} &= MaxYield_{jlk_{SRU}} &&\text{if } AllocY_{ijl} > MaxYield_{jlk_{SRU}} \\ ModAllocY_{ijl} &= AllocY_{ijl} &&\text{if } MinYield_{jlk_{SRU}} \leq AllocY_{ijl} \leq MaxYield_{jlk_{SRU}} \end{aligned} \quad (26)$$

For production quantity, we scale the $AllocP_{ijl}$ to the national FAO average:

$$AdjAllocP_{ijl} = \frac{AllocP_{ijl}}{\sum_{i \in k_0} \sum_l AllocP_{ijl}} \times AvgCropP_{jk_0} \quad (27)$$

Then we calculate the total production in the grid by summing overall production levels:

$$AdjAllocP_{ij} = \sum_l AdjAllocP_{ijl}, \; \forall l \quad (28)$$

Finally, we re-calculate the allocated yield from the allocated harvested area and allocated production to effectively scale yields to the national FAO average:

$$AdjAllocY_{ijl} = \frac{AdjAllocP_{ijl}}{AdjAllocH_{ijl}} \quad (29)$$

To simplify, the grid-cell yield is calculated from the reported yield at statistical reporting unit, the allocated area from model results and the potential yield (at grid-cell level) from GAEZ (global Agroecological Zone). These are illustrated in Equations 15 and 16: The spatial variation of yield within a statistical reporting unit follows the same spatial variation of the potential yield of that crop. In other words, the more suitable (higher potential yield) cells would have a relatively higher yield while the average yield of all the grid cells would be equal to the statistically reported yield of the administrative unit.

## 5 Results

In this section, we briefly showcase some of the main SPAM2010 results, which mainly focus on the staple crops, to illustrate how SPAM2010 has been produced.

## 5.1 Disaggreaged crop statistics

Disaggregation of crop statistics is the first step for running the SPAM model. Table 3 summarizes the disaggregated rice harvested area and yield (area-weighted) for global rice production by four farming systems in SPAM2010. At the global level, the world has harvested about 160 million ha. of rice around year 2010. The majority of rice production area is irrigated, i.e., about 98 million ha., which accounts for 61.2% of the total rice harvested area. This share is followed by high input rainfed farming system (17.3%, approximately 27 million ha.), subsistence farming system (16.0%, approximately 26 million ha.) and low input rainfed farming system (5.5%, approximately 9 milling ha.). The global average rice yield is 4,374 kg/ha, which stands at the average yield between irrigated farming system (5,528 kg/ha) and high input rainfed farming system (3,663 kg/ha) and is much higher than the average yield of low input rainfed farming system (1,810 kg/ha) and the average yield of subsistence farming system (1,604 kg/ha). At the regional level, Asia (South Asia, South East Asia, East Asia together) is the largest rice producing region, which has harvested approximately 142 million ha. of rice around year 2010. The majority of Asia rice production area is also irrigated, and the share, i.e., 63.7 %, is close to the global share of irrigated rice farming system. South Asia has more rice area harvested (approximately 60 million ha.) than South East Asia (approximately 49 million ha.) and East Asia (approximately 33 million ha.). However, the average rice yield in South Asia (3,553 kg/ha) is lower than South East Asia (4,125 kg/ha) and East Asia (6,566 kg/ha). Consequently, the total rice production in these regions is very close to each other. Rice production in North America is completely irrigated, and the average yield is relatively high in this region. Subsistence rice production is mainly in Sub-Sahara Africa (SSA) and South Asia and the rice yield under subsistence condition is also the lowest among the four farming systems.

**Table 3: Regional values for area and yield of rice from SPAM2010. Unit: area (1000 ha); yield (kg/ha).**

*(insert Table 3 here)*

**5.2 Allocated harvested area and yield**

After applying the optimization model in GAMS, the disaggregated crop statistics are spatially allocated to produce the SPAM maps. Figure 3 and Figure 4 present the maps of harvested area and yield (after adjustment) for maize, respectively. For all farming systems, as shown in Figure 3(e), maize area is highly concertrated in Northen China and Northen America. However, maize production in North America is mainly rainfed with high input, while in China, rainfed farming system is
mainly located in the North-east part (Figure 3e) and irrigated farming system is mainly found in the Central-north part (Figure 3a). The rainfed low input farming system (Figure 3c) and subsistance farming system (Figure 3d) for maize production are manily located in South America and SSA, while the rainfed high input maize farming system is also widely distributed outside China and Northen America, including Central America, Europe other regions (Figure 3b). As shown in Figure 4(e), the average maize yield is very high in North America and Europe, and is relatively high in South America and
Asia.

*(insert Figure 3 here)*

**Figure 3: Harvested area maps for maize in irrigated (a), rainfed–high (b), rainfed–low (c), subsistence (d) and all (e) farming systems.**


*(insert Figure 4 here)*

**Figure 4: Yield maps for maize in irrigated (a), rainfed–high (b), rainfed–low (c), subsistence (d) and all (e) farming systems.**

**5.3 Value of Production**

Finally, we use the average 2009-2010 base year price (international dollar, I$) to compute value of production in each grid, for each crop and farming system. Table 4 shows value of production for all crops, food and non-food crops in all regions, as well as the percentage of each category value in relation to the total value. Asia (South Asia, South East Asia, East Asia together) accounts for nearly half (49.2%) of the total value of crop production in 2010, while Middle East and North Africa, Central America, Russia and Oceania account for less than 5% each. Globally, food crops accounts for 86.2% of the total
crop production value, with minor regional differences (the classification of crops into food and non-food is detailed in Table S3).

**Table 4: Value of production for all crops, food and non-food crops in various regions.**

*(insert Table 4 here)*


## 6 Data availability

The SPAM2010 provides four essential output indicators, including (a) PHYSICAL AREA: it is measured in a hectare and represents the actual area where a crop is grown, not counting how often production was harvested from it. Physical area is calculated for each production system and crop, and the sum of all physical areas of the four production systems constitute

the total physical area for that crop. The sum of the physical areas of all crops in a pixel may not be larger than the pixel size. (b) HARVESTED AREA: also measured in a hectare, harvested area is at least as large as physical area, but sometimes more, since it also accounts for multiple harvests of a crop on the same plot. Like for physical area, the harvested area is calculated for each production system and the sum of all harvested areas of all production systems in a pixel amount to the total harvested area of the pixel. The sum of all the harvested areas of the crops in a pixel can be larger than the pixel size.

(c) PRODUCTION: for each production system and crop, production is calculated by multiplying area harvested with its corresponding yield. It is measured in metric tons. The total production of a crop includes the production of all production systems of that crop. and (d) YIELD: it is a measure of productivity, the amount of production per harvested area, and is measured in kilogram/hectare. The total yield of a crop, when considering all production systems, is not the sum of the individual yields, but the weighted average of the four yields.


The SPAM2010 can be downloaded from Harvard Dataverse https://doi.org/10.7910/DVN/PRFF8V (IFPRI, 2019), which includes all results of maps, tables and figures. Registered users can find more information for the SPAM model and the previous versions of SPAM datasets, via the dedicated MapSPAM website (http://mapspam.info/). The formal SPAM products in 2000 and 2005 are also available on MapSPAM website. Their Dataverse addresses are:

https://doi.org/10.7910/DVN/A50I2T (SPAM 2000), https://doi.org/10.7910/DVN/DHXBJX (SPAM2005). All these three datasets are in the same place grouped under IFPRI Harvest Choice Dataverse (https://dataverse.harvard.edu/dataverse/harvestchoice).

## 7 Discussion

### 7.1 Model uncertainty and validation

The first SPAM product was the regional level agricultural production maps produced for Brazil circa the year 1994 (You et al. 2006), since then the model and products of SPAM have been continuedly improved and updated. Beside the evolution of method (see Section 3), the evaluation of SPAM model performance is also improving. In one of our early works, the

uncertainty of model, i.e. the variance explained by the cross-entropy approach, is evaluated by comparing it with the performance of simplified proportional approaches, which have been used by Monfreda et al. (2008) for producing the M3 dataset. It proved that the cross-entropy approach was most successful in estimating crop areas than the proportional approaches, no matter in proportion to the total land area, to the cropland area, or to the amount of (biophysically) suitable land for the production of each crop (You et al. 2006). Moreover, many researchers believed that the inclusion of economic factors, i.e. market, would increase the performance of crop disaggregation model, see the discussion in (You et al. 2014), though it does not automatically guarantee that model outputs. This partly explained the considerable discrepancies between SPAM2000 and M3 (Anderson et al. 2015), and partly confirmed that using more sophisticated approaches for production allocation would reduce uncertainty (Donaldson and Storeygard, 2016). In one of our recent works, the sensitivity of the variant of the standard SPAM model output to a few methodological-data choices had been evaluated. These include the spatial allocation method, the crop coverage, the treatment of a "rest-of-crops" aggregate, the incorporation of a "crop potential suitability" data layer, the inclusion of rudimentary economic elements, and the administrative units details of the primary source statistics. It showed that the standard SPAM estimates are unsensitive to the inclusion of crude economic elements, moderate sensitive to the set of crops or crop aggregates being modelled, but mostly dependent on the degree of disaggregation of the underlying national and subnational production statistics (Joglekar et al. 2019). This implies that the improvements on methodological aspect of SPAM have limited effect on reducing uncertainty. By contrast, the quality and accuracy of the underlying statistics used to prime the model is particularly pertinent (Joglekar et al. 2019).

SPAM products are estimates with various uncertainty. Inaccuracy surely exists, and varies from region to region, and even from crop to crop. Although there were efforts paid to the evaluation of model conceptualization and performance, those previous validations should not be taken for granted for the latest updates. Therefore, we carried out extensive validation works to assess the accuracy of the output maps of SPAM2010. Firstly, we relied on a system through which we are able to send the crop maps to collaborators and users alike for comments or assessment. For example, the CGIAR is a global partnership which unites 15 centers engaged in agricultural researches. As each center has its own mandate crops, e.g., IRRI (International Rice Research Institute) for rice and CIMMYT (International Maize and Wheat Improvement Center) for maize and wheat. We took advantage of their vast network of field offices and local expertise to help us to validate the SPAM results. Many researchers from these institutes have been involved in the production of SPAM2010, which increases the reliability of the results. The Chinese Academy of Agricultural Sciences (CAAS) undertook the regional level validation for SPAM2010, following the approaches they have applied for the evaluations of previous SPAM products (Liu et al., 2013;Li et al., 2016;Chen et al., 2016). Moreover, field level validating information have either been collected by crowdsourcing tools such as Geo-Wiki (Fritz et al., 2012) and eFarm (Yu et al., 2017a), or through field trips and workshops onsite or online where local experts were asked to confirm or validate the crop production maps by providing hand-written comments or posting comments online at the our MapSPAM website. Most of these reports were collected crop by crop, and country by country. An example of detailed validation process is provided in the SI (see Section S16). The complete

validation process could take a great deal of effort and time, but these users' feedbacks are quite important and valuable. We took these feedbacks and re-run SPAM model and further released the updated versions of SPAM. The previous SPAM
products have been updated substantially with the help of those comments. For example, SPAM2000 and SPAM2005 are at version 3.07 and version 3.20, respectively. The current product, i.e., SPAM2010v1.10, is released after extensive validations already, it is still open and ready to receive more comments. Such an iterative process would enable a continued update to improve the product quality.

Secondly, we qualitatively evaluated the uncertainty of input data. Like any models, the results depend on the input data and the modelling process. For SPAM, the most important input data is the sub-national crop data, which has large impact on the final product accuracy as mentioned before. We built our SPAM uncertainty rating mainly on the availability and confidence on our sub-national data. In addition, we added the parameters and constraints we have to adjust to solve the SPAM model. For example, we sometimes have to abandon some crop potential suitability constraints in order to solve a country. For some
countries, we may have to allow cropland per pixel to increase by 5 or even 10% than the original input to make the model run. In addition, we collected feedback and comments from users, local experts and collaborators as discussed above. They are sporadic but very useful. We combine all the information together to give a subjective rating on how confidence we, SPAM team, think of our final crop maps (both area and yield) based on the judgment on the reliability of input data. Figure 5 shows the country-level uncertainty rating with 5 categories (1 represents the lowest uncertainty, 5 the highest). The
complete rating list is presented in Section S17 in the SI. Not surprisingly, the uncertainty in Africa and Southeast Asia is higher than those countries in Europe and America. Although such a validation process is not vigorous, but the result is convincing and such a rating is highly demanded and explicitly requested by users.

*(insert Figure 5 here)*
**Figure 5: Subjective uncertainty rating for SPAM2010 by individual countries.**

Thirdly, we quantitatively evaluated the results by cross comparing the results with statistics at another administrative level that have not be used in running the model. We ran SPAM with complete statistics (ADM0, ADM1 and ADM2), and then ran them with only ADM0 and ADM1 statistics, to see how the aggregated results to ADM2 compare to the original
statistics at ADM2, or at least to the aggregated original results at ADM2. The runs were all done at ADM1 and then combined to give results for the whole country. We then calculated the coefficient of determination ($R^2$) between the values allocated from model and obtained from statistics to assess the model performance. In general, a higher $R^2$ indicates for a better performance. This approach has already been used for evaluating the performance of SPAM2000 (You et al. 2014). The upper part of Figure 6 shows the results of such approach applied to Brazil in SPAM2000 for its main food crops, while
the bottom part of Figure 6 shows the results of the same approach applied to the same country for the same crops in SPAM2010. The figure clearly indicates that the model performed better in allocating rice than other crops. Moreover, the

performance improved greatly from SPAM2000 to SPAM2010, especially for soybean and potato. We further selected a few smaller countries in Asia and Africa to undertake the same assessment, which are believed to have a relatively higher uncertainty in terms of input data (Figure 5). Bangladesh, Benin, Senegal, Tanzania were selected as they have good statistical data coverage in SPAM2010. Figure 7 shows that the $R^2$ for selected crops (i.e. maize, rice and cotton) ranged between 0.66 to 0.94, suggesting that the overall performance of SPAM2010 is good in these selected countries for those selected crops.

*(insert Figure 6 here)*

**Figure 6: Comparison between the allocated crop area and statistics crop area at the ADM2 level in Brazil (log-log scale plot, unit: ha.). The upper part is for SPAM2000 and the bottom part is for SPAM2010.**

*(insert Figure 7 here)*

**Figure 7: Comparison between the allocated crop area and statistics crop area at the ADM2 level in Bangladesh, Benin, Senegal and Tanzania for maize, rice and cotton (log-log scale plot, unit: ha.).**

Finally we did regional-level quantitative validations in case that the third-party independent crop maps are available, given that it is impossible for us to collect the true spatial distribution of crops (both area and yield) for the time of 2010 on a global scale. Among the limited third-party, independent spatial crop distribution data, the Cropland Data Layer (CDL, https://nassgeodata.gmu.edu/CropScape/) is a crop-specific land cover dataset created for the continental United State using moderate resolution satellite imagery and extensive agricultural ground truth, which has been applied to validate our SPAM2010 product at the regional scale by correlating the grid level crop area. We focus on the three most popular staple crops in the United States, i.e. maize, wheat and soybean, and obtain the crop area maps of 2009, 2000, and 2011 from CDL. We calculate the 2009-2011 average crop areas at a 5 arc-minute resolution for CDL according to the scheme of SPAM2010, and further calculate the coefficient of determination ($R^2$) and the root mean square error (*RMSE*) between the grid level values derived from the two datasets (Figure 8). The values of $R^2$ are between 0.71 and 0.91 and the values of *RMSE* are between 231 and 307 ha., indicating a relatively high reliability. In particular, the higher $R^2$ and lower *RMSE* suggest our maize and soybean maps are more reliable than the wheat map. There are potentially many factors affecting the different results if we treat CDL as the truth, for example, the different accuracy or availability of input data, suitability layers and parameters for the area shares and yield ratios. Another possible reason is that we did not distinguish spring wheat and winter wheat in SPAM, which partly explains that the agreement for wheat is lower than that for maize and soybean. Moreover, the National Land Cover Dataset of China mapped paddy field distribution as a special cropland cover at a 1×1km grid level (NLCD, http://www.resdc.cn/data.aspx?DATAID=99). By assuming paddy field will be mostly used for growing rice, we evaluate the rice area map in China by correlating SPAM2010_rice and NLCD2010_paddy according to the same scheme described above. The values of $R^2$ is 0.49 and the value of *RMSE* is 1024 ha. (Figure 9). Although this result seems not as good as the results from the United States by using CDL, it is fairly acceptable because NLCD measures land cover

rather than land use and is in a relatively coarse spatial resolution. Moreover, the $R^2$ is substantially increased comparing to its predecessors. For example, the $R^2$ is assessed as 0.42 for SPAM2005 by using the same approach according to Liu et al. (2013). In addition, there are regional-level crop distribution maps produced by independent efforts on interpreting remotely sensed images. For example, Zhang et al. (2017) provided annual paddy area time series from 2000 to 2010 based on satellite remote sensing for China and India. We compared these remote-sensing derived paddy maps with the rice area estimated by SPAM for the year 2010. The $R^2$ values are 0.36 and 0.34 for China and India respectively (Figure 10). We could expand this quantitative evaluation when more third-party independent crop maps are available. However, it should be noted that errors might exit in the third-party independent crop maps as well, hence this quantitative evaluation approach also might result in uncertainty. Our results show that the uncertainty gradually increase when applying CDL, NLCD and Zhang et al. (2017).

*(insert Figure 8 here)*

**Figure 8: Grid-by-grid comparison of crop area for maize (a), wheat (b) and soybean (c) between SPAM2010 and CDL2010 in the continental US.**

*(insert Figure 9 here)*

**Figure 9: Grid-by-grid comparison between SPAM2010 rice area and NLCD2010 paddy field area in China.**

*(insert Figure 10 here)*

**Figure 10: Grid-by-grid comparison between SPAM2010 and Zhang et al. (2017) rice area in China and India.**

## 7.2 Data comparison

There are a few reports which compare SPAM with M3, MIRCA and GAEZ, especially their output maps circa 2000 (Anderson et al., 2015;Donaldson and Storeygard, 2016). Although it is difficult to make statements about which one is better, there are several features that distinguish SPAM products from the M3, MIRCA and GAEZ data. First, the estimates from SPAM can be customized using user provided data for one or more of the inputs variables and return results to the provider in a short turnaround period. Second, although SPAM runs mainly at a 5 arc-minute resolution, it can be run at higher resolutions provided that at least some of the rasterized inputs have also higher resolution data to support such an exercise. Third, considerable effort is made to compile sub-national crop statistics at administrative level two (e.g., district or county) for all possible countries. Fourth, if there is knowledge of crop existence in any area, for any crop, this can be incorporated into the model to make a more accurate crop allocation. Moreover, SPAM does not have a large coverage of crops (compared to M3) and does not include detailed biophysical parameters (compared to MIRCA and GAEZ), instead it focuses more on agricultural production by providing data on crop harvested area and yield disaggregated by farming

systems. Finally, SPAM results are readily available on the internet in several formats (also tabular), for all interested users.

We are currently building a SPAM model on the cloud where we let any user to supply his/her own input data and run SPAM on his/her own under the Github platform. This SPAM on the cloud will be published and communicated to SPAM user community once it is ready.

Anderson et al. (2015) conclude that substantial discrepancies exist across these four global spatially explicit crop production

datasets circa 2000, and the disagreement between models serves as a reminder of the ongoing challenges to the creation of spatially explicit estimates of harvested area and yield based on crop statistics. However, it is more challenging to assess the disaggregated farming system results such as irrigated rice vs rainfed rice, subsistence maize vs high input rainfed rice, which have not been systematically explored in Anderson et al. (2015). We collected additional global datasets which are relevant to agricultural production mapping, e.g. the average irrigated and rainfed yields (ca. year 2000) from Siebert and

Doll (2010), and the harvested area and yield for 4 crops (ca. year 2005) from http://www.earthstat.org/. We compared these datasets with our SPAM products at the corresponding period. We found that the results are differed from crop to crop, and from farming system to farming system. In general, the yield estimates on maize and wheat are better than the other crops, and the irrigated yields are better than the rainfed yields (Figure 11 and Figure 12).

*(insert Figure 11 here)*

**Figure 11: Grid-by-grid comparison between SPAM2000 and Siebert and Doll (2010) in average irrigated and rainfed yields (log-log scale plot, unit: kg/ha.).**

*(insert Figure 12 here)*

**Figure 12: Grid-by-grid comparison between SPAM2005 and EARTHSTAT2005 in crop yields. (log-log scale plot, unit: kg/ha.).**

However, as M3, MIRCA and GAEZ do not provide subsequent global spatially explicit crop production ca. year 2000, it is impossible to compare the current SPAM2010 with other data products. In order to illustrate the continuity of SPAM products, we present a grid-by-grid comparison between SPAM2010 and SPAM2005. Figure 13 shows that rice production

in 2010 increased notably in East Europe, Africa, Northeast China, Northwest India, South Australia, etc., while decreased notably in Central Asia and South America. Maize production displays an overall increase across the globe between 2005 and 2010, except for some places in Central Asia which have shown a decrease trend. It is also noticeable that maize production in the US and Europe have kept relatively stable. This result is accordant with the "maize boom" which had taken place around the globe (Herrmann, 2013), especially in the developing countries (Cairns et al., 2013;Ornetsmüller et al.,

2019). It should be noted that the current type of comparison may not be a perfect comparison (because differences exist in methodologies and data input applied in SPAM2005 and SPAM2010), and that the current comparison only shows the rate of change, thus a higher value does not necessarily indicate a huge change in absolute crop production.

**Figure 13: Comparison between SPAM2010 and SPAM2005: (a) relative difference of rice production; (b) relative difference of maize production.**

In addition, we compared the changes in crop area between SPAM products and the above mentioned regional-level independent crop maps once they are available in time series. We calculated the area changes in maize, wheat and soybean
by overlaying CDL2005 and CDL2010, and undertook the same procedure for SPAM. We then plot these changes (i.e. ΔCDL and ΔSPAM) in Figure 14. Likewise, we compared the changes in SPAM rice area and the changes in paddy rice area obtained from Zhang et al. (2017) (Figure 15). Figure 14 and Figure 15 both show that the coefficient of determination is extremely low between changes yielded from different data products, which further reminds that it is inappropriate to directly compare SPAM products over time, although we are confident with the spatial accuracy of SPAM products at each
time stage. This is mainly because SPAM requires for a large amount of input data, yet the sources of these multiple data inputs can not be guaranteed as the same across different time stages. Therefore, such changes reflected by SPAM products over time not only mix real changes on the ground, but also largely depend on the input data. For example, the cropland layers (one of the most important data inputs) are accessed from different sources to make sure the cropland data and the statistical data are adopted for the same year. We did not evaluate the continuity of these input data, which is almost
impossible and is beyond the purpose of SPAM. Consequently, it is suggested to use the SPAM products with, at least, acknowledgement to the corresponding cropland layer, e.g. Lu et al. (2020) for SPAM2010. Moreover, we do not recommend users to cross compare the SPAM products over time, because the differences may have more input data errors/inaccuracies than detecting the real change on the ground.

**Figure 14: Comparison between SPAM crop area change and CDL crop area change (log-log scale plot, unit: ha.).**

**Figure 15: Comparison between SPAM rice area change and Zhang et al. (2017) paddy rice change (unit: ha.).**


### 7.3 Limitations

As stated previously, the SPAM estimates are dependent on the extent and veracity of the primary input data like most models (Joglekar et al. 2019). SPAM2010 requires data on 42 crops in over 200 countries for the production season. Ideally, this data should be collected at an ADM2 level, however this is not always possible. It is particularly difficult to a few
countries such as Somali and Nigeria where reliable data is not available or different input data just conflict each other For

example, only one crop area (i.e. millet) for a district is already larger than the total cropland area, yet we know there are still five more crops growing in this district. In these cases, we have to adjust the conflicting data, using expert judgment, to make the model solvable. Since most cropping statistics are not delineated by farming system, estimates of the shares of production under each of the four systems in question are required. To convert harvested area statistics to the physical area

statistics used in the model, additional data on cropping intensities by crop and farming systems must be collected. We have made every effort to collect official or published data and we reply on expert judgments as the last resort when we simply could not find other sources. For example, no country publishes official statistics on crop yield ratio (yield conversion factor) between irrigated vs rainfed crop. We surveyed published papers, personal communication with FAO's Agriculture to 2030 team, and gray literature to collect such data. While indeed a series of expert judgments are used, the scope (e.g. crops and

regions) is quite limited in the overall input data. Once the data on disaggregate cropping practices is compiled, several variables at a gridded scale are needed to disaggregate these cropping statistics into the desired spatial units. This data includes estimates of cropland, irrigated land, suitable area and yield, population density and protected areas. The variety and sheer volume required to run the SPAM (and related) models raises questions of reliability and comprehensiveness of estimates across different cropping statistics, geographic areas and countries.


In terms of reliability, different sources of information may lead to inconsistent and even incompatible information. For example, the data on the estimated cropland extent within a grid is compiled from several sources, which in turn deploy different methods to generate their estimates. The extent of cropland within a grid is crucial information for the allocation model, but the confidence regarding its actual location varies regionally, see Lu et al. (2020). Crop statistics on area

harvested and yield may not have been consistently collected and processed across different countries, so these major data may be unreliable to begin with. Additionally, two of the major conversion factors used, farming system shares and cropping intensities, are often not available for each crop and farming system within a country. Lacking raw data on these statistics for a particular crop-country combination, this data was simply assigned from a similar crop or country or created using expert judgment. Neither data on cropping intensities nor farming system shares have been validated for reliability. In terms of

comprehensiveness, notably less sub-national coverage exists in developing countries, and only global average commodity price data was used to account for the economic influences on crop production.

The wide range of data sources, coverage and regional nuances of crop production, have methodological implications. First, there are possible trade-offs between data consistency and data reliability. For example, there are requirements of the model

(i.e., cropping intensities and farming system shares) that are not consistently available within a country at the administrative level needed. Often, these numbers are taken from national-level values, even though they may not reflect the reality in the administrative level. Second, multi- and inter-cropping is not handled in a sophisticated manner within SPAM. These types of cropping patterns are only accounted for using a single cropping intensity value per crop, farming system, and (possibly) sub-national unit. Finally, we rely on population density for an indirect representation of market proximity in SPAM, which

might cause confusion. We use rural population to calculate a prior for *subsistence portion* of a crop (i.e. subsistence among the four farming systems). For subsistence farmers, by definition they mostly consume what they produce and so we indeed assume their production is closely correlated with their rural population size. For all other farmers, we assume they produce for the market (local, regional or even international markets). In fact, market is important for both subsistence farmers and commercial ones. Even in poor countries where self-consumption is high, a large majority of households still purchase food

products produced by others (Losch et al., 2012). This is the assumption behind the revenue calculation to break down the total cropland into individual crop areas in the prior calculation. We did not, explicitly or implicitly, assume rural population (even subsistence farmers) is entirely fed by local agriculture. The confusion comes from our using rural population as a proxy for market access in the revenue calculation. As this crop-specific revenue is divided by the total revenue within a pixel in equations (9 and 10), the prior is not affected by market accessibility if it is not crop-specific. In other words, crop-

specific market accessibility is preferable for the current SPAM model. Yet such accessibility data at the global scale doesn't exist now. We would consider including a more direct representation of market proximity by actual travel times to markets or road networks, as the global roads and railways database are becoming available (Kok et al. 2019). Several trade-offs were made to ensure the complex allocation method was tractable, and it is important to recognize that these trade-offs likely affect the plausibility of results.


Last but not least, we admit that it is inappropriate to compare SPAM products directly across time stages (Figure 13-15), although we have paid every effort to guarantee the spatial accuracy of SPAM products at each time stage. It is largely because of the system errors exist across various data products. In a latest publication, Iizumi and Sakai (2020) released a time-series product of global gridded crop yields. Although they applied a different approach (i.e. spatial adjustment) which

is conceptually different from the spatial disaggregation approach applied in SPAM, it provides great implications to further integrate and standardize the SPAM and the similar gridded Earth System datasets for broader applications. There is an ongoing consortium called The Land Use Change Knowledge Integration Network (LUCKiNet, www.luckinet.org). SPAM team is part of this consortium which aims to integrate tools and standardize approaches across various ongoing projects that develop gridded information on land-use dynamics for applications in food security, climate change, biodiversity, and other

related issue area. Not only LUCKiNet aims to create crop maps comparable over time, we also want to have these maps consistent across land uses such as cropland, grassland, forest. The modelling techniques would consider the spatiotemporal dynamics of different land use forms in an integrative framework.

### 7.4 Concluding remarks

In this paper, we present SPAM2010—the latest global gridded agricultural production dataset in 2010. SPAM2010 uses an updated cross-entropy approach to make plausible estimates of crop distribution for 42 crops and four farming systems within disaggregated units, which shows great improvement than its predecessors: SPAM2000 and SPAM2005. For

example, the expanded crop list not only enables the analysis for staple food crops but also for cash crops. A recent study has analyzed the global beer supply by using SPAM2000 (Xie et al., 2018). It will be very promising to analyze the global coffee and tea supply by using the latest dataset, as these crops are newly included in SPAM2010 which are in an increasing demand with superior economic value but also highly sensitive to climate change (Bunn et al., 2015).

SPAM2010 substantially extends the SPAM family and fills the gap for the work of global agricultural production mapping, by successfully creating a global gridscape at the confluence between earth and farming systems. In particular, it helps to better understand land management practices characterized by concomitant data and knowledge gaps (e.g. crop selection, and element of crop harvest) (Erb et al., 2017). It not only allows analysts and policymakers to better target agricultural and rural development policies and investments, increasing food security and growth with minimal environmental impacts, but also enables scientists to better address the global change challenges within the anthroposphere and beyond by providing the only possibility to update the global agricultural and environmental assessments from year 2000 (when M3, MIRCA, GAEZ, and SPAM2000 are available) to year 2005 and 2010 (when SPAM2005 and SPAM2010 are available as well). All the SPAM maps and tabular data in multiple time stages are freely available on the MapSPAM website (http://mapspam.info/), which also acts as a platform for validating and improving the performance of the SPAM maps by collecting feedbacks from users.

**Supplement.**

The supplement information related to this article is available online.

**Author contributions.**

QY, LY, WW, PY framed the work. QY, LY, UWS, and YR developed the SPAM2010 dataset. QY, LY, ML, WW, PY developed the cropland layer which has been used as the main input data for SPAM2010. SF and WX helped the validation of SPAM2010. UWS, AJ prepared a technical document. QY prepared the manuscript with contributions from all the co-
authors wrote the final paper.

**Competing interests.**

The authors declare that they have no conflict of interest.

**Acknowledgements.**

The work is financially supported by the National Natural Science Foundation of China (41921001 and 41871358), and by
the National Key Research and Development Program of China (2017YFE0104600). It forms part of the CGIAR Research Program on Water, Land and Ecosystems led by International Water Management Institute (IWMI), the CGIAR Research Program on Policies, Institutions, and Markets (PIM) led by International Food Policy Research Institute (IFPRI), and the International Science Technology Practice and Policy Center and GEMS Agroinformatics Initiative at the University of Minnesota. All people and institutes provided data or comments/feedback to SPAM are acknowledged. This work
contributes to the Global Land Programme (http://www.glp.earth).

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

**Figure and tables**

**Figure 1: Overview of the global spatially explicit datasets on agricultural production.**

Each dataset is plotted in a coordinate system with the *x*-axis representing the timespan and the *y*-axis representing the number of crops that have been included. For each dataset, the first row indicates the major measurement(s) of agricultural production, the second row indicates the cropland cover layer, and the third row indicates the main approach for allocating production. The dash line within the chart indicates the evolution of a dataset family.

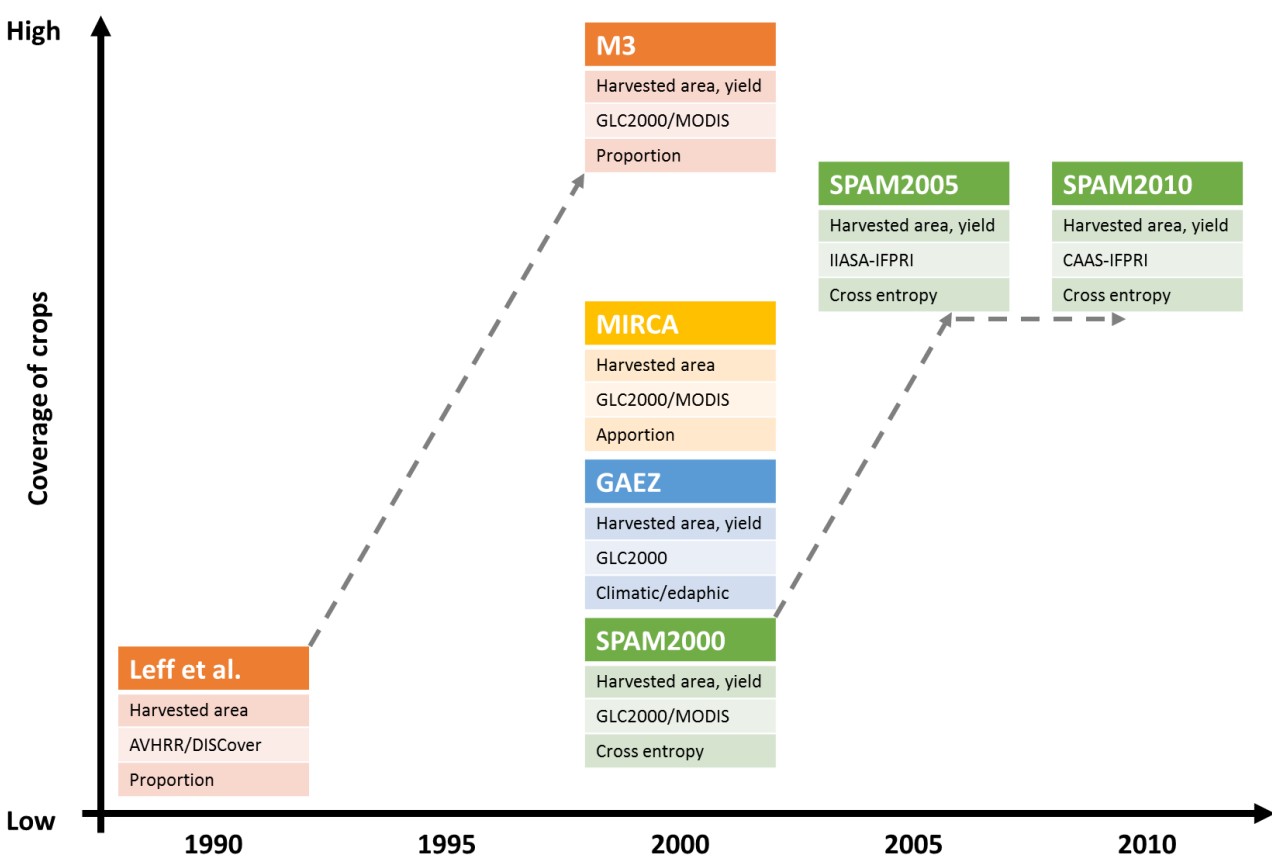

**Figure 2: The overall structure of the SPAM model.**

The rhombuses indicate spatial data inputs/outputs, while the other shapes indicate non-spatial data inputs (see the detailed data description in the following section).

The orange color indicates how crop statistics are disaggregated by administrative unit (k), crop type (j), and farming system (l). The green color indicates how the spatial parameters are collected and prepared at a unified spatial resolution (i) and in a harmonized manner. The yellow color indicates the spatial allocation inputs/outputs.

The darker colors, either in orange or in green, highlight the essential elements in SPAM: the former indicates the farming system disaggregation scheme while the later indicates (i.e., priors of physical area) a key parameter with which the spatial and non-spatial data are connected and the iterative spatial allocation is able to take place.

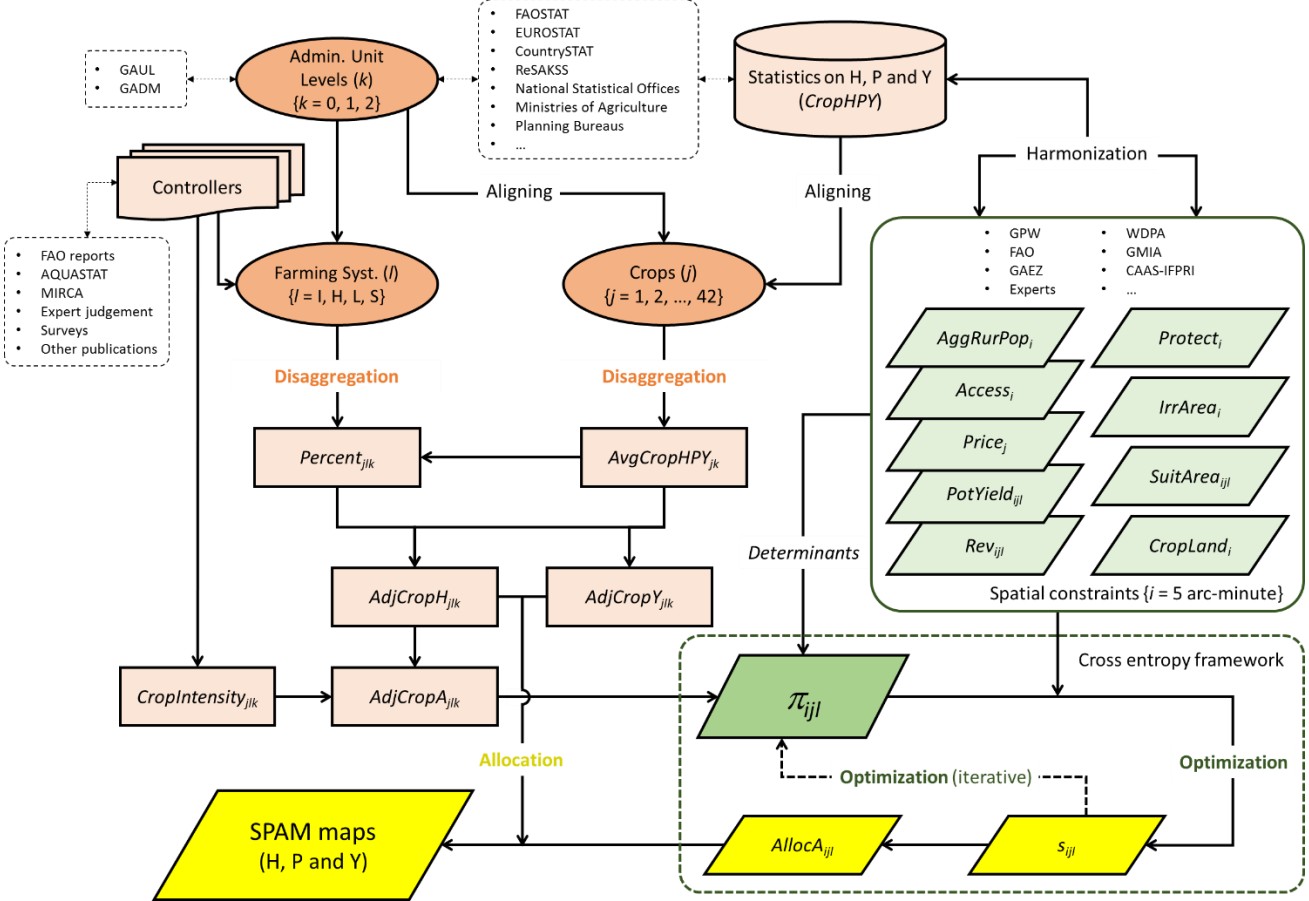

**Figure 3: Harvested area maps for maize in irrigated (a), rainfed–high (b), rainfed–low (c), subsistence (d) and all (e) farming systems.**

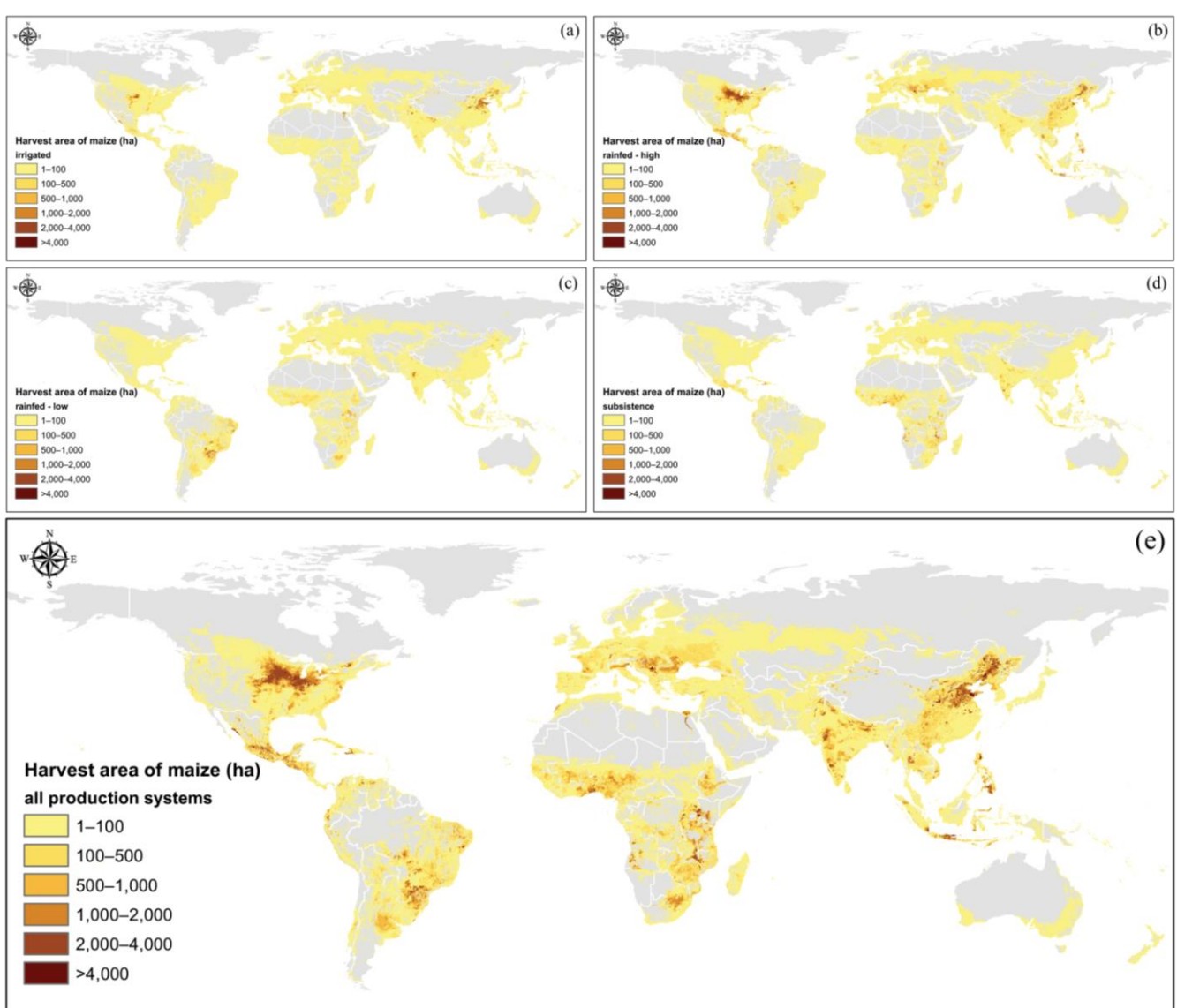

**Figure 4: Yield maps for maize in irrigated (a), rainfed–high (b), rainfed–low (c), subsistence (d) and all (e) farming systems.**

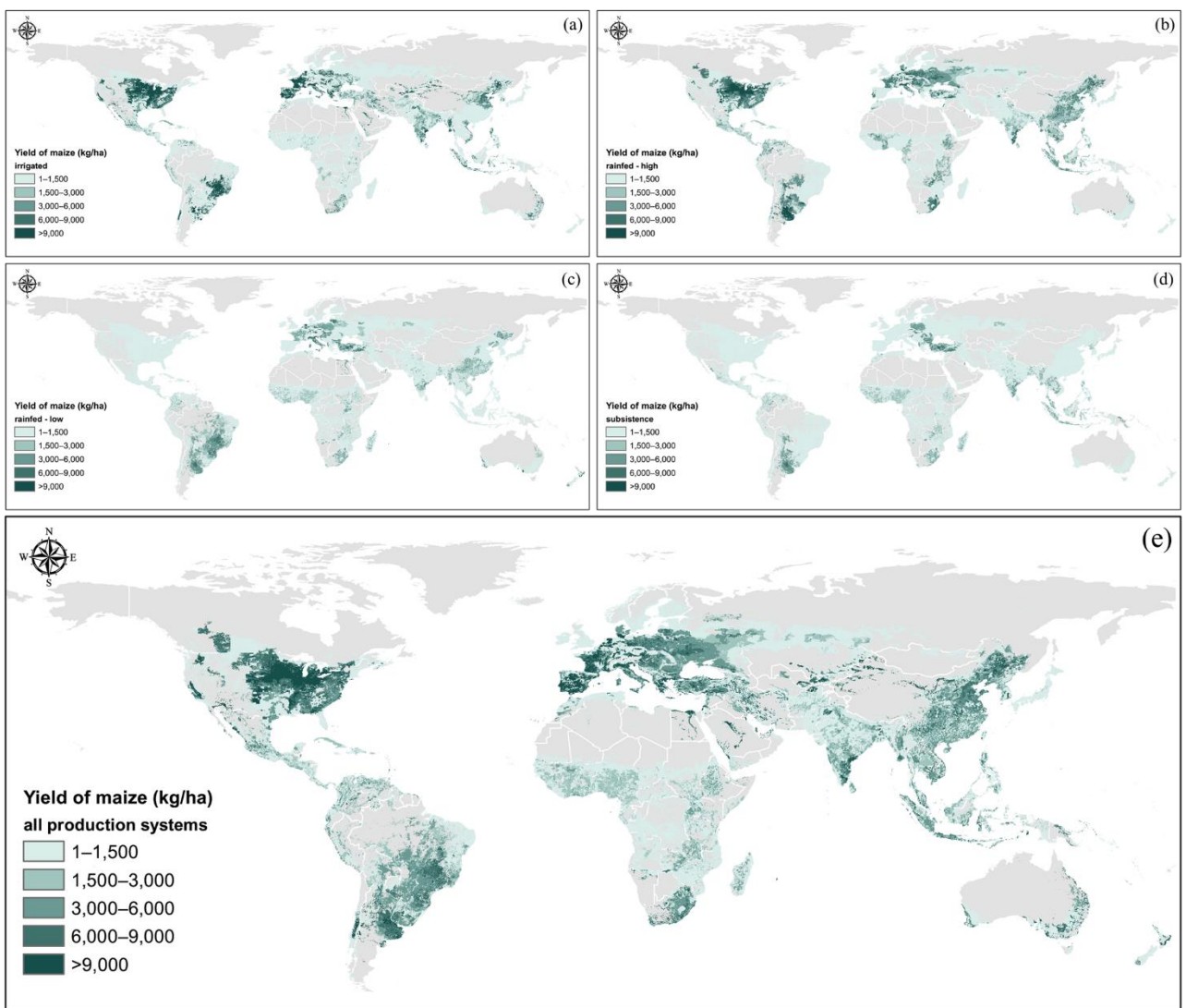

**Figure 5: Subjective uncertainty rating for SPAM2010 input data by individual countries.**

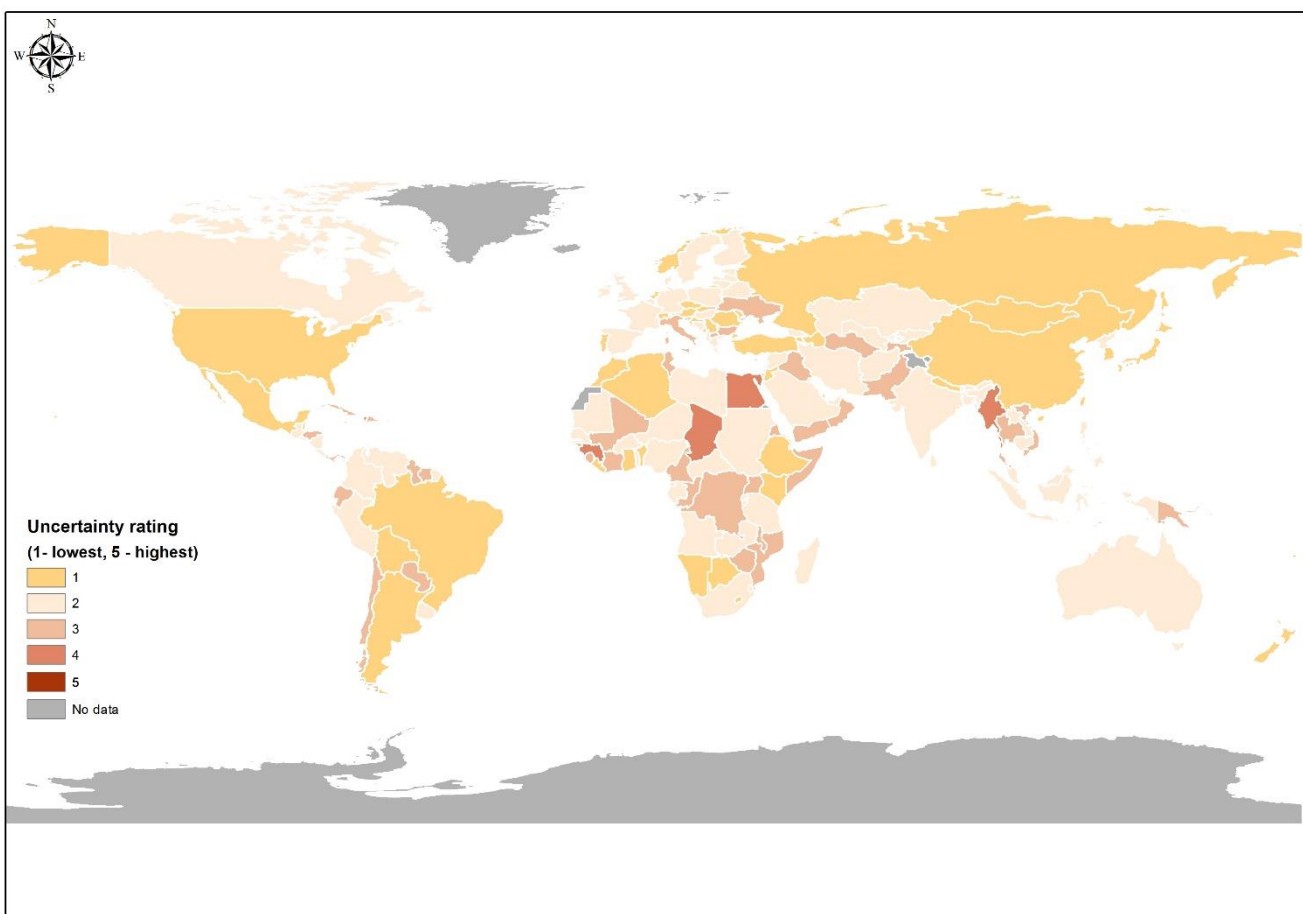


**Figure 6: Comparison between the allocated crop area and statistics crop area at the ADM2 level in Brazil (log-log scale plot, unit: ha.). The upper part is for SPAM2000 and the bottom part is for SPAM2010.**

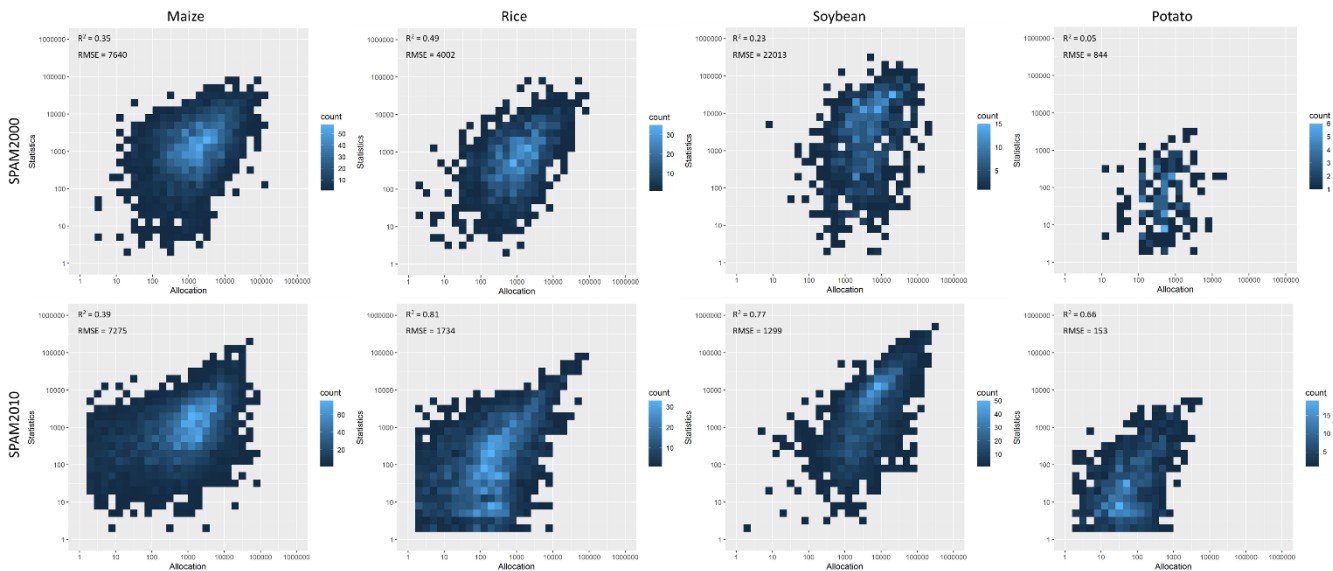

**Figure 7: Comparison between the allocated crop area and statistics crop area at the ADM2 level in Bangladesh, Benin, Senegal and Tanzania for maize, rice and cotton (log-log scale plot, unit: ha.).**

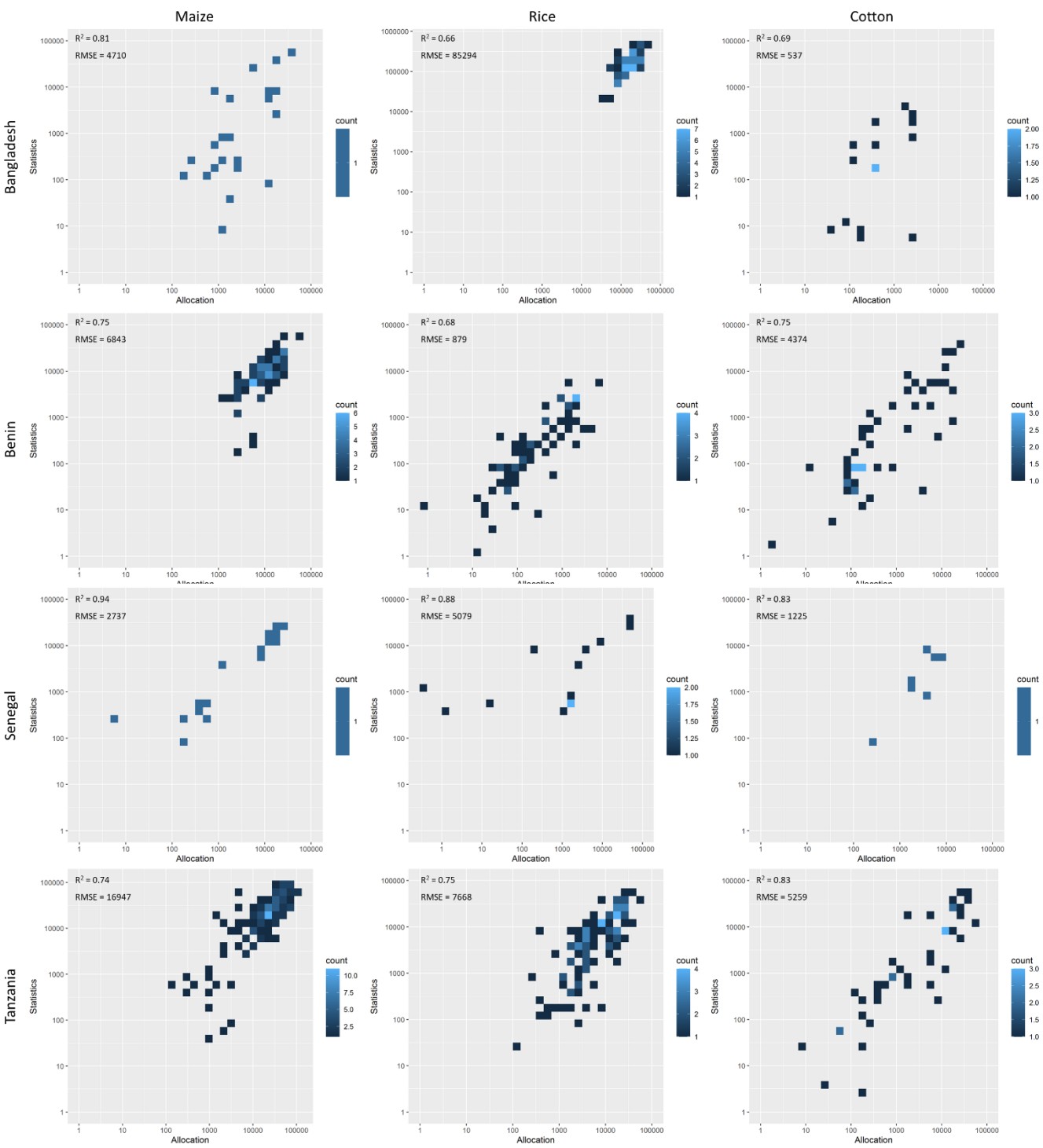

**Figure 8: Grid-by-grid comparison of crop area for maize (a), wheat (b) and soybean (c) between SPAM2010 and CDL2010 in the continental US.**

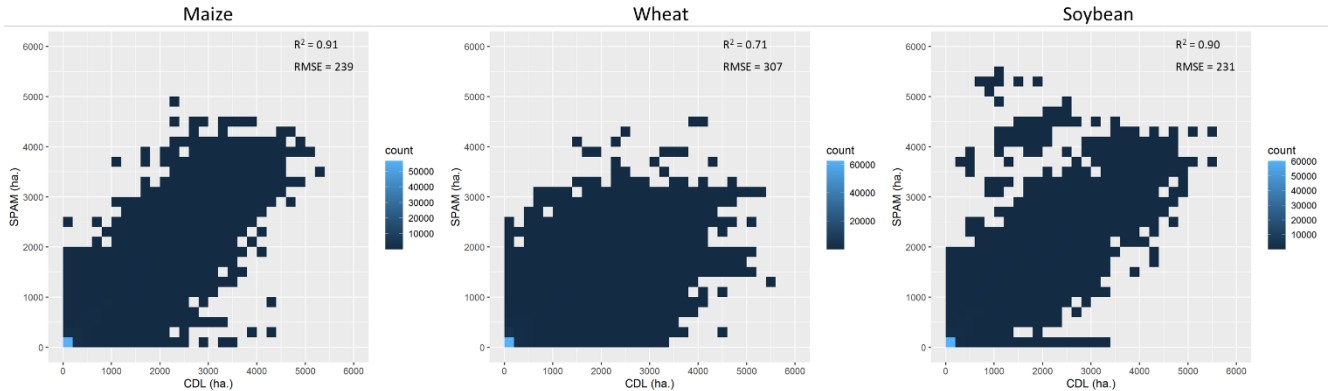

 **Figure 9: Grid-by-grid comparison between SPAM2010 rice area and NLCD2010 paddy field area in China.**

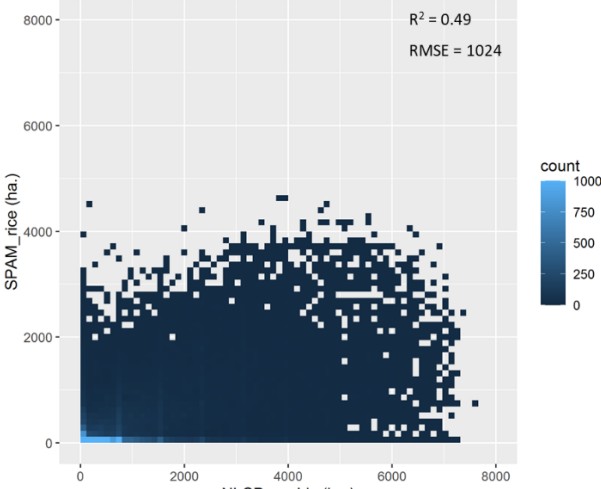

**Figure 10: Grid-by-grid comparison between SPAM2010 and Zhang et al. (2017) rice area in China and India.**

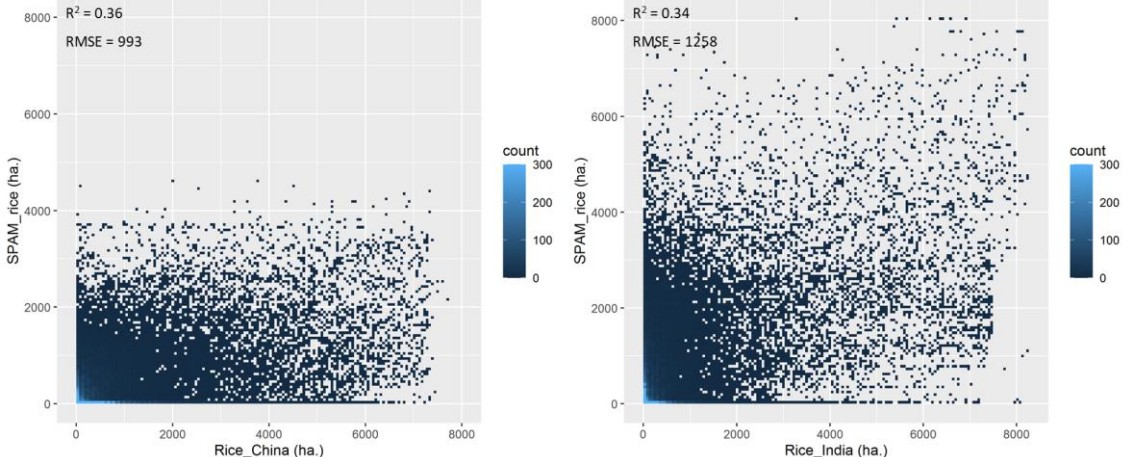

**Figure 11: Grid-by-grid comparison between SPAM2000 and Siebert and Doll (2010) in average irrigated and rainfed yields (log-log scale plot, unit: kg/ha.).**

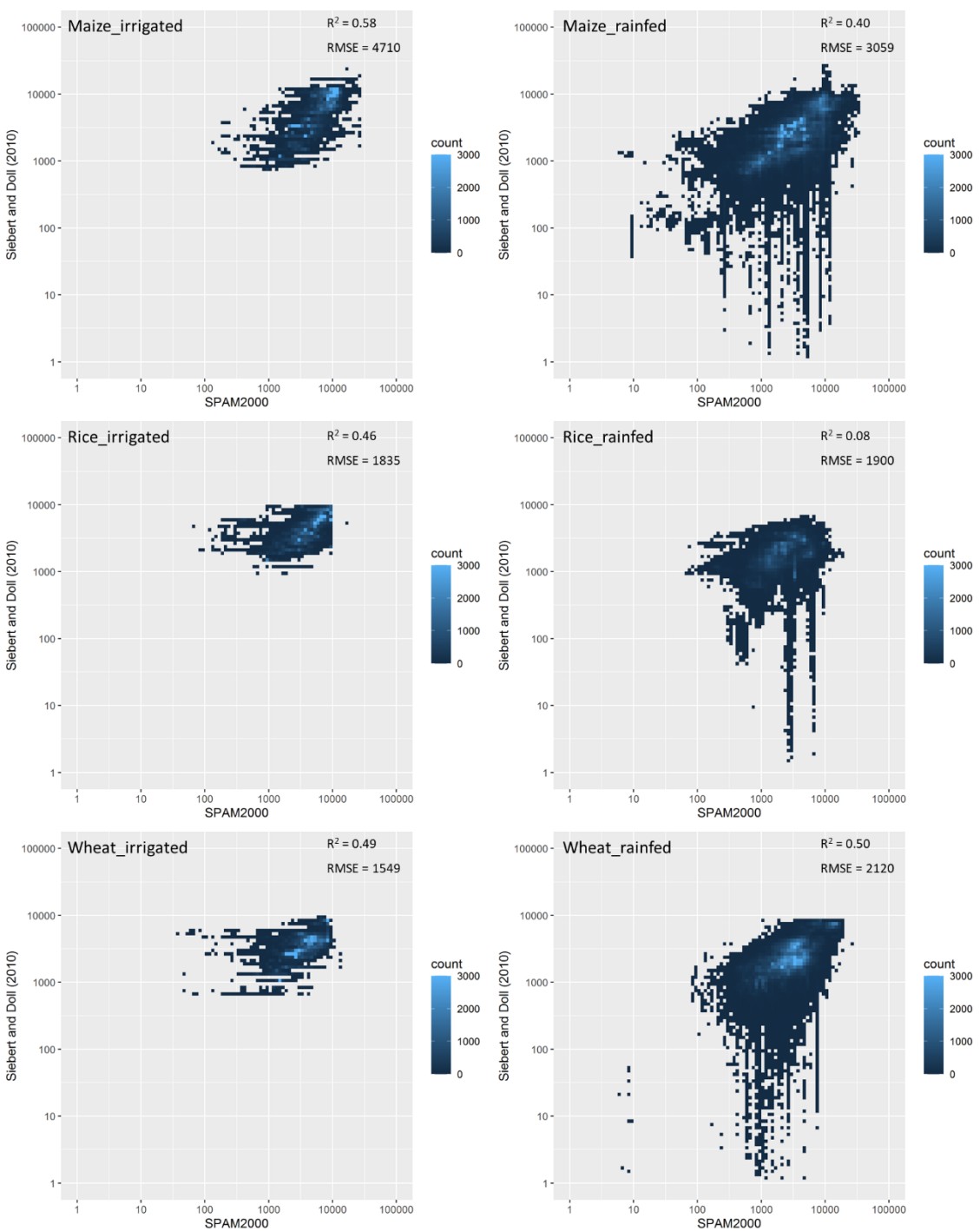

**Figure 12: Grid-by-grid comparison between SPAM2005 and EARTHSTAT2005 in crop yields. (log-log scale plot, unit: kg/ha.).**

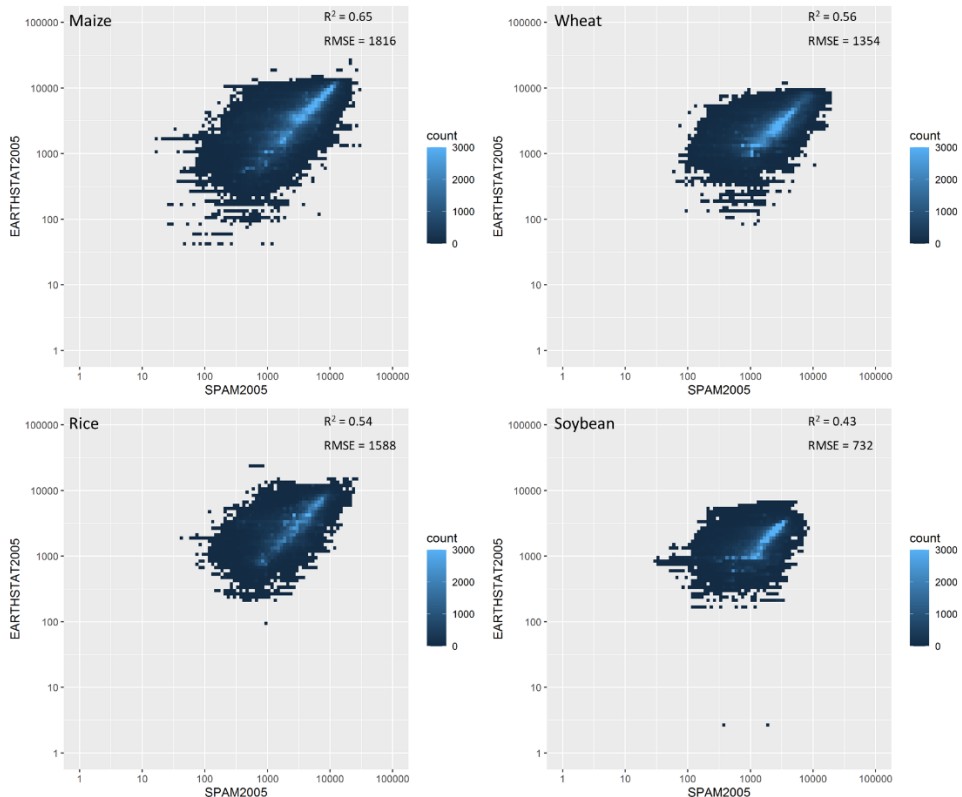

**Figure 13: Comparison between SPAM2010 and SPAM2005: (a) relative difference of rice production; (b) relative difference of maize production.**

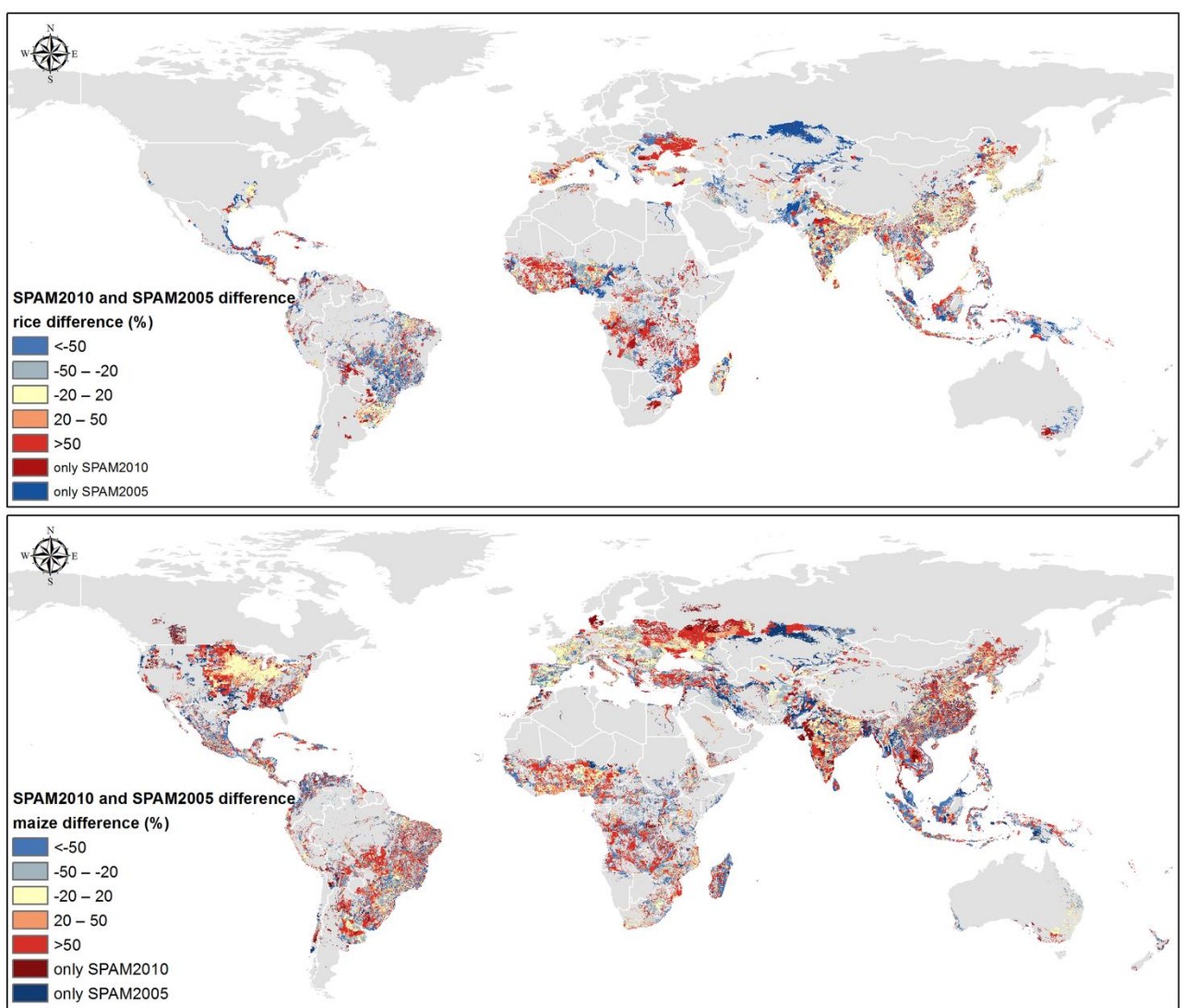

**Figure 14: Comparison between SPAM crop area change and CDL crop area change (log-log scale plot, unit: ha.).**

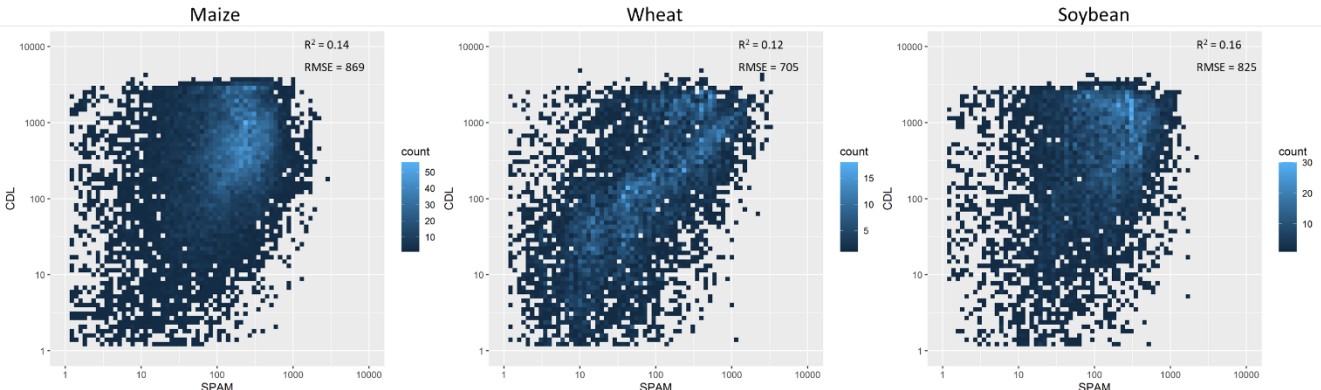

**Figure 15: Comparison between SPAM rice area change and Zhang et al. (2017) paddy rice change (unit: ha.).**

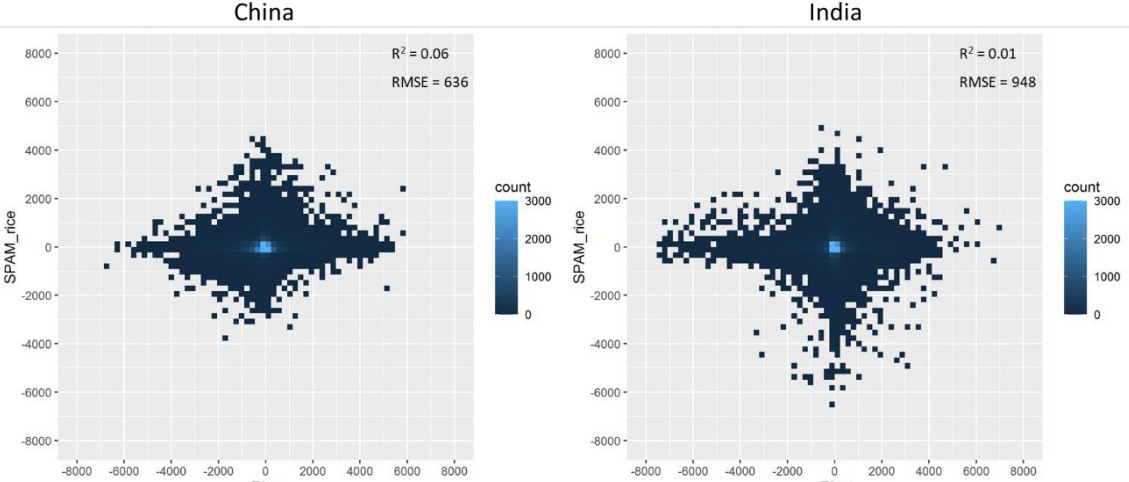

**Table 1: Sub-national coverage of crop production statistics by region.**

| | Region | Countries | ADM1 | ADM2 | Full-Crop Coverage | | National Harvested Area Coverage | | |
| | | | | | ADM1 | ADM2 | Harvested Area | ADM1 | ADM2 |
|---|---|---|---|---|---|---|---|---|---|
| | | (count) | | | (percent) | | (1,000 ha) | (percent) | |
| SPAM2010 | | | | | | | | | |
| | Asia | 25 | 477 | 9,513 | 90.9 | 86.4 | 535,759 | 99.0 | 84.3 |
| | Canada | 1 | 13 | 202 | 100.0 | 81.3 | 25,841 | 100.0 | 85.0 |
| | Europe | 47 | 581 | 3,323 | 86.0 | 74.0 | 169,506 | 91.6 | 45.3 |
| | Central America | 42 | 479 | 11,721 | 84.7 | 92.7 | 146,396 | 97.5 | 81.0 |
| | Middle East | 14 | 167 | 949 | 81.1 | 53.6 | 23,220 | 79.2 | 3.4 |
| | Northern Africa | 6 | 132 | 1,685 | 80.6 | 85.1 | 21,345 | 84.0 | 58.2 |
| | Oceania | 19 | 68 | 789 | 85.5 | 48.8 | 26,379 | 97.3 | 1.5 |
| | Russia | 1 | 91 | 91 | 100.0 | 0.0 | 52,079 | 100.0 | 100.0 |
| | Sub-Saharan Africa | 50 | 687 | 3,928 | 79.0 | 68.6 | 215,896 | 91.5 | 30.9 |
| | United States | 1 | 51 | 3,106 | 100.0 | 87.0 | 98,991 | 99.8 | 94.0 |
| | Total | 206 | 2,746 | 35,307 | 85.1 | 85.2 | 1,315,412 | 96.1 | 68.0 |
| SPAM2005 | Total | 201 | 2,799 | 33,425 | 66.2 | 43.2 | 1,239,026 | 93.4 | 54.6 |

*Source*: Assembled by authors.

Note: Full-crop coverage refers to the percentage of crops at administrative level with positive values or zero in relation to all possible crops. Percent of national harvested area covered by ADM1 or ADM2 is the share of national area harvested reported by ADM1 or ADM2 units. In Russia we had no data for ADM2 units.

The last row presents an overview of data coverage applied for SPAM2005 as a comparison.


**Table 2: The main input variables used in SPAM2010.**

| Variables | Definition | Sources |
|---|---|---|
| $k$ | Administrative unit levels ($k$ = 0, 1, 2) | GAUL, GADM |
| $j$ | Crop type (Total = 42) | FAOSTAT |
| $l$ | Farming system ($l$ = I, H, L, S) | FAO reports etc. |
| $CropHPY$ | Statistics on harvested area ($H$), production ($P$) and yield ($Y$) | FAOSTAT etc. |
| $AvgCropHPY_{jk}$ | $CropHPY$ averaged to 2009-2011 | FAOSTAT etc. |
| $AdjCropHPY_{jk}$ | $AvgCropHPY_{jk}$ averaged to 2009-2011 and scaled to FAO statistics | Developed by authors |
| $Percent_{jlk}$ | Shares of farming systems $l$ by crop $j$ and administrative unit $k$. | Developed by authors |
| $AdjCropH_{jlk}$ | Adjusted harvested area ($H$) by $j$, $l$ and $k$ | Developed by authors |
| $AdjCropY_{jlk}$ | Adjusted yield ($Y$) by $j$, $l$ and $k$ | Developed by authors |
| $AdjCropA_{jlk}$ | Physical area ($A$) by by $j$, $l$ and $k$, $AdjCropH_{jlk}$ divided by $CropIntensity_{jlk}$ | Developed by authors |
| $CropIntensity_{jlk}$ | Harvesting frequency per year per unit cropland by $j$, $l$ and $k$ | Expert judgements etc. |
| $i$ | 5 arc-minute grid cell | Developed by authors |
| $AggMedCropLand_i$ | Median cropland in each grid $i$ | CAAS-IFPRI cropland |
| $AggMaxCropLand_i$ | Maximum cropland in each grid $i$ | CAAS-IFPRI cropland |
| $AggProbCropLand_i$ | Probability that estimated cropland amount is correct in each grid $i$ | CAAS-IFPRI cropland |
| $AdjCropLand_i$ | Total cropland in each grid $i$, after adjustments | Developed by authors |
| $SuitArea_{ijl}$ | Total suitable area in each grid $i$ by crop $j$ and farming system $l$ | GAEZv3.0 |
| $AdjSuitArea_{ijl}$ | Total suitable area in each grid $i$ by crop $j$ and farming system $l$, after adjustments | Developed by authors |
| $IrrArea_i$ | Area equipped for irrigation in each grid $i$ | GMIAv5.0 |
| $AdjIrrArea_i$ | Total irrigated area in each grid $i$ by crop $j$ and farming system $l$, after adjustments | Developed by authors |
| $Protect_i$ | Indicator of protected area in each grid $i$ | WDPA |
| $Price_j$ | Prices for $j$ calculated as a weighted average from world totals | FAO |
| $AggRurPop_i$ | Population density in each grid $i$ | GPWv4.0 |
| $Access_i$ | Market accessibility in each grid $i$ | Developed by authors |
| $PotHarvYield_{ijl}$ | Potential harvested yield from GAEZ in grid $i$ by crop $j$ and farming system $l$ | GAEZv3.0 |
| $PotYield_{ijl}$ | Potential yield calculated from $PotHarvYield_{ijl}$ in grid $i$ by crop $j$ and farming system $l$ | Developed by authors |
| $Rev_{ijl}$ | Potential revenue in each grid $i$ by crop $j$ and farming system $l$ | Developed by authors |
| $\overline{CropA}_{ijl}$ | Prior allocation for physical area in each grid $i$ by crop $j$ and farming system $l$ | Developed by authors |
| $\pi_{ijl}$ | Informed prior of physical area by $i$, $j$ and $l$, calculated from $\overline{CropA}_{ijl}$ | Developed by authors |
| $s_{ijl}$ | Allocated shares of physical area in each grid $i$ by crop $j$ and farming system $l$ | Developed by authors |
| $AllocA_{ijl}$ | Allocated physical area in each grid $i$ by crop $j$ and farming system $l$ | Developed by authors |

*Source:* Developed by authors.

**Table 3: Regional values for area and yield of rice from SPAM2010. Unit: area (1000 ha); yield (kg/ha).**

| Region | Irrigated | | Rainfed–high | | Rainfed–low | | Subsistence | | Total | |
|---|---|---|---|---|---|---|---|---|---|---|
| | Area | Yield | Area | Yield | Area | Yield | Area | Yield | Area | Yield |
| North America | 1,259 | 7,779 | - | - | - | - | - | - | 1,259 | 7,779 |
| Central America | 624 | 4,408 | 147 | 2,582 | 150 | 2,228 | 49 | 2,492 | 969 | 3,698 |
| South America | 2,126 | 7,510 | 536 | 3,320 | 1,261 | 2,536 | 995 | 2,691 | 4,917 | 4,803 |
| Europe | 540 | 6,352 | 146 | 3,315 | 122 | 7,531 | 6 | 5,340 | 814 | 5,977 |
| Meast and Nafrica | 1,034 | 6,848 | - | - | 114 | 4,340 | - | - | 1,148 | 6,599 |
| SSA | 2,172 | 3,844 | 415 | 3,542 | 2,513 | 1,653 | 4,413 | 1,412 | 9,514 | 2,124 |
| South Asia | 32,156 | 4,572 | 11,122 | 3,536 | 3,755 | 1,405 | 12,746 | 1,628 | 59,779 | 3,553 |
| South East Asia | 25,620 | 5,170 | 14,954 | 3,774 | 850 | 1,745 | 7,500 | 1,522 | 48,924 | 4,125 |
| East Asia | 32,609 | 6,610 | 561 | 4,037 | 2 | 626 | - | - | 33,173 | 6,566 |
| Russia | 195 | 5,173 | - | - | - | - | - | - | 195 | 5,173 |
| Oceania | 34 | 9,620 | - | - | 5 | 3,242 | 0 | 1,960 | 39 | 8,665 |
| World | 98,369 | 5,528 | 27,881 | 3,663 | 8,773 | 1,810 | 25,709 | 1,604 | 160,732 | 4,374 |

*Source*: Developed from own-calculations.

**Table 4: Value of production for all crops, food and non-food crops in various regions.**

| Region | All Crops (million I$) | Food Crops (million I$) | (percent) | Non-Food Crops (million I$) | (percent) |
|---|---|---|---|---|---|
| North America | 139,173 | 125,655 | 0.90 | 13,518 | 0.10 |
| Central America | 33,174 | 26,340 | 0.79 | 6,834 | 0.21 |
| South America | 135,222 | 99,950 | 0.74 | 35,272 | 0.26 |
| Europe | 202,180 | 165,761 | 0.82 | 36,419 | 0.18 |
| Meast and Nafrica | 60,643 | 51,358 | 0.85 | 9,285 | 0.15 |
| SSA | 110,406 | 96,474 | 0.87 | 13,931 | 0.13 |
| South Asia | 201,196 | 171,814 | 0.85 | 29,382 | 0.15 |
| South East Asia | 129,189 | 105,878 | 0.82 | 23,311 | 0.18 |
| East Asia | 367,338 | 344,799 | 0.94 | 22,539 | 0.06 |
| Russia | 26,489 | 22,782 | 0.86 | 3,707 | 0.14 |
| Oceania | 13,426 | 11,337 | 0.84 | 2,089 | 0.16 |
| World | 1,418,435 | 1,222,147 | 0.86 | 196,288 | 0.14 |

*Source*: Developed from own-calculations.