# Peer review of "A cultivated planet in 2010: 2. the global gridded agricultural production maps"

_Earth System Science Data, 2020_

## Referee Comment (RC1) · Anonymous Referee #1 · 5 Apr 2020

General comments SPAM products are one of well-known spatially-explicit global agricultural production datasets. An update of SPAM products can be potentially a great contribution to scientific communities (Earth system modeling and global food security monitoring in particular). However, I think, the current form of the Discussion paper is not sufficiently persuasive for some aspects. An evaluation of the validity of the spatial disaggregation method is lacking. Particularly, although the method estimates harvested area and yield for each of the four farming systems (irrigated, rainfed high input, rainfed low input and subsistence) and this is the most unique characteristics of SPAM products, no evaluation is presented in this Discussion paper (because SPAM products are model estimates, earlier papers (You et al. 2006, 2014) cannot justify skipping evaluation in the paper). A comparison between the latest SPAM product and other

independent datasets is partly presented, but there is a space for improvements. For these reasons, I would suggest major revision. My comments are elaborated below.

Specific comments 1. An evaluation of the spatial disaggregation model is required. The most prominent uniqueness of SPAM products, including the latest one (i.e., SPAM2010), is a distinction in harvested area and yield across the farming systems. Currently, global datasets other than SPAM products provide no information on area and yield specific to farming system. However, area and yield for each farming system in SPAM products are "estimates" derived using a spatial disaggregation model optimized using the entropy method. Although the authors may claim that this is a data-fusion approach but not a model prediction approach, a model evaluation against the validation subset (that is independent of the training subset) is essential even for a data-fusion approach. This is a common practice across studies using models even in global crop yield dataset compilation (Iizumi et al. 2014; grid-cell yield estimates derived using national yield statistics as the model input are compared with reported subnational yield statistics which are not used as the model input). Note that M3 and MIRCA2000 use a simple allocation rule rather than modeling; and GAEZ is a model output but for "potential" geographic distribution of crop suitable area. However, the purpose of this Discussion paper is to present "actual" distributions of area and yield for specific farming systems. Therefore, an evaluation of the model used is a mandate. Probably, for some crop-region combinations, the authors have farming-system-specific area and yield statistics at subnational levels. I strongly encourage the authors testing and reporting the performance of their model in disaggregating national agricultural statistics into subnational ones when national statistics are used as the model inputs. 2. A comparison of SPAM products and other independent datasets has a space for further improvements. The key shortfalls in the current Discussion paper are: (1) although CDL2010 for the United States and NLCD2010 for China are compared with SPAM2010, these are for harvested area and no comparison is presented for area and yield for the specific farming systems; and (2) although the relative changes in area between 2005 and 2010 are presented in the paper (Fig. 8),

these need be compared with other independent datasets (for instance, CDL2005 and CDL2010 for the United States). The updated M3 dataset which offers the average harvested area and yield for three time points, 1995 (1993–1997), 2000 (1998–2002) and 2005 (2003–2007) is a candidate for the independent dataset and is available online at: http://www.earthstat.org/ (see the dataset labelled "Harvested Area and Yield for 4 Crops (1995-2005)"). For a consistent comparison, if possible, I would encourage the authors updating the earlier SPAM2000 and SPAM2005 products by utilizing the model used for SPAM2010. Such updating is a common practice in global agricultural dataset compilation and important to ensure the continuity of data in products (Iizumi and Sakai 2020, Sloat et al. 2020). 3. Related to the comment #2, Zhang et al. (2017) provides annual paddy area time series from 2000 to 2010 based on satellite remote sensing for China and India. Because recent satellite-based paddy area estimates are quite accurate, this dataset can be a useful source of information to evaluate the relative changes of paddy area in SPAM products. 4. Related to the comment #2, a distinction between average irrigated and rainfed yields for the 1998–2002 period at the global scale is made in Siebert and Doll (2010). These estimates are also used in recent study (Sloat et al. 2020). I think, these estimates can be a useful source of information when evaluating the reliability of farming-system-specific estimates in the SPAM products once updating of SPAM2000 and SPAM 2005 using the latest model is done. 5. A more in-depth discussion on advantages, disadvantages and limitations of the spatial disaggregation model is required. Although the authors hypothetically assume that the use of economic factors, including prices and access to markets, in the disaggregation model is superior to other methods, such as the proportional allocation. However, this working hypothesis has never been tested (at least, I could not find any result neither in this Discussion paper nor in earlier work (You et al. 2009, 2014)). "garbage in garbage out" is a well-known behavior of models. In general, price statistics are less reliable than other variables (e.g., production). I have the same concerns for the quality of data on production share by farming system and the indicator of market access. If some of model inputs are not reliable, model outputs are expected to be unreliable, depending

on the sensitivity of model output to specific inputs. I like the idea that economic factors are considered in disaggregation, but the idea does not automatically guarantee that model outputs (disaggregated area and yield by farming system) is correct. I think, the advantages of the model relative to simpler methods are stated too bold throughout the Discussion paper. The authors' claims might be true, but need be tested in a standard way of model evaluation (e.g., by using the cross-validation technique).

Technical corrections 6. L71-73. I strongly suggest removing this description. Researchers would use the latest version once global agricultural dataset is updated, but no such update is available to date. This is the reason why the studies cited here use an earlier version. The authors' criticism made here is inappropriate. 7. L107. The current text is a bit misleading. This text should read "M3 has no distinction across farming systems . . ." or similar. 8. L156. Country crop-specific production costs for a specific year (e.g., 2011) are available via GTAP9 database (Aguiar et al. 2016). Just for your information. 9. L158. GAEZ only provides "potential" crop suitability area. Please consider keeping precise terminology in the Discussion paper. 10. Eq. 7. What is "CE"? The abbreviation suddenly appears without definition. And I would appreciate it if the authors could provide a brief explanation what is the difference between {s ln s} versus {s ln $\pi$}. 11. Eq. 16. AdjCropY suddenly appears in main text although it is explained in Supplement. A brief explanation need be added in main text for readability. 12. L304-305. Are the yield conversion factors in the text same with those shown in Table S6? Table S6 shows only for irrigated versus rainfed. Where is rainfed high input versus rainfed low input? 13. 372-373. This assumption is too crude. Dong et al. (2017) presents a nice global dataset in specifying urban areas. It can be useful to distinguish rural and urban areas more accurately. 14. L532. I do not understand "methodological-cum-data". Please consider rephrasing. 15. L626-627. This is true but has not been demonstrated yet. I would suggest removing this statement unless a comparison in area and yield for each farming system against subnational statistics is presented. 16. L636. Zhang et al. (2017) reports the northward shift of paddy area in China and the westward shift of paddy area in India for the 2000-2010 period. These

tendencies seem be inconsistent with the upper panel of Fig. 8. 17. L679-680. Global roads and railways database used in Koks, E.E. et al. (2019) is maybe of your interest to more accurately define accessibility to markets. Just for your information.

References ïĆǎ Aguiar, A. et al. (2016) An Overview of the GTAP 9 Data Base. Journal of Global Economic Analysis, 1,181–208, http://dx.doi.org/10.21642/JGEA.010103AF ïĆǎ Dong, Y. et al. (2017) Global anthropogenic heat flux database with high spatial resolution. Atmospheric Environment, 150, 276-294, https://doi.org/10.1016/j.atmosenv.2016.11.040. ïĆǎ Koks, E.E. et al. (2019) A global multi-hazard risk analysis of road and railway infrastructure assets. Nature Communications, 10, 2677, https://doi.org/10.1038/s41467-019-10442-3 ïĆǎ Iizumi, T. et al. (2014), Historical changes in global yields. Global Ecology and Biogeography, 23, 346-357, doi:10.1111/geb.12120 ïĆǎ Iizumi, T., Sakai, T. (2020) The global dataset of historical yields for major crops 1981–2016. Sci Data 7, 97, https://doi.org/10.1038/s41597-020-0433-7 ïĆǎ Siebert, S. & Doll, P. (2010) Quantifying blue and green virtual water contents in global crop production as well as potential production losses without irrigation. Journal of Hydrology, 384, 198–217, https://doi.org/10.1016/j.jhydrol.2009.07.031 ïĆǎ Sloat, L. L., et al. (2020) Climate adaptation by crop migration. Nature Communications, 11, 1243, https://doi.org/10.1038/s41467-020-15076-4 ïĆǎ Zhang, G, et al. (2017) Spatiotemporal patterns of paddy rice croplands in China and India from 2000 to 2015, Science of The Total Environment, 579, 82-92, https://doi.org/10.1016/j.scitotenv.2016.10.223.

---

## Referee Comment (RC2) · Patrick Meyfroidt (Referee) · 10 Apr 2020

\*\*\*\*\*\*\*\*\*\*\*\*\*\* General comments: \*\*\*\*\*\*\*\*\*\*\*\*\*\*

This paper presents the latest update of the SPAM global gridded crop maps for 2010. Overall, this is a very valuable effort. Yet, in its current form, I have several general remarks:

1/ The method is insufficiently explained and unclear in some places. A series of expert judgments are used along the way, and although this is acknowledged in the description of the methods, this seems insufficiently acknowledged in the Abstract and Introduction. Overall, this raises concerns about the transparency and reproducibility of the work, but also makes it very unclear what is the same and what is changed

compared to previous versions, thereby justifying a new paper.

2/ Validation: This is a model (mixing reproducible rules and expert judgments), and as such, one would expect more rigorous and transparent validation efforts. Here it appears very thin and, in the words of the Authors themselves, the uncertainty assessment "is not a scientific, rigorous" one.

3/ Beyond operational uses for agencies focusing on crop production, the paper does not discuss how can these efforts serve more broadly scientific agendas regarding an improved understanding of the role of land management in global environmental change, earth system dynamics and other global sustainability issues (e.g., see Erb et al. 2016 in GCB for a discussion)? This would be useful to make the paper more valuable in itself beyond "just" presenting the dataset (no offence here, this is of course a great achievement!).

I return to these main comments below.

************** Methods: **************

* If this is an update with just purely the same methodology, it should not be an extra scientific paper. If there are substantial changes (improvements) in the methodology, then previous validations should not be taken for granted. Here, it is not totally clear what is new and should be validated, versus what is standard.

* Overall, the explanation of the method is unclear in many places. Being familiar with many of these gridded products, but not very much with the previous versions of SPAM in particular, I really have a hard time understanding the approach here. I am dubious that a reader that has not read the previous methods papers can understand what the Authors have done here.

* The method is insufficiently explained: 3.1: The 4 farming systems are explained, but not how the disaggregation between these 4 is done. The answer seems to be actually in Section 4.1.3, but here the answer is essentially "we do it, based on multiple

information and stuff, trust us". Figure 2 is supposed to present an "illustration" ("We present an illustration for obtaining the farming system shares by crop j and administrative unit k (Percent jlk ) in Figure 2"), but Figure 2 doesn't give any information on how this disaggregation is done.

\* Then Section 3.2 explains the optimization but honestly, I understand the equations but it doesn't allow me to understand the process itself.

\* Section 3 does not clarify explicitly what methodological aspects are the same as in previous versions, versus those that have been modified or are new.

\* 4.1.3 Crop statistics disaggregated by farming systems: » This seems to be a mix of various approaches. Can you at least clarify the share of cropland disaggregation achieved based on statistics versus some expert knowledge or assumptions?

\* p.5: "The rainfed subsistence farming system (S), which is also low input as well, and is introduced to account for situations where cropland and suitable areas do not exist, but farmland is still present in some way." » This is very unclear.

\* Accessibility: This comes in Eq. 1 in Section 3.2, and then is detailed in 4.2.5, but I don't understand what is the rationale for creating / using an "accessibility" to market dataset based only on rural population? Is there an assumption that urban populations are fed from anywhere on the planet through global supply chains without this creating any particular incentive for farmers in surroundings (so that only rural population create a revenue incentive as per Section 4.2.6)? This would be quite a strong assumption. What is the rationale behind?

\* Overall, there is a lot of expert judgment and wiggling with the data (see S4, S13-15 etc) (e.g., Section 4.3.1 "Under these circumstances, we set several options to "force" a solution, including adjusting the entropy conditions, and adjusting the data harmonization rules. We elaborate on the details for adjusting areas (Section S13), entropy conditions (Section S14) and harmonization rules (Section S15) respectively in

the SI."). So this is far from resulting from a clean and reproducible algorithm based on simple economic rules. I don't want to distrust the work done by the Authors, but given this large amount of expert-driven decisions, this should be very clearly stated in the abstract and main results / Conclusion, so that the reader understands clearly that this is largely an expert-driven process, with multiple human decisions and assumptions, more than a simply reproducible algorithmic work that produces a transparent output.

\*\*\*\*\*\*\*\*\*\*\*\*\*\* Validation: \*\*\*\*\*\*\*\*\*\*\*\*\*\*

\* Same as above and general comment: First, this is a model; and thus it should be validated properly as far as possible. I understand of course that by the nature of the work done, there is no simple, global, adequate validation data ready to be used. But still, (i) there are ways to do more & better, and (ii) the current efforts are reported in an unclear manner.

\* If, as you explain, you run most countries with data at ADM0 level, but you do have incomplete data at finer administrative levels, then you can at least validate against these incomplete subnational data. This is explored in Figure 5 but given the breadth of the map, just one example is not sufficient.

\* Partial validation could also be achieved through a sampling of points, with visual interpretation of high-resolution imagery to at least identify irrigated systems versus non-irrigated intermediate categories versus the subsistence category. Even some specific crops could be assessed, at least some perennial crops like oil palm, banana, or others.

\* You can't just say (l.539): "As the coverage, quality and spatial precision of data input are much better for SPAM2010 than for its predecessors (see Section 4), the reliability of the data product is believed to improve as well."

\* l.548: "Firstly, we evaluate the results by sending the crop maps to collaborators and users alike for comments or assessment. For example, the CGIAR..." » I don't

understand how this is an "example". Either you did it and you report the results, or you explicitly state that this is something that you have not done but could do.

* "We took advantage of their vast network of field offices and local expertise to help us to validate the SPAM results. Many researchers from these institutes have been involved in the production of SPAM2010, which increases the reliability of the results." » If this has been done, then you should report in more details the outcome of this process, the validation data collected...

* "The validating information could either be collected by" » "Could be", or it has been done? If the former, then it's not useful. If the latter, then provide the results.

* "We take these feedbacks and re-run SPAM model and release updated versions of SPAM. The complete validation process could take a great deal of effort and time, but these users' feedbacks are quite important and valuable." » Same, not clear, is this something you plan to do, or something you have done and can provide data about? The use of present tense makes it confusing.

* "The current product, i.e., SPAM2010v1.1, is also expected to have major updates" » Then is it the right time to release it? Wouldn't it be better to have this round of validation - improvement first?

* "Secondly we do a regional validation in case that the third-party independent crop maps are available," » Same, present time: Does that mean you have done it? Or does that mean this is an aspirational goal that at some point you hope you can do it? Here, as you provide the comparison with US data in Figure 5, it appears that this is something that you have actually done. But (i) we have to guess it, and (ii) it's not clear for all the above.

* Figure 8: Differences are huge. I understand that this mixes real changes on the ground and changes in the methods. But over - nominally - 5 years, this appears to be predominantly dues to changes in the methods. Please elaborate further (note, this is

in relation to the above point on Methods, as it is not fully clear what is stable and what has changed in the Methods).

* l.604: "In addition, we collect feedback and comments from users, local experts and collaborators as discussed above. They are sporadic but very useful. We combine all the information together to give a subjective rating on how confidence we, SPAM team, think of our final crop maps (both area and yield). This is the uncertainty rating we provided here. It is not a scientific, rigorous rating and so we put it only into 1 to 5 categories (1 represents the lowest uncertainty, 5 the highest)." » If this is not a "scientific" rating does it belong to a "scientific" paper?

************** Minor comments: **************

* Abstract: I don't understand this sentence: "but also dedicates as platform providing archived global agricultural production maps for better targeting the Sustainable Development Goals by making proper agricultural and rural development policies and investments"

* Overall the writing is good, but there's a series of weird words, typos and stuff like l. 363: "protected areas. but if the" or l.371 "rural pulation density" (just to give examples, there's plenty of these). Please triple-check through.
* * *

---

## Author Comment (AC1) · 20 Jun 2020

Dear Referee,

Thank you for the comments concerning our Discussion paper entitled "A cultivated planet in 2010: 2. the global gridded agricultural production maps" (Ref. essd-2020-11). These comments were very helpful for revising and improving our paper. To make the reply more readable, we list the comments and corresponding responses one by one in the Authors' Response (AC). The detailed revisions are embedded in the manuscript with the line numbers indicated in the AC.

\*\*\*\*\*\*\*\*\*\*\*\*\*\*\*\*\*\*\*\*\*\*\*\*\*\*\*\*\*\*\*\*\*\*\*\*\*\*\*\*\*\*\*\*\*\*\*\*\*\*\*\*\*\*\*\*\*\*\*\*\*\*\*\*\*\*\*\*\*\*\*\*\*\*\*\*\*\*\*\*\* General comments
SPAM products are one of well-known spatially-explicit global agricultural production

datasets. An update of SPAM products can be potentially a great contribution to scientific communities (Earth system modeling and global food security monitoring in particular). However, I think, the current form of the Discussion paper is not sufficiently persuasive for some aspects. An evaluation of the validity of the spatial disaggregation method is lacking. Particularly, although the method estimates harvested area and yield for each of the four farming systems (irrigated, rainfed high input, rainfed low input and subsistence) and this is the most unique characteristics of SPAM products, no evaluation is presented in this Discussion paper (because SPAM products are model estimates, earlier papers (You et al. 2006, 2014) cannot justify skipping evaluation in the paper). A comparison between the latest SPAM product and other independent datasets is partly presented, but there is a space for improvements. For these reasons, I would suggest major revision. My comments are elaborated below.

Authors' Response: Thanks for these general comments and they are very constructive and helpful for improving the paper. We were aware that previous validations should not be taken for granted for the latest updates. As suggested, we underwent a major revision and added several additional analyses, in particular on the evaluation and validation of the results, which mainly include: (1) Cross-checking the national and subnational level statistics. (Comment#1) (2) Cross-checking with the paddy area maps in China and India. (Comment#3) (3) Comparing the changes existing in SPAM products (e.g. between SPAM2005 and SPAM2010) with the changes detected from other products (e.g. between CDL2005 and CDL2010). (Comment#2 and #3) (4) Comparing the yields and farming system yields with other products. (Comment#2 and #4) More details are in the following point-by-point responses.

**************************************************************************** Specific comments
Comment#1. An evaluation of the spatial disaggregation model is required. The most prominent uniqueness of SPAM products, including the latest one (i.e., SPAM2010), is a distinction in harvested area and yield across the farming systems. Currently, global datasets other than SPAM products provide no information on area and yield

specific to farming system. However, area and yield for each farming system in SPAM products are "estimates" derived using a spatial disaggregation model optimized using the entropy method. Although the authors may claim that this is a data-fusion approach but not a model prediction approach, a model evaluation against the validation subset (that is independent of the training subset) is essential even for a data-fusion approach. This is a common practice across studies using models even in global crop yield dataset compilation (Iizumi et al. 2014; grid-cell yield estimates derived using national yield statistics as the model input are compared with reported subnational yield statistics which are not used as the model input). Note that M3 and MIRCA2000 use a simple allocation rule rather than modeling; and GAEZ is a model output but for "potential" geographic distribution of crop suitable area. However, the purpose of this Discussion paper is to present "actual" distributions of area and yield for specific farming systems. Therefore, an evaluation of the model used is a mandate. Probably, for some crop-region combinations, the authors have farming-system-specific area and yield statistics at subnational levels. I strongly encourage the authors testing and reporting the performance of their model in disaggregating national agricultural statistics into subnational ones when national statistics are used as the model inputs.

Authors' Response: Thanks for the constructive comment. Actually, the validation by cross-checking national and subnational level statistics has been applied for SPAM2000 (e.g. Brazil). Following the comment, we have re-applied the approach for the current SPAM2010 for a few selected countries such as Brazil, Bangladesh, Benin, Senegal, Tanzania. We find that the performance has generally improved comparing to the performance of SPAM2000 though this varies from country to country, and from crop to crop. We add Figure 6 and Figure 7 and the relevant description of the validation process in the revised manuscript.

Comment#2. A comparison of SPAM products and other independent datasets has a space for further improvements. The key shortfalls in the current Discussion paper are: (1) although CDL2010 for the United States and NLCD2010 for China are

compared with SPAM2010, these are for harvested area and no comparison is presented for area and yield for the specific farming systems; and (2) although the relative changes in area between 2005 and 2010 are presented in the paper (Fig. 8), these need be compared with other independent datasets (for instance, CDL2005 and CDL2010 for the United States). The updated M3 dataset which offers the average harvested area and yield for three time points, 1995 (1993–1997), 2000 (1998–2002) and 2005 (2003–2007) is a candidate for the independent dataset and is available online at: http://www.earthstat.org/ (see the dataset labelled "Harvested Area and Yield for 4 Crops (1995-2005)"). For a consistent comparison, if possible, I would encourage the authors updating the earlier SPAM2000 and SPAM2005 products by utilizing the model used for SPAM2010. Such updating is a common practice in global agricultural dataset compilation and important to ensure the continuity of data in products (Iizumi and Sakai 2020, Sloat et al. 2020).

Authors' Response: We have carefully considered this comment by referring to relevant literature and datasets, e.g. Iizumi and Sakai (2020) and "Harvested Area and Yield for 4 Crops (1995-2005)". We would like to elaborate that we have been updating SPAM products over the years by using the same approach (i.e. the cross-entropy model), although not in the same way as Iizumi and Sakai (2020) did with their global crop yield dataset. These suggested comparisons (over time) might improve the reliability of the datasets. Yet further uncertainties might be introduced as well. The main reasons are: (1) "Harvested area" is conceptually different from "yield". For example, the value of harvested area at the country level needs to be equivalent, in theory, to the summed value of all sub-national administrative units. While the value of yield at the country level could be equaling to any value at the sub-national level. This means that the idea of Iizumi and Sakai (2020), i.e. adjusting country-level average yield to spatial grid by considering the spatial variation of NPP, can not be directly applied for disaggregating harvested area from coarser spatial units to finer spatial units. (2) The general framework of cross-entropy model remains the same for SPAM2000, SPAM2005, and SPAM2010. The major difference among them is the input data such as cropland, subnational statistics. In fact, we have kept updating all SPAM products over the years with different versions (e.g. after feedbacks from users, and new input data are available). For example, the latest SPAM 2000 is Version 3.07, the latest SPAM 2005 Version 3.20, and the current version of SPAM 2010 is Version 1.1. (3) Even if SPAM2000, SPAM2005 and SPAM2010 were produce by the same approach (i.e. the cross-entropy model), it does not mean the products can be compared directly across years. Because SPAM requires for a large amount of input data, yet the sources of these multiple data inputs can not be guaranteed as the same across different time stages. For example, the cropland layers (one of the most important data inputs) are accessed from different sources to make sure the cropland data and the statistical data are adopted for the same year. We do not evaluate the continuity of these input data, which is almost impossible and is beyond the purpose of SPAM. Therefore, we do not recommend users to cross compare the SPAM products, because such differences may have more input data errors/inaccuracies than detecting the real change on the ground. Nevertheless, we have added the following comparisons as suggested: (1) Comparing yield for four crops by referring to EARTHSTAT2005. (2) Comparing the area changes in maize, wheat and soybean between CDL2005 and CDL2010 (i.e. $\Delta$CDL), and then compare the $\Delta$CDL between $\Delta$SPAM. We find and admit that these comparison results are not so good. You raised a very good question and there is an ongoing consortium called The Land Use Change Knowledge Integration Network (LUCKiNet, www.luckinet.org). SPAM team is part of this consortium which aims to integrate tools and standardize approaches across various ongoing projects that develop gridded information on land-use dynamics for applications in food security, climate change, biodiversity, and other related issue area. Not only LUCKiNet aims to create crop maps comparable over time, we also want to have these maps consistent across land uses such as cropland, grassland, forest. The modelling techniques would consider the spatiotemporal dynamics of different land use forms in an integrative framework. We have acknowledged the latest publication i.e. Iizumi and Sakai (2020) and included these two comparisons in the revised manuscript. Please see the newly added Figure 12 and 14, and the relevant

text in section 7.

Comment#3. Related to the comment#2, Zhang et al. (2017) provides annual paddy area time series from 2000 to 2010 based on satellite remote sensing for China and India. Because recent satellite-based paddy area estimates are quite accurate, this dataset can be a useful source of information to evaluate the relative changes of paddy area in SPAM products.

Authors' Response: Thanks for the comment. As suggested, we have obtained the paddy rice maps from Zhang et al. (2017) and added the comparison between these maps with the rice area estimated by SPAM2010. In addition, we compared the $\Delta$Rice (difference between the rice map in 2005 and 2010) between $\Delta$SPAMrice (difference between SPAM2005 and 2010). Please see the newly added Figure 10 and 15, and the relevant text in section 7.1 and 7.2.

Comment#4. Related to the comment #2, a distinction between average irrigated and rainfed yields for the 1998–2002 period at the global scale is made in Siebert and Doll (2010). These estimates are also used in recent study (Sloat et al. 2020). I think, these estimates can be a useful source of information when evaluating the reliability of farming-system-specific estimates in the SPAM products once updating of SPAM2000 and SPAM 2005 using the latest model is done.

Authors' Response: As we have responded in comment#2, we are not able to update SPAM following the same way as Iizumi ad Sakai (2020) did with their global crop yield dataset. In fact, SPAM has been compared with MIRCA in terms of irrigated and rainfed area in one of our previous paper (Anderson et al., 2015). As suggested, we underwent a new comparison between SPAM and Siebert and Doll (2010), in terms of irrigated and rainfed yields. Please see the newly added Figure 11, and the relevant text in section 7.2.

Comment#5. A more in-depth discussion on advantages, disadvantages and limitations of the spatial disaggregation model is required. Although the authors hypothetically assume that the use of economic factors, including prices and access to markets, in the disaggregation model is superior to other methods, such as the proportional allocation. However, this working hypothesis has never been tested (at least, I could not find any result neither in this Discussion paper nor in earlier work (You et al. 2009, 2014)). "garbage in garbage out" is a well-known behavior of models. In general, price statistics are less reliable than other variables (e.g., production). I have the same concerns for the quality of data on production share by farming system and the indicator of market access. If some of model inputs are not reliable, model outputs are expected to be unreliable, depending on the sensitivity of model output to specific inputs. I like the idea that economic factors are considered in disaggregation, but the idea does not automatically guarantee that model outputs (disaggregated area and yield by farming system) is correct. I think, the advantages of the model relative to simpler methods are stated too bold throughout the Discussion paper. The authors' claims might be true, but need be tested in a standard way of model evaluation (e.g., by using the cross-validation technique).

Authors' Response: Thanks for the comment. We have examined the manuscript thoroughly and carefully avoided such self-judgment statements. In the revised manuscript, we only keep the explanations by citing literature on the inclusion of economic factors. For example, Market is important for both subsistence farmers and commercial ones. So many researchers have assumed that farmers are risk averse and profit maximizers (e.g. Hazell and Norton, 1986; Roundevell et al., 2003). See the discussion in Section 7.1 (Line 580). In addition, we elaborated more on the indicator of market access and admitted that the idea of including economic factors does not automatically guarantee that model outputs. We have revised the text along with other discussion on the limitations of SPAM in Sections 7.2 and 7.3.

*************************************************************************** Technical corrections
Comment#6. L71-73. I strongly suggest removing this description. Researchers would use the latest version once global agricultural dataset is updated, but no such update is

available to date. This is the reason why the studies cited here use an earlier version. The authors' criticism made here is inappropriate.

Authors' Response: We have rephrased the sentence to avoid any inappropriate criticism. Now the rational is more focused: an update of existing global agricultural production maps is very desirable. Please see Line 74. Thank you very much.

Comment#7. L107. The current text is a bit misleading. This text should read "M3 has no distinction across farming systems ..." or similar.

Authors' Response: Revised accordingly. See Line 107.

Comment#8. L156. Country crop-specific production costs for a specific year (e.g., 2011) are available via GTAP9 database (Aguiar et al. 2016). Just for your information.

Authors' Response: Thanks for the comment. After a careful consideration we choose to retain the use of FAO gross production value, because: 1) the GTAP provides data on 2004, 2007, 2011 and 2014, yet data on 2010 is not available; 2) the two data source is very close to each other, as GTAP database is constructed by referring to the FAO data; 3) the values between GTAP and FAO has no significant variance, for example, the rice production value in the US in 2014 is recorded as 2938 million from GTAP and 2973 million from FAO, respectively.

Comment#9. L158. GAEZ only provides "potential" crop suitability area. Please consider keeping precise terminology in the Discussion paper.

Authors' Response: Revised accordingly throughout the paper (8 places in total). Thanks for the kind reminder.

Comment#10. Eq. 7. What is "CE"? The abbreviation suddenly appears without definition. And I would appreciate it if the authors could provide a brief explanation what is the difference between {s ln s} versus {s ln $\pi$}.

Authors' Response: CE is the abbreviation for cross entropy. As entropy is defined as

the log function of probability, the difference between {s ln s} versus {s ln $\pi$} means the estimated probability s and its prior probability $\pi$ are minimized subject to certain constrains. The more detailed explanation is provided in Line 190.

Comment#11. Eq. 16. AdjCropY suddenly appears in main text although it is explained in Supplement. A brief explanation need be added in main text for readability.

Authors' Response: revised accordingly. See Line 253.

Comment#12. L304-305. Are the yield conversion factors in the text same with those shown in Table S6? Table S6 shows only for irrigated versus rainfed. Where is rainfed high input versus rainfed low input?

Authors' Response: Yes, indeed Table S6 shows part of the yield factors. In fact, Table S6 showed both the factor of crop yield under irrigated versus crop yield under rainfed (with a "I") and that of yield under rainfed high input versus yield under rainfed low input (with a "R"). See the Note under the table: "Production systems – irrigated (I) lists factor for irrigated vs. rainfed; rainfed (R) lists factor for rainfed high vs. rainfed low".

Comment#13. 372-373. This assumption is too crude. Dong et al. (2017) presents a nice global dataset in specifying urban areas. It can be useful to distinguish rural and urban areas more accurately.

Authors' Response: We have carefully considered the comment, then we find that the original text in the manuscript is misleading. We do not aim to distinguish rural area from urban area. The aim of introducing the variable AggRurPopi is to estimate the market accessibility and to account for subsistence production. We have further revised the text. Please see line 413.

Comment#14. L532. I do not understand "methodological-cum-data". Please consider rephrasing.

Authors' Response: It literally means the combination of method and data. We have revised the expression as "methodological-data" to avoid confusion. Please see line

584.

Comment#15. L626-627. This is true but has not been demonstrated yet. I would suggest removing this statement unless a comparison in area and yield for each farming system against subnational statistics is presented.

Authors' Response: The sentence has been removed accordingly. Please see Line 705.

16. L636. Zhang et al. (2017) reports the northward shift of paddy area in China and the westward shift of paddy area in India for the 2000-2010 period. These tendencies seem be inconsistent with the upper panel of Fig. 8.

Authors' Response: This is a misreading. The SPAM results are consistent to Zhang et al. (2017). According to the color schemes in Figure 13 (Figure 8 in the original submission), red means "increase" and blue means "decrease". The northeast part of China and northwest part of India are colored as red, suggesting a notable expansion of rice planting in these regions.

17. L679-680. Global roads and railways database used in Koks, E.E. et al. (2019) is maybe of your interest to more accurately define accessibility to markets. Just for your information.

Authors' Response: Thanks for the great suggestion. In the current SPAM, market accessibility is used to calculate the gross revenue of crop production which is then used to estimate a prior for the crop area (Equation (10) in the revised manuscript). As this crop-specific revenue is divided by the total revenue within a pixel (Eequation (11) and (12) in the revised manuscript), the prior is not affected by market accessibility if it is not crop-specific. In other words, crop-specific market accessibility is preferable for the current SPAM model. Such accessibility data doesn't exist now. We would consider modifying the role of market accessibility in the next version of SPAM model and then will probably use the global roads and railways database.
*************************************************************************** Reference Aguiar,A. et al. (2016) An Overview of the GTAP9 Data Base. Journal of Global Economic Analysis, 1,181–208, http://dx.doi.org/10.21642/JGEA.010103AF Dong, Y. et al. (2017) Global anthropogenic heat flux database with high spatial resolution. Atmospheric Environment, 150, 276-294, https://doi.org/10.1016/j.atmosenv.2016.11.040. Koks, E.E. et al. (2019) A global multi-hazard risk analysis of road and railway infrastructure assets. Nature Communications, 10, 2677, https://doi.org/10.1038/s41467-019-10442-3 Iizumi, T. et al. (2014), Historical changes in global yields. Global Ecology and Biogeography, 23, 346-357, doi:10.1111/geb.12120 Iizumi, T., Sakai, T. (2020) The global dataset of historical yields for major crops 1981–2016. Sci Data 7, 97, https://doi.org/10.1038/s41597-020-0433-7 Siebert, S. & Doll, P. (2010) Quantifying blue and green virtual water contents in global crop production as well as potential production losses without irrigation. Journal of Hydrology, 384, 198–217, https://doi.org/10.1016/j.jhydrol.2009.07.031 Sloat, L. L., et al. (2020) Climate adaptation by crop migration. Nature Communications, 11, 1243, https://doi.org/10.1038/s41467-020-15076-4 Zhang, G, et al. (2017) Spatiotemporal patterns of paddy rice croplands in China and India from 2000 to 2015, Science of The Total Environment, 579, 82-92, https://doi.org/10.1016/j.scitotenv.2016.10.223.

[Figure]

[Figure]

**Fig. 1.** Figure 6: Comparison between the allocated crop area and statistics crop area at the ADM2 level in Brazil (log-log scale plot, unit: ha.). The upper part is for SPAM2000 and the bottom part is for SPA

[Figure]

**Fig. 2.** Figure 7: Comparison between the allocated crop area and statistics crop area at the ADM2 level in Bangladesh, Benin, Senegal and Tanzania for maize, rice and cotton (log-log scale plot, unit: ha.).

[Figure]

**Fig. 3.** Figure 10: Grid-by-grid comparison between SPAM2010 and Zhang et al. (2017) rice area in China and India.

[Figure]

**Fig. 4.** Figure 11: Grid-by-grid comparison between SPAM2000 and Siebert and Doll (2010) in average irrigated and rainfed yields (log-log scale plot, unit: kg/ha.).

[Figure]

Maize

Wheat

Rice

Soybean

**Fig. 5.** Figure 12: Grid-by-grid comparison between SPAM2005 and EARTHSTAT2005 in crop yields. (log-log scale plot, unit: kg/ha.).

[Figure]

**Fig. 6.** Figure 14: Comparison between SPAM crop area change and CDL crop area change (log-log scale plot, unit: ha.).

[Figure]

**Fig. 7.** Figure 15: Comparison between SPAM rice area change and Zhang et al. (2017) paddy rice change (unit: ha.).

---

## Author Comment (AC2) · 20 Jun 2020

Dear Referee,

Thank you for the comments concerning our Discussion paper entitled "A cultivated planet in 2010: 2. the global gridded agricultural production maps" (Ref. essd-2020-11). These comments were very helpful for revising and improving our paper. To make the reply more readable, we list the comments and corresponding responses one by one in the Authors' Response (AC). The detailed revisions are embedded in the manuscript with the line numbers indicated in the AC.

*************************************************************************** General comments
This paper presents the latest update of the SPAM global gridded crop maps for 2010.

Overall, this is a very valuable effort. Yet, in its current form, I have several general remarks:

1/ The method is insufficiently explained and unclear in some places. A series of expert judgments are used along the way, and although this is acknowledged in the description of the methods, this seems insufficiently acknowledged in the Abstract and Introduction. Overall, this raises concerns about the transparency and reproducibility of the work, but also makes it very unclear what is the same and what is changed compared to previous versions, thereby justifying a new paper.

Authors' Response: Thanks for the general comment. You pointed out three problems, which indeed have not been clearly stated in the previous manuscript. We have carefully considered the suggestions and have made the changes accordingly. Please see the detailed responses, the revised manuscript, and the much-expanded Supplementary Information, through which we have improved: (1) the flow of method (comment#2-4 and #6) (2) the explanation on model parameter (comment#8) (3) the acknowledge of expert knowledge (comment#9) (4) and the differences between SPAM2010 and the previous products (comment#1 and #5). Please see the detailed responses below.

2/ Validation: This is a model (mixing reproducible rules and expert judgments), and as such, one would expect more rigorous and transparent validation efforts. Here it appears very thin and, in the words of the Authors themselves, the uncertainty assessment "is not a scientific, rigorous" one.

Authors' Response: Thanks for the comment. Indeed, validation is critical, and we have improved quite a lot in this revision, which include: (1) Cross-checking the national and subnational level statistics (comment#11) (2) Cross-checking with the paddy area maps in China and India (comment#12) (3) Cross-checking with EARTHSTAT (comment#12) (4) Cross-checking with Siebert and Doll (2010) (comment#12) (5) Providing more supporting evidence on the transparent validation process (comment#14-17) (6) and rephrasing the explanation and justification of the validation process
(comment#18-21) Please see the detailed responses below.

3/ Beyond operational uses for agencies focusing on crop production, the paper does not discuss how can these efforts serve more broadly scientific agendas regarding an improved understanding of the role of land management in global environmental change, earth system dynamics and other global sustainability issues (e.g., see Erb et al. 2016 in GCB for a discussion)? This would be useful to make the paper more valuable in itself beyond "just" presenting the dataset (no offence here, this is of course a great achievement!).

Authors' Response: Thanks for the very good comments. We have added a discussion in the final paragraph, which highlights the contribution of our dataset for better understanding land management in facing with the global change challenges. Please see Line 835.

I return to these main comments below. ************** Methods: ************** Comment#1. If this is an update with just purely the same methodology, it should not be an extra scientific paper. If there are substantial changes (improvements) in the methodology, then previous validations should not be taken for granted. Here, it is not totally clear what is new and should be validated, versus what is standard.

Authors' Response: Thanks for the comment. The model has been substantially improved comparing to the original version, i.e. SPAM2000. While SPAM2010 still keeps the cross-entropy approach, the notable changes/improvements include: (1) update the base year from 2000 to 2010 (2) double the crops included (from 20 crops to 42+ crops) and (3) apply the latest hybrid cropland input with an uncertainty associated with cropland estimate. Considering the huge amount of input data and multi-year effort, such an update is not trivial. As crop type maps change much more dramatically from year to year than, say, cropland map, such an update and improvement is critical for the user community. We agree previous validations should not be taken for granted. In the revised manuscript, we have added a lot of additional validation works which are

elaborated in detail below. The methodological and data improvements are highlighted in the Abstract and Lines 123-126.

Comment#2. Overall, the explanation of the method is unclear in many places. Being familiar with many of these gridded products, but not very much with the previous versions of SPAM in particular, I really have a hard time understanding the approach here. I am dubious that a reader that has not read the previous methods papers can understand what the Authors have done here.

Authors' Response: Thanks a lot. We have carefully considered the comment and revised the method section thoroughly. Firstly, we combined some of the descriptions on data into the description of method, see the updated description on the "disaggregation module". Secondly, we adjusted the flow of the optimization module by first introducing the optimization objective, followed by detailed introduction on each parameter. Please see the revised Section 3.

Comment#3. The method is insufficiently explained: 3.1: The 4 farming systems are explained, but not how the disaggregation between these 4 is done. The answer seems to be actually in Section 4.1.3, but here the answer is essentially "we do it, based on multiple information and stuff, trust us". Figure 2 is supposed to present an "illustration" ("We present an illustration for obtaining the farming system shares by crop j and administrative unit k (Percent jlk ) in Figure 2"), but Figure 2 doesn't give any information on how this disaggregation is done.

Authors' Response: Indeed, the explanation on "how the disaggregation is done" is presented in the "data preparation section" and some details are even presented in the supplementary information. We follow your suggestion by moving and reorganizing some of these content into section 3.1, which looks better and clearer. Please see Lines 156-183. Figure 2 is a conceptual framework that shows how these sub-modules are connected rather than a technique flowchart. We noticed that the text description of Figure 2 is misleading and then removed it accordingly.

Comment#4. Then Section 3.2 explains the optimization but honestly, I understand the equations but it doesn't allow me to understand the process itself.

Authors' Response: We have carefully considered the comment and we admit that the flow is a bit confusing. To make a clearer explanation, we have reorganized section 3.2 by firstly elaborating the objective of optimization, followed by the explanation on how the optimization is processed. Please see the detailed response to comment#2 and Lines 185-250 in the revised manuscript.

Comment#5. Section 3 does not clarify explicitly what methodological aspects are the same as in previous versions, versus those that have been modified or are new.

Authors' Response: Please see the detailed response to comment#1. We have now specified the differences between SPAM2010 and the previous SPAM models, which mainly include the update of base year, the expansion of sub-national administrative unit coverage, the extension of crop list, and the substitution of the latest hybrid cropland map as the basic allocation layer. Please see the Abstract and Lines 123-126 in the revised manuscript.

Comment#6. 4.1.3 Crop statistics disaggregated by farming systems: Âż This seems to be a mix of various approaches. Can you at least clarify the share of cropland disaggregation achieved based on statistics versus some expert knowledge or assumptions?

Authors' Response: This is relevant to comment#3. We have specified the process of disaggregation module in Section 3.1. For example, we extended the explanation on how statistics, existing data and reports, and expert knowledge are applied in disaggregation. Please see line 156-183. Now Section 4.1.3. is more focused on data preparation.

Comment#7. p.5: "The rainfed subsistence farming system (S), which is also low input as well, and is introduced to account for situations where cropland and suitable areas do not exist, but farmland is still present in some way." Âż This is very unclear.

[Figure]

Authors' Response: We have now revised the sentence as: "The rainfed subsistence farming system (S) is introduced to account for situations where cropland and suitable areas do not exist, but farmland is still present in some way. Production is mostly for own consumption, which is also low input as well." Please see Line 153-155.

Comment#8. Accessibility: This comes in Eq. 1 in Section 3.2, and then is detailed in 4.2.5, but I don't understand what is the rationale for creating / using an "accessibility" to market dataset based only on rural population? Is there an assumption that urban populations are fed from anywhere on the planet through global supply chains without this creating any particular incentive for farmers in surroundings (so that only rural population create a revenue incentive as per Section 4.2.6)? This would be quite a strong assumption. What is the rationale behind?

Authors' Response: We have carefully considered the comment, then we find that the original text in the manuscript is indeed misleading. In the current SPAM, market accessibility is used to calculate the gross revenue of crop production which is then used to estimate a prior for the crop area (Equation (10) in the revised manuscript). Yet, it does not mean the accessibility of getting food. Moreover, the aim of introducing the variable AggRurPopi is to estimate the market accessibility and to account for subsistence production, rather than aiming to distinguish rural area from urban area. As this crop-specific revenue is divided by the total revenue within a pixel in equation (3), the prior is not affected by market accessibility if it is not crop-specific. In other words, crop-specific market accessibility is preferable for the current SPAM model. Such accessibility doesn't exist now. We have further revised the text too make a clearer introduction of this variable. Please see Lines 419-425.

Comment#9. Overall, there is a lot of expert judgment and wiggling with the data (see S4, S1315 etc) (e.g., Section 4.3.1 "Under these circumstances, we set several options to "force" a solution, including adjusting the entropy conditions, and adjusting the data harmonization rules. We elaborate on the details for adjusting areas (Section S13), entropy conditions (SectionS14) and harmonization rules (SectionS15) respectively in

the SI."). So this is far from resulting from a clean and reproducible algorithm based on simple economic rules. I don't want to distrust the work done by the Authors, but given this large amount of expert-driven decisions, this should be very clearly stated in the abstract and main results / Conclusion, so that the reader understands clearly that this is largely an expert-driven process, with multiple human decisions and assumptions, more than a simply reproducible algorithmic work that produces a transparent output.

Authors' Response: Thanks for the comment. First we want to say that such "expert judgment and wiggling" is quite rare and is small part of the overall cases running of the model (less than 1%). These small cases happen to those difficult countries such as Somali and Nigeria where reliable data is not available or different input data just conflict each other. For example, only one crop (i.e. millet) area for a district is already larger than the total cropland area, yet we know there are still five more crops growing in this district. In these cases, we have to adjust the conflicting data, using expert judgment, to make the model solvable. Second, we have made every effort to collect official or published data and we only reply on expert judgments as the last resort when we simply could not find other sources. For example, no country publishes official statistics on crop yield ratio (yield conversion factor) between irrigated vs rainfed crop (e.g. rice). We surveyed published papers, personal communication with FAO's Agriculture to 2030 team, and gray literature to collect such data. While indeed a series of expert judgments are used, the scope (e.g. crops and regions) is quite limited in the overall input data. We have a long documentation of such instances in the supplementary information (SI) file. Following your advice, we have included more discussion on the application of expert knowledge in Section 7.3 (Lines 770-785). In addition, we provided more supporting evidence on how expert judgments were applied for validating our data product. Please see Section 7.1 (Lines 595-610) and the newly-added Section S16 in the SI file, The SI file is much enhanced and expanded, taking advantage of no limit on the length of the SI file. The SPAM model is a reproducible work despite that it occasionally relies on the expert judgment to get a solution. In fact. we are building a SPAM model on the cloud where we let any user to supply his/her

own input data and run SPAM on his/her own under the Github platform. This SPAM on the cloud will be published and communicated to SPAM user community once it is ready. Please see the discussion in Section 7.2 (Lines 705-710).

*************** Validation: *************** Comment#10. Same as above and general comment: First, this is a model; and thus it should be validated properly as far as possible. I understand of course that by the nature of the work done, there is no simple, global, adequate validation data ready to be used. But still, (i) there are ways to do more & better, and (ii) the current efforts are reported in an unclear manner.

Authors' Response: Thanks for the comment. We have now enhanced the validation works, which include: (1) Cross-checking the national and subnational level statistics (comment#11) (2) Cross-checking with the paddy area maps in China and India (comment#12) Moreover, we have reorganized the entire section, in particular, we have rewritten the qualitative validation part (comment#14-17) and rephrased the explanation and justification of the validation process (comment#18-21) to make the description clearer. Please see the detailed responses elaborated below and the revised text in the entire Sections 7.1 and 7.2.

Comment#11. If, as you explain, you run most countries with data at ADM0 level, but you do have incomplete data at finer administrative levels, then you can at least validate against these incomplete subnational data. This is explored in Figure 5 but given the breadth of the map, just one example is not sufficient.

Authors' Response: Thanks for the constructive comment. Actually, the validation by cross-checking national and subnational level statistics has been applied for SPAM2000 (e.g. Brazil). Following your comment, we have re-applied the approach for the current SPAM2010 for a few selected countries such as Brazil, Bangladesh, Benin, Senegal, Tanzania. We find that the performance has generally improved comparing to the performance of SPAM2000 though this varies from country to country, and from crop to crop. We add Figure 6 and Figure 7 and the relevant description of the validation process in the revised manuscript. We believe this newly added comparison, along with additional validation works (as described in the response to comment#12), expands the breadth of validation and thus substantially improve the reliability of the SPAM2010 product.

Comment#12. Partial validation could also be achieved through a sampling of points, with visual interpretation of high-resolution imagery to at least identify irrigated systems versus non-irrigated intermediate categories versus the subsistence category. Even some specific crops could be assessed, at least some perennial crops like oil palm, banana, or others.

Authors' Response: Thanks for the very good comment. As suggested, we have undergone three additional analysis: (1) Zhang et al. (2017) have provided annual paddy area time series maps from 2000 to 2010 based on satellite remote sensing for China and India. We have compared these remote-sensing derived paddy maps with the rice area estimated by SPAM2010. In addition, we compare the $\triangle$Rice (difference between the rice map in 2005 and 2010) between $\triangle$SPAMrice (difference between SPAM2005 and 2010). (Figure 10 and Figure 15) (2) We collect the "Harvested Area and Yield for 4 Crops (1995-2005)" from an independent dataset at: http://www.earthstat.org/. Then we compare the yields for specific farming systems of SPAM2005 by referring to EARTHSTAT_2005. (Figure 12) (3) We collect the average irrigated and rainfed yields for the 1998–2002 period at the global scale (Siebert and Doll, 2010). Then we compare the irrigated and rainfed yields between SPAM2000 and Siebert and Doll (2010). (Figure 11) In one of our previous papers, we have compared the difference farming systems based on the global datasets around 2000 (Anderson et al., 2015). We find and admit that some of these comparison results can not be used directly to support the latest SPAM2010. Unfortunately, we do not find any global maps (e.g. farming system, perennial crops) available for the year 2010. However, these comparisons will provide implications to integrate tools and standardize approaches across various ongoing projects that develop gridded information on land-use dynamics for applications

in food security, climate change, biodiversity, and other related issue area. There is an ongoing consortium called The Land Use Change Knowledge Integration Network (LUCKiNet, www.luckinet.org), which aims at this integration, and SPAM team is part of this consortium. Not only LUCKiNet aims to create crop maps comparable over time, we also want to have these maps consistent across land uses such as cropland, grassland, forest. The modelling techniques would consider the spatiotemporal dynamics of different land use forms in an integrative framework. We have revised the manuscript by adding these additional works along with more clarifications to support the current SPAM2010 products. Please see the revisions in the entire Section 7, including Figure 10, 11, 12, 14, 15.

Comment#13. You can't just say (l.539): "As the coverage, quality and spatial precision of data input are much better for SPAM2010 than for its predecessors (see Section 4), the reliability of the data product is believed to improve as well."

Authors' Response: We agree that the existing description is inappropriate, therefore we have removed the statement accordingly. Now it is more objective and could be partly supported by the comparison between national and subnational statistics. This cross-validation work has been newly added in the revised manuscript as well. Please see Section 7.1.

Comment#14. l.548: "Firstly, we evaluate the results by sending the crop maps to collaborators and users alike for comments or assessment. For example, the CGIAR..." Âż I don't understand how this is an "example". Either you did it and you report the results, or you explicitly state that this is something that you have not done but could do.

Authors' Response: Thanks for the comment. we did the assessments and we have the reports. These reports were collected from collaborators and users alike, mostly crop by crop, and country by country. We admit that the statement is not clear enough. We have revised the text and provided more supporting evidence in the Supplementary

Information (SI).

Comment#15. "We took advantage of their vast network of field offices and local expertise to help us to validate the SPAM results. Many researchers from these institutes have been involved in the production of SPAM2010, which increases the reliability of the results." Âż If this has been done, then you should report in more details the outcome of this process, the validation data collected...

Authors' Response: This is a similar comment above. We did such validation. Please see the revisions in Section 7.1 and more supporting evidence in the SI.

Comment#16. "The validating information could either be collected by" Âż "Could be", or it has been done? If the former, then it's not useful. If the latter, then provide the results.

Authors' Response: This is a similar comment above. We did such a validation, and we have changed the tense accordingly.

Comment#17. "We take these feedbacks and re-run SPAM model and release updated versions of SPAM. The complete validation process could take a great deal of effort and time, but these users' feedbacks are quite important and valuable." Âż Same, not clear, is this something you plan to do, or something you have done and can provide data about? The use of present tense makes it confusing.

Authors' Response: This is a similar comment above. We did such a validation, and we have changed the tense accordingly.

Comment#18. "The current product, i.e., SPAM2010v1.1, is also expected to have major updates" Âż Then is it the right time to release it? Wouldn't it be better to have this round of validation – improvement first?

Authors' Response: This statement is indeed confusing. We mean that when additional information is available, SPAM is open and ready to update. Nevertheless, the current version has been validated extensively so far and therefore could be released. We

have revised the text accordingly. Please see Line 613.

Comment#19. "Secondly we do a regional validation in case that the third-party independent crop maps are available," Âż Same, present time: Does that mean you have done it? Or does that mean this is an aspirational goal that at some point you hope you can do it? Here, as you provide the comparison with US data in Figure 5, it appears that this is something that you have actually done. But (i) we have to guess it, and (ii) it's not clear for all the above.

Authors' Response: This has actually been done. We have corrected all these to avoid confusions. Please see the revisions in Lines 657-683. Thanks a lot!

Comment#20. Figure 8: Differences are huge. I understand that this mixes real changes on the ground and changes in the methods. But over - nominally - 5 years, this appears to be predominantly dues to changes in the methods. Please elaborate further (note, this is in relation to the above point on Methods, as it is not fully clear what is stable and what has changed in the Methods).

Authors' Response: Thanks for the comment. We have noticed that in some cases the changes are huge. However, the overall pattern is acceptable. In fact, it is inappropriate to compare SPAM products across time stages. Because the changes not only mix real changes on the ground and changes in the methods, but also largely depend on the input data such as statistics and cropland layer. This is inevitable as we should not apply the cropland layer in 2005 for SPAM2010. However, we do not evaluate the continuity of this input data, which is almost impossible and is beyond the purpose of SPAM. Therefore, it is suggested to use the SPAM products with acknowledgement to the corresponding cropland layer. We have submitted the cropland layer dataset as a sister paper to support the current paper, please refer to Lu et al, (A cultivated planet in 2010: 1.). This problem exists in other gridded land use datasets as well. As we responded to comment#12, there is an ongoing consortium, LUCKiNet, which aims to integrate tools and standardize approaches across various existing products,

and the SPAM team is part of this consortium. We hope the problems of systematic inconsistency across datasets will be quantified through large amount of integrative efforts under the consortium. We added this explanation in the revised manuscript. Please see Sections 7.2 (Lines 730-760) and 7.3 (Lines 811-823).

Comment#21. l.604: "In addition, we collect feedback and comments from users, local experts and collaborators as discussed above. They are sporadic but very useful. We combine all the information together to give a subjective rating on how confidence we, SPAM team, think of our final crop maps (both area and yield). This is the uncertainty rating we provided here. It is not a scientific, rigorous rating and so we put it only into 1 to 5 categories (1 represents the lowest uncertainty, 5 the highest)." Âż If this is not a "scientific" rating does it belong to a "scientific" paper?

Authors' Response: Thanks for the comment. The subjective rating was just one example among many validating works. We admit that it is not vigorous, but the result is convincing and such a rating is highly demanded and explicitly requested by users. Therefore, we believe it is appropriate to add this result into the main text. In the revised manuscript, we have carefully explained how the uncertainty rating was performed and why this is useful. Please see Lines 616-629.

************** Minor comments: ************** Comment#22. Abstract: I don't understand this sentence: "but also dedicates as platform providing archived global agricultural production maps for better targeting the Sustainable Development Goals by making proper agricultural and rural development policies and investments"

Authors' Response: This is now revised as: "but also dedicates as platform providing archived global agricultural production maps for better targeting the Sustainable Development Goals." Please see Line 23.

Comment#23. Overall the writing is good, but there's a series of weird words, typos and stuff like l. 363: "protected areas. But if the "or l.371" rural population density"(just to give examples, there's plenty of these). Please triple-check through.

Authors' Response: Thanks for the comment. The sentence has been revised as: "During the initial allocation process SPAM allows for crop allocation in protected areas to allow for this reality, but if the model does not solve, one option is to increase the area designated as cropland, suitable land or irrigated land." (Line 408) and the word "pulation" (Line 416) has been corrected. Moreover, we have triple-checked the language and corrected a few minor mistakes. The revised manuscript has been proof-read by all coauthors before resubmission.

[Figure]

**Fig. 1.** Figure 6: Comparison between the allocated crop area and statistics crop area at the ADM2 level in Brazil (log-log scale plot, unit: ha.). The upper part is for SPAM2000 and the bottom part is for SPA

[Figure]

**Fig. 2.** Figure 7: Comparison between the allocated crop area and statistics crop area at the ADM2 level in Bangladesh, Benin, Senegal and Tanzania for maize, rice and cotton (log-log scale plot, unit: ha.).

[Figure]

**Fig. 3.** Figure 10: Grid-by-grid comparison between SPAM2010 and Zhang et al. (2017) rice area in China and India.

[Figure]

**Fig. 4.** Figure 11: Grid-by-grid comparison between SPAM2000 and Siebert and Doll (2010) in average irrigated and rainfed yields (log-log scale plot, unit: kg/ha.).

Maize

R²= 0.65

Wheat

R²= 0.56

Rice

R²= 0.54

Soybean

R²= 0.43

**Fig. 5.** Figure 12: Grid-by-grid comparison between SPAM2005 and EARTHSTAT2005 in crop yields. (log-log scale plot, unit: kg/ha.).

[Figure]

**Fig. 6.** Figure 14: Comparison between SPAM crop area change and CDL crop area change (log-log scale plot, unit: ha.).

[Figure]

**Fig. 7.** Figure 15: Comparison between SPAM rice area change and Zhang et al. (2017) paddy rice change (unit: ha.).

---

## Referee Report (RR1)

General comments

I appreciate the authors' efforts devoted for this revision (and thank you for listing my comments one by one, which are more readable. There was a technical barrier for me to do so). The methodological details for the SPAM2010 are much clearer than the previous manuscript. The authors' responses regarding the validation of SPAM2010 are sufficiently strong to justify the publication of the paper, although I comment on the way of presentation in this review report. Nevertheless, some methodological details remain unclear and their clarifications are still needed. I should mention all my questions at the first round of review (sorry for this), but some descriptions added in the revision require further clarifications. I pointed out several concerns below, which are relatively major but not substantively major.

Relatively major concerns

1. L67-71. This is misleading and needs edit carefully. I understood that SPAM2010 product offers global crop production maps for 2010, which has the latest base year compared to other products, such as M3 and MIRCA for 2000. However, as the authors discuss in L811-816, it is inappropriate to compare SPAM2010 with its predecessors (SPAM2000 and SPAM2005) to explore historical change in area, yield and production. I admire the authors' honesty to show this result and agree with this statement. Therefore, it is misleading to emphasize the importance of area change over time because SPAM products cannot be used for this purpose.

2. L161. It is still unclear which production and area you intend to refer when you state "share". I found some explanations for this distinction in Supplementary Information (SI) Section 4, but a clearer distinction between production share and area share throughout main text is needed. This comment is applied throughout the manuscript, for instance, L334, L347, Table 2 (e.g., Percentjlk) and Table S5 (the title).

3. L323-324. Although FAO country statistics are the most reliable source of data for global agriculture, these include many data values from unofficial sources or estimation. This is especially true for developing countries. This arise a question whether adjusting national and subnational statistics against FAO data is always valid. I don't request the authors recalculating SPAM2010 because this type of uncertainty is widely observed not only FAO data versus subnational statistics but also different global yield maps (Anderson et al. 2015 (this is already cited); Fig.9 of Müller et al. 2017; Schauberger et al. 2017, Iizumi et al. 2018). However, the authors are encouraged to explain a bit more (in addition to L323-324) why FAO data are used as the baseline in the adjustment of country statistics despite the possible uncertainties in FAO data.

4. For Figs. 6, 8, 9,10, 11, 12, 14 and 15 presenting grid-cell level comparisons between SPAM2010 and other products, I would strongly suggest the authors using the density scatter plot instead of the ordinary scatter plot to improve the visibility of the agreement between two data sources. For instance, see Scatter plots with rectangular bins at http://www.sthda.com/english/wiki/ggplot2-scatter-plots-quick-start-guide-r-software-and-data-visualization for R script. And some of these figures show RMSE (e.g., Figs. 8, 9 and 10) but others don't. For consistency and informativeness, I ask the authors adding RMSE values for all of these scatter plots.

Technical corrections
5. L48. "geo-political boundaries". I think, "administrative units" would be more suitable to be consistent with the remaining portion of the manuscript.
6. L58. "the harvested area and yield" should read "the potential harvested area and yield" as GAEZ estimates potential levels but not actual ones.
7. L119. It would be nice if a brief introduction of SPAM2005 (e.g., "SPAM2005 expands its coverage to xx crops with the updated base year of 2005" or similar) could be added here to give readers a short history on the improvements to SPAM products.
8. L131-132. This is a bit hard to understand when I read for the first time. Crop statistics are disaggregated to what? Perhaps, you want to say hear that national-level statistics are disaggregated into sub-national level; statistics for crop aggregates are divided into individual crop types; and area, production and yield statistics are separated for each of rainfed and irrigated conditions (rainfed conditions are further disaggregated into input levels – high, low and subsistence).
9. L179. "the statistical demand of a crop". I don't understand this term. Please consider rephrasing. I suspect that the authors intend to indicate a measure of the completeness in disaggregation in terms of area extent or production quantity. The same comment is applied to SI Section 4.
10. L352. I realized that there are two types of yield conversion factor from Response to Reviewer Comments, (i) rainfed yield to irrigated yield and (ii) rainfed low-input yield to rainfed high-input yield. This information is important but currently lacking in main text. Please consider adding it to main text around here.
11. L386. "optimal potential" can simply read "potential".
12. L395-401. Why did you use GMIA to derive information on irrigation equipped area? MIRCA can provide crop-specific irrigated area, whereas GMIA cannot (but MIRCA is for 2000). Indeed, MIRCA also used for some countries to collect irrigation information, as described in Table S4. And HID (Siebert et al. 2015) also provide global irrigation area for 2005 and that for 2010 is likely estimated using historical trends. I would appreciate if the authors' underlying thought on the use of GMIA could be explained briefly here.
13. L480-490. Here the procedure is hard to follow. To me, it seems that yield and production at different scales are adjusted several times (maybe my understanding is incorrect). Can you edit a bit more to increase readability? More importantly, it seems that yield at statistical reporting unit is used as the first guess of grid-cell yields. However, yields at finer scales (e.g., farm field level) could largely differ from subnational statistical yield and there is a space for better modeling (Gerlt et al. 2014, Porth et al. 2017). Therefore, I'd like to ask the authors to add a brief justification and possible limitations of the current method here (or elsewhere in Discussion).
14. L511. "high in than region". Probably, this is a typo.
15. L537. "I\$". Do you mean international dollar? If yes, please explicitly state so. What is the base year?
16. L577. "predicting crop areas". In general, the term "prediction" is used to derive a value of a variable of interest in the time t using inputs at the time t-1. Here, inputs at the time t are used to derive value of crop area at the time t. In this case, "estimation" is an appropriate term.

17. L667. "231 and 307". What is the unit? ha? The same comment is applied to RMSE value in L672.
18. L668. Why is the agreement for wheat in the US between SPAM2010 and CDL worse than that for maize and soybean? This is interesting because the result suggests that a key factor is likely lacking in wheat modeling.
19. L749. "the correlation". This should read "the coefficient of determination".
20. L813. "Izumi et al. (2020)". Do you mean "Iizumi and Sakai (2020)"?
21. Table S2. What is the difference between zero (e.g., ADM2 harvested area for Afghanistan) and – (e.g., ADM2 harvested area for Albania)? Can you add a brief explanation in footnote of the table? And the difference between #N/A (Liechtenstein) and – is unclear as well.
22. The third paragraph, SI Section 4. Which "harvested area shares for different production systems" and "production quantity shares sourced from different production systems" does "production system shares" indicate?
23. Table S5. Please be specific whether the values indicate either production share or area share.
24. Table S7. I feel difficulty in interpreting the cropping intensities. If I take double rice cropping in monsoonal Asia (India, Indonesia, Philippines etc.) as the example, it operates rainfed condition in wet season and irrigated condition in dry season (e.g., Koide et al. (2013) for Philippines). Therefore, in my interpretation, the sum of cropping intensities over rainfed and irrigated condition can exceed one. However, the cropping intensity values exceed one for both rainfed and irrigated conditions (for instance, cereals in Indonesia). Can you explain a bit more what is the definition and how you calculate these values?
25. SI Section 16. In the figures in this section, what is the area in gray for SPAM harvested area maps?
26. Thailand, SI Section 16. "F on 12" should read "F on 13", isn't it?

References
- Gerlt, S. et al. 2014. Exploiting the relationship between farm-level yields and county-level yields for applied analysis. J Agric Resour Econ. 39, 253–270.
- Iizumi, T. et al., 2018. Uncertainties of potentials and recent changes in global yields of major crops resulting from census- and satellite-based yield datasets at multiple resolutions. PLoS ONE 13(9): e0203809. https://doi.org/10.1371/journal.pone.0203809
- Koide, N. et al., 2013. Prediction of Rice Production in the Philippines Using Seasonal Climate Forecasts. J. Appl. Meteor. Climatol., 52, 552–569, https://doi.org/10.1175/JAMC-D-11-0254.1
- Müller, C. et al., 2017. Global gridded crop model evaluation: benchmarking, skills, deficiencies and implications, Geosci. Model Dev., 10, 1403–1422, https://doi.org/10.5194/gmd-10-1403-2017,
- Porth, L. et al., 2017. Farm-level crop yield forecasting in the absence of farm-level data. https://www.soa.org/research-reports/2016/2016-farm-level-forecasting
- Schauberger, B. et al. 2017. Global evaluation of a semiempirical model for yield anomalies and application to within‐season yield forecasting. Glob Change Biol. 23, 4750–4764. https://doi.org/10.1111/gcb.13738

- Siebert, S. at al., 2015. A global data set of the extent of irrigated land from 1900 to 2005, Hydrol. Earth Syst. Sci., 19, 1521–1545, https://doi.org/10.5194/hess-19-1521-2015

---

## Author Response (AR2)

Dear Editor,

We appreciated your extra time and the Referees' additional comments concerning the revised manuscript entitled "A cultivated planet in 2010: 2. the global gridded agricultural production maps" (Ref. essd-2020-11). These comments were very helpful and instructive for revising and improving our paper. To make the reply more readable, we list the comments and corresponding responses one by one in the following Authors' Response. In particular, we made more detailed elaborations on the assumption related to market accessibility and rural population. The detailed revisions are embedded in the manuscript as well as in the SI (marked in red) with the line numbers indicated in the response.

Again, thanks a lot for handling our paper.

Bets regards,
Qiangyi Yu
On behalf of the coauthors

**Authors' Responses to Referee 1:**

**General comments**

**I appreciate the authors' efforts devoted for this revision (and thank you for listing my comments one by one, which are more readable. There was a technical barrier for me to do so). The methodological details for the SPAM2010 are much clearer than the previous manuscript. The authors' responses regarding the validation of SPAM2010 are sufficiently strong to justify the publication of the paper, although I comment on the way of presentation in this review report. Nevertheless, some methodological details remain unclear and their clarifications are still needed. I should mention all my questions at the first round of review (sorry for this), but some descriptions added in the revision require further clarifications. I pointed out several concerns below, which are relatively major but not substantively major.**

**Authors' Response**: Thanks for the very positive general comment on the first-round revision of the paper. Indeed we spent quite some time and effort to do additional analyses and revise the paper, and even requested additional time for the revision. Thanks a lot for pointing out a few technical mistakes which have been overlooked by us. We have addressed these comments carefully and made revisions accordingly.

**Relatively major concerns**

**1. L67-71. This is misleading and needs edit carefully. I understood that SPAM2010 product offers global crop production maps for 2010, which has the latest base year compared to other products, such as M3 and MIRCA for 2000. However, as the authors discuss in L811-816, it is inappropriate to compare SPAM2010 with its predecessors (SPAM2000 and SPAM2005) to explore historical change in area, yield and production. I admire the authors' honesty to show this result and agree with this statement. Therefore, it is misleading to emphasize the importance of area change over time because SPAM products cannot be used for this purpose.**

**Authors' Response**: We agree that the original statement in L67-71 has overstated the historical changes. We revised the statement to focus more on the importance of data update, rather than detecting the changes. See Lines 70-74. Here our main point is to emphasize the necessity and value of updating the global maps to a later year. Our logical flow on that issue is like this: 1. Agricultural production system is constantly changing, and these changes are not trivial; 2. We update the SPAM dataset to capture the latest status of agricultural production system; 3. Although we have produced the latest version, yet it is inappropriate to explore historical changes as we discussed in the Discussion section.

**2. L161. It is still unclear which production and area you intend to refer when you state "share". I found some explanations for this distinction in Supplementary Information (SI) Section 4, but a clearer distinction between production share and area share throughout main text is needed. This comment is applied throughout the manuscript, for instance, L334, L347, Table 2 (e.g., Percentjlk) and Table S5 (the title).**

**Authors' Response**:   Thanks a lot for that. The term "production" in "production share" does not mean "total production", which is indeed confusing when aligning with the term of "total production" of crop statistics. We have area share and yield ratio by different farming system (e.g. irrigated vs rainfed), and from there we break down the production by farming system (production = area * yield). "Production share" simply means "farming system share", which is further applied to disaggregate "total production" and "harvested area". We have unified the terminology "farming system share" throughout the manuscript (e.g. Lines 160-170), including the SI to avoid this confusion.

**3. L323-324. Although FAO country statistics are the most reliable source of data for global agriculture, these include many data values from unofficial sources or estimation. This is especially true for developing countries. This arise a question whether adjusting national and subnational statistics against FAO data is always valid. I don't request the authors recalculating SPAM2010 because this type of uncertainty is widely observed not only FAO data versus subnational statistics but also different global yield maps (Anderson et al. 2015 (this is already cited); Fig.9 of Müller et al. 2017; Schauberger et al. 2017, Iizumi et al. 2018). However, the authors are encouraged to explain a bit more (in addition to L323-324) why FAO data are used as the baseline in the adjustment of country statistics despite the possible uncertainties in FAO data.**

**Authors' Response**: We totally agree with your assessment on FAO country data. In fact, the SPAM team, over the years, has had long debates on the adjustment of our results to FAO data. In the end we decided to be consistent with FAO country totals for these few reasons: (1). As FAO data is the most widely acknowledged global agricultural statistics, it is the most appropriate source for the purpose. And we simply have no other comparable sources (2) SPAM products have been used by many global models such as IFPRI's IMPACT (https://www.ifpri.org/project/ifpri-impact-model), IIASA's GLOBIOM (https://iiasa.ac.at/web/home/research/GLOBIOM/GLOBIOM.html). These models use FAO country data for cross-country comparisons and they need our maps to be consistent with FAO data. In fact, the idea of conceptualizing SPAM is to spatially allocate statistics from administrative units to spatial grids, and the maps could be easily adjusted to any other country data. We have added this discussion in the revised manuscript. See Lines 330-337.

**4. For Figs. 6, 8, 9,10, 11, 12, 14 and 15 presenting grid-cell level comparisons between SPAM2010 and other products, I would strongly suggest the authors using the density scatter plot instead of the ordinary scatter plot to improve the visibility of the agreement between two data sources. For instance, see Scatter plots with rectangular bins at http://www.sthda.com/english/wiki/ggplot2-scatter-plots-quick-start-guide-rsoftware-and-data-visualization for R script. And some of these figures show RMSE (e.g., Figs. 8, 9 and 10) but others don't. For consistency and informativeness, I ask the authors adding RMSE values for all of these scatter plots.**

**Authors' Response:** Thanks for the comment and excellent advice. As suggested, we have replaced all ordinary scatter plots by density scatter plots, and added the corresponding RMSE in all plot figures. Indeed the density scatter plots look much better. Thank you very much!

**Technical corrections**

**5. L48. "geo-political boundaries". I think, "administrative units" would be more suitable to be consistent with the remaining portion of the manuscript.**

**Authors' Response:** Thanks for the comment. We have replaced all imprecisely used expressions accordingly. Please see Lines 48, 284, and 605.

**6. L58. "the harvested area and yield" should read "the potential harvested area and yield" as GAEZ estimates potential levels but not actual ones.**

**Authors' Response:** Revised accordingly. Please see Line 58.

**7. L119. It would be nice if a brief introduction of SPAM2005 (e.g., "SPAM2005 expands its coverage to xx crops with the updated base year of 2005" or similar) could be added here to give readers a short history on the improvements to SPAM products.**

**Authors' Response:** Following the comment we have added a statement: "SPAM2005 acts as an intermediate update which expands the coverage of crops from SPAM2000. The 42 crop categories are further adopted in SPAM2010 (Figure 1)." See Lines 120-121.

**8. L131-132. This is a bit hard to understand when I read for the first time. Crop statistics are disaggregated to what? Perhaps, you want to say hear that national-level statistics are disaggregated into sub-national level; statistics for crop aggregates are divided into individual crop types; and area, production and yield statistics are separated for each of rainfed and irrigated conditions (rainfed conditions are further disaggregated into input levels – high, low and subsistence).**

**Authors' Response:** Thanks for the comment. We have improved the description by clarifying what has been disaggregated. Please see: "The first step for SPAM is to disaggregate crop statistics of agricultural production (e.g. the yield, harvested area, and total production) by administrative unit levels ($k$), crop type ($j$), and farming system ($l$) from coarser scale to finer scale. For example, the national-level statistics are disaggregated into sub-national levels, statistics for crop aggregates are divided into individual crop types, and the crop statistics are further separated by rainfed and irrigated conditions.". See Lines 132-136.

**9. L179. "the statistical demand of a crop". I don't understand this term. Please consider rephrasing. I suspect that the authors intend to indicate a measure of the completeness in disaggregation in terms of area extent or production quantity. The same comment is applied to SI Section 4.**

**Authors' Response:** Yes it exactly means the "completeness" of disaggregated crop statistics in terms of area extent and/or production quantity. We have revised the expression accordingly. See Line 184.

**10. L352. I realized that there are two types of yield conversion factor from Response to Reviewer Comments, (i) rainfed yield to irrigated yield and (ii) rainfed low-input yield to rainfed high-input yield. This information is important but currently lacking in main text. Please consider adding it to main text around here.**

Authors' Response: Added accordingly. See Lines 360-364.

**11. L386. "optimal potential" can simply read "potential".**

Authors' Response: Simplified accordingly. See Lines 399.

**12. L395-401. Why did you use GMIA to derive information on irrigation equipped area? MIRCA can provide crop-specific irrigated area, whereas GMIA cannot (but MIRCA is for 2000). Indeed, MIRCA also used for some countries to collect irrigation information, as described in Table S4. And HID (Siebert et al. 2015) also provide global irrigation area for 2005 and that for 2010 is likely estimated using historical trends. I would appreciate if the authors' underlying thought on the use of GMIA could be explained briefly here.**

Authors' Response: In fact the SPAM team has been collaborating and discussing with MIRCA team for a long time. Beyond the reason that MIRCA is for 2000 (e.g. too old) as you pointed out, SPAM modelling technique is very different from MIRCA's and we don't want to bring their modelling errors into SPAM. Instead, we used GMIA (MIRCA also used it) and derived some of irrigation input parameters from MIRCA as you rightly pointed out. Anderson et al (2015) compared MIRCA, SPAM and M3 results and had a good discussion on that. We have added this clarification in the SI. Please see Section S9.

**13. L480-490. Here the procedure is hard to follow. To me, it seems that yield and production at different scales are adjusted several times (maybe my understanding is incorrect). Can you edit a bit more to increase readability? More importantly, it seems that yield at statistical reporting unit is used as the first guess of grid-cell yields. However, yields at finer scales (e.g., farm field level) could largely differ from subnational statistical yield and there is a space for better modeling (Gerlt et al. 2014, Porth et al. 2017). Therefore, I'd like to ask the authors to add a brief justification and possible limitations of the current method here (or elsewhere in Discussion).**

Authors' Response: Thanks. The grid-cell yield is calculated from the reported yield at statistical reporting unit, the allocated area from model results and the potential yield (at grid-cell level) from GAEZ (global Agroecological Zone). These are illustrated in Equations 15,16. These equations basically says that: The spatial variation of yield within a statistical reporting unit follows the same spatial variation of the potential yield of that crop. In other words, the more suitable (higher potential yield) cells would have a relatively higher yield while the average yield of all the gridcells would be equal to the statistically reported yield of the administrative unit. We have clarified that in the text. Please see Lines 504-508.

**14. L511. "high in than region". Probably, this is a typo.**

**Authors' Response:** Thanks for the comment. It is indeed a typo and has been corrected as "high in this region". See Line 529.

**15. L537. "I\$". Do you mean international dollar? If yes, please explicitly state so. What is the base year?**

**Authors' Response:** Yes, I\$ is the abbreviation for international dollar. We have added the full expression before using the abbreviation. And the average 2009-2010 base year price is applied. See Line 555.

**16. L577. "predicting crop areas". In general, the term "prediction" is used to derive a value of a variable of interest in the time t using inputs at the time t-1. Here, inputs at the time t are used to derive value of crop area at the time t. In this case, "estimation" is an appropriate term.**

**Authors' Response:** Thanks for the comment. Revised accordingly. Please see Line 596.

**17. L667. "231 and 307". What is the unit? ha? The same comment is applied to RMSE value in L672.**

**Authors' Response:** The unit (ha.) has been added accordingly. Please see Lines 686 and 694.

**18. L668. Why is the agreement for wheat in the US between SPAM2010 and CDL worse than that for maize and soybean? This is interesting because the result suggests that a key factor is likely lacking in wheat modeling.**

**Authors' Response:** There are potentially many factors affecting the different results if we treat CDL as the truth, for example, the different accuracy or availability of input data, suitability layers and parameters for the area shares and yield ratios. Another possible reason is that we did not distinguish spring wheat and winter wheat in SPAM. Indeed this is a very interesting question, and may be worth further investigation. But this is beyond the scope of this paper. Please see revisions in Lines 687-690.

**19. L749. "the correlation". This should read "the coefficient of determination".**

**Authors' Response:** Thanks for the comment. Revised accordingly. Please see Line 772.

**20. L813. "Izumi et al. (2020)". Do you mean "Iizumi and Sakai (2020)"?**

**Authors' Response:** Yes. Sorry for the mistake. Revised accordingly. Please see Line 847.

**21. Table S2. What is the difference between zero (e.g., ADM2 harvested area for Afghanistan)**

**and – (e.g., ADM2 harvested area for Albania)? Can you add a brief explanation in footnote of the table? And the difference between #N/A (Liechtenstein) and – is unclear as well.**

**Authors' Response:** Thanks for point this out. Zero means no production of that crop in that administrative unit while "-" or "N/A" means no data available (so it doesn't mean it doesn't produce this crop. Just we don't know). In the model, we convert this "N/A" into "-999" while zero is "0" – a real number. We have replaced all "#N/A" by "-" to keep the consistency. Please see Table S2 in the revised SI.

**22. The third paragraph, SI Section 4. Which "harvested area shares for different production systems" and "production quantity shares sourced from different production systems" does "production system shares" indicate?**

**Authors' Response:** This refers to area shares of different farming systems. The production in different farming system is calculated by considering the yield difference of the different farming system (e.g. yield ratio between irrigated rice vas rainfed rice). We have kept the terminology consistent in the revised SI.

**23. Table S5. Please be specific whether the values indicate either production share or area share.**

**Authors' Response:** Thanks. This is area shares. We revised it.

**24. Table S7. I feel difficulty in interpreting the cropping intensities. If I take double rice cropping in monsoonal Asia (India, Indonesia, Philippines etc.) as the example, it operates rainfed condition in wet season and irrigated condition in dry season (e.g., Koide et al. (2013) for Philippines). Therefore, in my interpretation, the sum of cropping intensities over rainfed and irrigated condition can exceed one. However, the cropping intensity values exceed one for both rainfed and irrigated conditions (for instance, cereals in Indonesia). Can you explain a bit more what is the definition and how you calculate these values?**

**Authors' Response:** The terms of irrigation/rainfed in the current study indicates farming systems rather than seasons. It means that the value of cropping intensity for a farming system indicates for a year around situation, regardless of the dry/wet seasons. The calculation of cropping intensity is based on the statistics in a few selected sampling areas: cropping intensity = harvested area / cropland area, and the values are further adjusted by expert judgements. We have added this clarification in the revised SI.

**25. SI Section 16. In the figures in this section, what is the area in gray for SPAM harvested area maps?**

**Authors' Response:** We do not apply grey for the legend of SPAM harvested area maps in this section. We guess you asked what does the area in light purple mean. The light purple means lower value while the pink and red colors mean higher value. We have added this clarification in the

revised SI.

**26. Thailand, SI Section 16. "F on 12" should read "F on 13", isn't it?**

**Authors' Response:** Yes. it should read "F on 13". Thanks for the carefully reading. We have double checked the entire manuscript including the SI to make sure there are no more technical mistakes.

**Authors' Response**: Thanks for the comment. This indeed is confusing. And your understanding is partially correct. Indeed we use rural population to calculate a prior for *subsistence portion* of a crop (Remember we break down each crop into 4 technology levels: irrigated, high-input rainfed, low-input rainfed and subsistence). 
[revised manuscript text omitted]

---

## Author Response (AR3)

Dear Editor,

We appreciated your extra time and the Reviewers' additional comment concerning the revised manuscript (ref. essd-2020-11). The comment was very helpful and instructive for revising and improving our paper. This time it is only a minor comment. We made more detailed elaborations on the assumption related to market accessibility and rural population. The detailed revisions are embedded in the manuscript and marked in red with the line numbers indicated in the response.

In addition, in accordant with our parallel paper entitled "A cultivated planet in 2010 – Part 1: The global synergy cropland map" which has been just published on ESSD, we slightly modified the title into "A cultivated planet in 2010 – Part 2: The global gridded agricultural production maps".

Again, thanks a lot for handling our paper.

Bets regards, Qiangyi Yu On behalf of the coauthors

Attachment: Authors' responses to Reviewer's Comments

Authors' Responses to Reviewer's Comment:

Reviewer's Comment: Regarding the explanations on market accessibility and rural population: I think I understand the explanations of the Authors, and they make sense - noting that this is a very crude proxy, indeed - , although in my opinion the actual writing of the paragraph l. 829-842 is not totally clear (some sentences are complex or weirdly articulated, or are formulated in such a way that they may appear to contradict each other - "In fact, ..." -, or use formulations that should be avoided such as "This" in the beginning of a sentence, etc).

If I understand well, you use rural population density because market accessibility datasets are not crop-specific, while using rural pop density as a proxy of market access allows cropspecific tuning depending on how crops are distributed across the four technology levels, is this correct?

If this is correct, and if there are still revisions requested by the second reviewer, or opportunity to upload a final manuscript, I would strongly encourage to try to express this paragraph more clearly. I know that the Authors have done efforts over several rounds to improve clarity, but overall, the explanations are still not always easy to understand, and, even if the paper is technically correct, making sure that the explanations are clear should improve the uptake from readers.

Authors' Response: Thanks for the additional comment on the second-round revision of the paper. Indeed the presentation on market accessibility and rural population was not crystal clear. In particular, the sentence of "In fact, ..." is contradicting and the use of "This" in the beginning of a sentence would further cause confusion. Following the comment, we have revised the paragraph accordingly. Now it looks much better. We hope it is not only technically correct but also clearly explained. The revisions can be found at Lines 814-825 and in red fonts.

**A cultivated planet in 2010 – Part 2: The global gridded agricultural production maps**

Qiangyi Yu1, Liangzhi You1,2, Ulrike Wood-Sichra2, Yating Ru2, Alison K. B. Joglekar3, Steffen Fritz4, Wei Xiong5, Miao Lu1, Wenbin Wu1,\*, Peng Yang1,\*

[revised manuscript text omitted]